# Efficient Model-Based Reinforcement Learning through Optimistic Policy Search and Planning

**Sebastian Curi** *
Department of Computer Science
ETH Zurich
`scuri@inf.ethz.ch`

**Felix Berkenkamp** *
Bosch Center for Artificial Intelligence
`felix.berkenkamp@de.bosch.com`

**Andreas Krause**
Department of Computer Science
ETH Zurich
`krausea@ethz.ch`

## Abstract

Model-based reinforcement learning algorithms with probabilistic dynamical models are amongst the most data-efficient learning methods. This is often attributed to their ability to distinguish between epistemic and aleatoric uncertainty. However, while most algorithms distinguish these two uncertainties for *learning* the model, they ignore it when *optimizing* the policy, which leads to greedy and insufficient exploration. At the same time, there are no practical solvers for optimistic exploration algorithms. In this paper, we propose a *practical* optimistic exploration algorithm (H-UCRL). H-UCRL reparameterizes the set of plausible models and *hallucinates* control directly on the *epistemic* uncertainty. By augmenting the input space with the *hallucinated* inputs, H-UCRL can be solved using standard greedy planners. Furthermore, we analyze H-UCRL and construct a general regret bound for well-calibrated models, which is provably sublinear in the case of Gaussian Process models. Based on this theoretical foundation, we show how optimistic exploration can be easily combined with state-of-the-art reinforcement learning algorithms and different probabilistic models. Our experiments demonstrate that optimistic exploration significantly speeds-up learning when there are penalties on actions, a setting that is notoriously difficult for existing model-based reinforcement learning algorithms.

## 1 Introduction

Model-Based Reinforcement Learning (MBRL) with probabilistic dynamical models can solve many challenging high-dimensional tasks with impressive sample efficiency (Chua et al., 2018). These algorithms alternate between two phases: first, they collect data with a policy and fit a model to the data; then, they simulate transitions with the model and optimize the policy accordingly. A key feature of the recent success of MBRL algorithms is the use of models that explicitly distinguish between *epistemic* and *aleatoric* uncertainty when learning a model (Gal, 2016). Aleatoric uncertainty is inherent to the system (noise), whereas epistemic uncertainty arises from data scarcity (Der Kiureghian and Ditlevsen, 2009). However, to optimize the policy, practical algorithms marginalize over both the aleatoric *and* epistemic uncertainty to optimize the expected performance under the current model, as in PILCO (Deisenroth and Rasmussen, 2011). This *greedy exploitation* can cause the optimization to

---

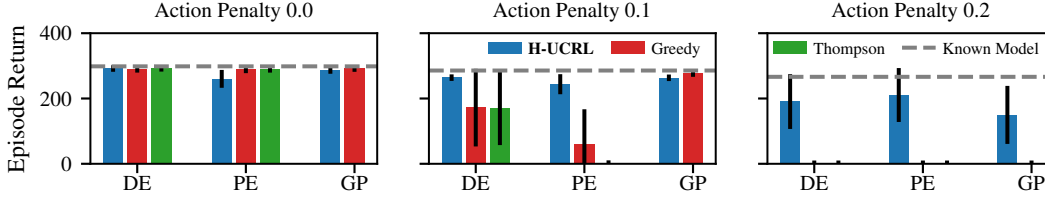

Figure 1: Final returns in an inverted pendulum swing-up task with sparse rewards. As the action penalty increases, exploration through noise is penalized and algorithms get stuck in a local minimum, where the pendulum is kept at the bottom position. Instead, H-UCRL is able to solve the swing-up task reliably. This holds for for all considered dynamical models: Deterministic- (DE) and Probabilistic Ensembles (PE) of neural networks as well as Gaussian Processes (GP) models.

get stuck in local minima even in simple environments like the swing-up of an inverted pendulum: In Fig. 1, all methods can solve this problem without action penalties (left plot). However, with action penalties, the expected reward (under the epistemic uncertainty) of swinging up the pendulum is low relative to the cost of the maneuver. Consequently, the greedy policy does not actuate the system at all and fails to complete the task. While optimistic exploration is a well-known remedy, there is currently a lack of efficient, principled means of incorporating optimism in deep MBRL.

**Contributions**     Our main contribution is a novel optimistic MBRL algorithm, *Hallucinated-UCRL* (H-UCRL), which can be applied together with state-of-the-art RL algorithms (Section 3). Our key idea is to *reduce optimistic exploration to greedy exploitation* by reparameterizing the model-space using a mean/epistemic variance decomposition. In particular, we augment the control space of the agent with *hallucinated* control actions that directly control the agent's *epistemic* uncertainty about the 1-step ahead transition dynamics (Section 3.1). We provide a general theoretical analysis for H-UCRL and prove sublinear regret bounds for the special case of Gaussian Process (GP) dynamics models (Section 3.2). Finally, we evaluate H-UCRL in high-dimensional continuous control tasks that shed light on when optimistic exploration outperforms greedy exploitation and Thompson sampling (Section 4). To the best of our knowledge, this is the first approach that successfully implements *optimistic* exploration with deep-MBRL.

**Related Work**     MBRL is a promising avenue towards applying RL methods to complex real-life decision problems due to its sample efficiency (Deisenroth et al., 2013). For instance, Kaiser et al. (2019) use MBRL to solve the Atari suite, whereas Kamthe and Deisenroth (2018) solve low-dimensional continuous-control problems using GP models and Chua et al. (2018) solve high-dimensional continuous-control problems using ensembles of probabilistic Neural Networks (NN). All these approaches perform *greedy exploitation* under the current model using a variant of PILCO (Deisenroth and Rasmussen, 2011). Unfortunately, greedy exploitation is *provably* optimal only in very limited cases such as linear quadratic regulators (LQR) (Mania et al., 2019).

Variants of *Thompson (posterior) sampling* are a common approach for *provable* exploration in reinforcement learning (Dearden et al., 1999). In particular, Osband et al. (2013) propose Thompson sampling for *tabular* MDPs. Chowdhury and Gopalan (2019) prove a $\tilde{\mathcal{O}}(\sqrt{T})$ regret bound for continuous states and actions for this theoretical algorithm, where $T$ is the number of episodes. However, Thompson sampling can be applied only when it is tractable to sample from the posterior distribution over dynamical models. For example, this is intractable for GP models with continuous domains. Moreover, Wang et al. (2018) suggest that approximate inference methods may suffer from variance starvation and limited exploration.

The *Optimism-in-the-Face-of-Uncertainty* (OFU) principle is a classical approach towards *provable* exploration in the theory of RL. Notably, Brafman and Tennenholtz (2003) present the R-Max algorithm for *tabular* MDPs, where a learner is optimistic about the reward function and uses the *expected* dynamics to find a policy. R-Max has a sample complexity of $\mathcal{O}(1/\epsilon^3)$, which translates to a sub-optimal regret of $\tilde{\mathcal{O}}(T^{2/3})$. Jaksch et al. (2010) propose the UCRL algorithm that is optimistic on the transition dynamics and achieves an optimal $\tilde{\mathcal{O}}(\sqrt{T})$ regret rate for tabular MDPs. Recently, Zanette and Brunskill (2019), Efroni et al. (2019), and Domingues et al. (2020) provide refined UCRL algorithms for tabular MDPs. When the number of states and actions increase, these *tabular algorithms are inefficient* and practical algorithms must exploit structure of the problem. The use of optimism in continuous state/action MDPs however is much less explored. Jin et al. (2019) present an

optimistic algorithm for *linear* MDPs and Abbasi-Yadkori and Szepesvári (2011) for linear quadratic regulators (LQR), both achieving $\tilde{\mathcal{O}}(\sqrt{T})$ regret. Finally, Luo et al. (2018) propose a trust-region UCRL meta-algorithm that asymptotically finds an optimal policy but it is intractable to implement.

Perhaps most closely related to our work, Chowdhury and Gopalan (2019) present GP-UCRL for continuous state and action spaces. They use optimistic exploration for the policy optimization step with dynamical models that lie in a Reproducing Kernel Hilbert Space (RKHS). However, as mentioned by Chowdhury and Gopalan (2019), their algorithm is intractable to implement and cannot be used in practice. Instead, we build on an implementable but expensive strategy that was heuristically suggested by Moldovan et al. (2015) for planning on *deterministic* systems and develop a principled and highly efficient optimistic exploration approach for deep MBRL. Partial results from this paper appear in Berkenkamp (2019, Chapter 5).

**Concurrent Work** Kakade et al. (2020) build tight confidence intervals for our problem setting based on information theoretical quantities. However, they assume an optimization oracle and do not provide a practical implementation (their experiments use Thompson sampling). Abeille and Lazaric (2020) propose an equivalent algorithm to H-UCRL in the context of LQR and proved that the planning problem can be solved efficiently. In the same spirit as H-UCRL, Neu and Pike-Burke (2020) reduce intractable optimistic exploration to greedy planning using well-selected reward bonuses. In particular, they prove an equivalence between optimistic reinforcement learning and exploration bonus (Azar et al., 2017) for tabular and linear MDPs. How to generalize these exploration bonuses to our setting is left for future work.

## 2    Problem Statement and Background

We consider a stochastic environment with states $\mathbf{s} \in \mathcal{S} \subseteq \mathbb{R}^p$, actions $\mathbf{a} \in \mathcal{A} \subset \mathbb{R}^q$ within a compact set $\mathcal{A}$, and *i.i.d.*, additive transition noise $\boldsymbol{\omega}_n \in \mathbb{R}^p$. The resulting transition dynamics are

$$\mathbf{s}_{n+1} = f(\mathbf{s}_n, \mathbf{a}_n) + \boldsymbol{\omega}_n \tag{1}$$

with $f \colon \mathcal{S} \times \mathcal{A} \to \mathcal{S}$. For tractability we assume continuity of $f$, which is common for any method that aims to approximate $f$ with a continuous model (such as neural networks). In addition, we also assume sub-Gaussian noise $\boldsymbol{\omega}$, which includes any zero-mean distribution with bounded support and Gaussians. This assumption allows the noise to depend on states and actions.

**Assumption 1** (System properties)**.** The true dynamics $f$ in (1) are $L_f$-Lipschitz continuous and, for all $n \geq 0$, the elements of the noise vector $\boldsymbol{\omega}_n$ are *i.i.d.* $\sigma$-sub-Gaussian.

### 2.1    Model-based Reinforcement Learning

**Objective** Our goal is to control the stochastic system (1) optimally in an *episodic* setting over a finite time horizon $N$. To control the system, we use any deterministic policy $\pi_n \colon \mathcal{S} \to \mathcal{A}$ from a set $\Pi$ that selects actions $\mathbf{a}_n = \pi_n(\mathbf{s}_n)$ given the current state. For ease of notation, we assume that the system is reset to a known state $\mathbf{s}_0$ at the end of each episode, that there is a known reward function $r \colon \mathcal{S} \times \mathcal{A} \to \mathbb{R}$, and we omit the dependence of the policy on the time index. Our results, easily extend to known initial state distributions and unknown reward functions using standard techniques (see Chowdhury and Gopalan (2019)). For any dynamical model $\tilde{f} \colon \mathcal{S} \times \mathcal{A} \to \mathcal{S}$ (e.g., $f$ in (1)), the performance of a policy $\pi$ is the total reward collected during an episode in expectation over the transition noise $\boldsymbol{\omega}$,

$$J(\tilde{f}, \pi) = \mathbb{E}_{\tilde{\boldsymbol{\omega}}_{0:N-1}} \left[ \sum_{n=0}^{N} r(\tilde{\mathbf{s}}_n, \pi(\tilde{\mathbf{s}}_n)) \,\Big|\, \mathbf{s}_0 \right], \quad \text{s.t. } \tilde{\mathbf{s}}_{n+1} = \tilde{f}(\tilde{\mathbf{s}}_n, \pi(\tilde{\mathbf{s}}_n)) + \tilde{\boldsymbol{\omega}}_n. \tag{2}$$

Thus, we aim to find the optimal policy $\pi^*$ for the true dynamics $f$ in (1),

$$\pi^* = \operatorname*{argmax}_{\pi \in \Pi} J(f, \pi). \tag{3}$$

If the dynamics $f$ were known, (3) would be a standard stochastic optimal control problem. However, in model-based reinforcement learning we do *not* know the dynamics $f$ and have to learn them online.

**Model-learning** We consider algorithms that iteratively select policies $\pi_t$ at each iteration/episode $t$ and conduct a single rollout on the real system (1). That is, starting with $\mathcal{D}_1 = \emptyset$, at each iteration $t$ we apply the selected policy $\pi_t$ to (1) and collect transition data $\mathcal{D}_{t+1} = \{(\mathbf{s}_{n-1,t}, \mathbf{a}_{n-1,t}), \mathbf{s}_{n,t}\}_{n=1}^{N}$.

---

**Algorithm 1** Model-based Reinforcement Learning

---

**Inputs:** Calibrated dynamical model, reward function $r(\mathbf{s}, \mathbf{a})$, horizon $N$, initial state $\mathbf{s}_0$
1: **for** $t = 1, 2, \ldots$ **do**
2:      Select $\pi_t$ based on (4), (5), or (7)
3:      Reset the system to $\mathbf{s}_{0,t} = \mathbf{s}_0$
4:      **for** $n = 1, \ldots, N$ **do**
5:          $\mathbf{s}_{n,t} = f(\mathbf{s}_{n-1,t}, \pi_t(\mathbf{s}_{n-1,t})) + \boldsymbol{\omega}_{n-1,t}$
6:      Update statistical dynamical model with the $N$ observed state transitions in $\mathcal{D}_t$.

---

We use a statistical model to estimate which dynamical models $\tilde{f}$ are compatible with the data in $\mathcal{D}_{1:t} = \cup_{0 < i \le t} \mathcal{D}_i$. This can either come from a frequentist model with mean and confidence estimate $\boldsymbol{\mu}_t(\mathbf{s}, \mathbf{a})$ and $\boldsymbol{\Sigma}_t(\mathbf{s}, \mathbf{a})$, or from a Bayesian perspective that estimates a posterior distribution $p(\tilde{f} \mid \mathcal{D}_{1:t})$ over dynamical models $\tilde{f}$ and defines $\boldsymbol{\mu}_t(\cdot) = \mathbb{E}_{\tilde{f} \sim p(\tilde{f} \mid \mathcal{D}_{1:t})}[\tilde{f}(\cdot)]$ and $\boldsymbol{\Sigma}_t^2(\cdot) = \mathrm{Var}[\tilde{f}(\cdot)]$, respectively. Either way, we require the model to be well-calibrated:

**Assumption 2** (Calibrated model). The statistical model is *calibrated* w.r.t. $f$ in (1), so that with $\boldsymbol{\sigma}_t(\cdot) = \mathrm{diag}(\boldsymbol{\Sigma}_t(\cdot))$ there exists a sequence $\beta_t \in \mathbb{R}_{>0}$ such that, with probability at least $(1 - \delta)$, it holds jointly for all $t \ge 0$ and $\mathbf{s}, \mathbf{a} \in \mathcal{S} \times \mathcal{A}$ that $|f(\mathbf{s}, \mathbf{a}) - \boldsymbol{\mu}_t(\mathbf{s}, \mathbf{a})| \le \beta_t \boldsymbol{\sigma}_t(\mathbf{s}, \mathbf{a})$, elementwise.

Popular choices for statistical dynamics models include *Gaussian Processes (GP)* (Rasmussen and Williams, 2006) and *Neural Networks (NN)* (Anthony and Bartlett, 2009). GP models naturally differentiate between aleatoric noise and epistemic uncertainty and are effective in the low-data regime. They provably satisfy Assumption 2 when the true function $f$ has finite norm in the RKHS induced by the covariance function. In contrast to GP models, NNs potentially scale to larger dimensions and data sets. From a practical perspective, NN models that differentiate aleatoric from epistemic uncertainty can be efficiently implemented using Probabilistic Ensembles (PE) (Lakshminarayanan et al., 2017). Deterministic Ensembles (DE) are also commonly used but they do not represent aleatoric uncertainty correctly (Chua et al., 2018). NN models are not calibrated in general, but can be re-calibrated to satisfy Assumption 2 (Kuleshov et al., 2018). State-of-the-art methods typically learn models so that the one-step predictions in Assumption 2 combine to yield good predictions for trajectories (Archer et al., 2015; Doerr et al., 2018; Curi et al., 2020).

## 2.2 Exploration Strategies

Ultimately the performance of our algorithm depends on the choice of $\pi_t$. We now provide a unified overview of existing exploration schemes and summarize the MBRL procedure in Algorithm 1.

**Greedy Exploitation**    In practice, one of the most commonly used algorithms is to select the policy $\pi_t$ that greedily maximizes the expected performance over the aleatoric uncertainty *and* epistemic uncertainty induced by the dynamical model. Other exploration strategies, such as dithering (e.g., epsilon-greedy, Boltzmann exploration) (Sutton and Barto, 1998) or certainty equivalent control (Bertsekas et al., 1995, Chapter 6.1), can be grouped into this class. The greedy policy is

$$\pi_t^{\mathrm{Greedy}} = \operatorname*{argmax}_{\pi \in \Pi} \mathbb{E}_{\tilde{f} \sim p(\tilde{f} \mid \mathcal{D}_{1:t})} \Big[ J(\tilde{f}, \pi) \Big]. \tag{4}$$

For example, PILCO (Deisenroth and Rasmussen, 2011) and GP-MPC (Kamthe and Deisenroth, 2018) use moment matching to approximate $p(\tilde{f} \mid \mathcal{D}_{1:t})$ and use *greedy* exploitation to optimize the policy. Likewise, PETS-1 and PETS-$\infty$ from Chua et al. (2018) also lie in this category, in which $p(\tilde{f} \mid \mathcal{D}_{1:t})$ is represented via ensembles. The main difference between PETS-$\infty$ and other algorithms is that PETS-$\infty$ ensures consistency by sampling a function per rollout, whereas PETS-1, PILCO, and GP-MPC sample a new function at each time step for computational reasons. We show in Appendix A that, in the bandit setting, the exploration is only driven by noise and optimization artifacts. In the tabular RL setting, dithering takes an exponential number of episodes to find an optimal policy (Osband et al., 2014). As such, it is *not* an efficient exploration scheme for reinforcement learning. Nevertheless, for some specific reward and dynamics structure, such as linear-quadratic control, greedy exploitation indeed achieves no-regret (Mania et al., 2019). However, it is the most common exploration strategy and many practical algorithms to efficiently solve the optimization problem (4) exist (cf. Section 3.1).

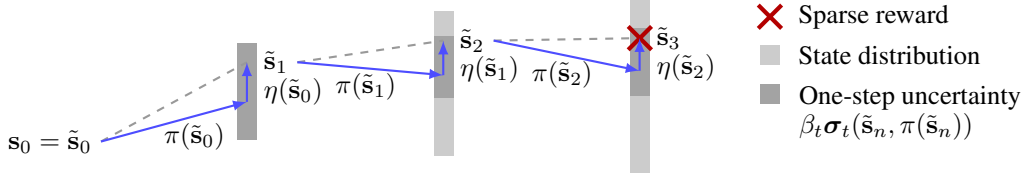

Figure 2: Illustration of the optimistic trajectory $\tilde{\mathbf{s}}_n$ from H-UCRL. The policy $\pi$ is used to choose the next-state distribution, and the variables $\eta$ to choose the next state optimistically inside the one-step confidence interval (dark grey bars). The true dynamics is contained inside the light grey confidence intervals, but, after the first step, not necessarily inside the dark grey bars. Even when the expected reward w.r.t. the epistemic uncertainty is small (red cross compared to light grey bar), H-UCRL efficiently finds the high-reward region (red cross). Instead, greedy exploitation strategies fail.

**Thompson Sampling**   A theoretically grounded exploration strategy is Thompson sampling, which optimizes the policy w.r.t. a single model that is sampled from $p(\tilde{f} \mid \mathcal{D}_{1:t})$ at every episode. Formally,

$$\tilde{f}_t \sim p(\tilde{f} \mid \mathcal{D}_{1:t}), \quad \pi_t^{\text{TS}} = \underset{\pi \in \Pi}{\operatorname{argmax}} J(\tilde{f}_t, \pi). \tag{5}$$

This is different to PETS-$\infty$, as the former algorithm optimizes w.r.t. the average of the (consistent) model trajectories instead of a single model. In general, it is intractable to sample from $p(\tilde{f} \mid \mathcal{D}_{1:t})$. Nevertheless, after the sampling step, the optimization problem is equivalent to greedy exploitation of the sampled model. Thus, the same optimization algorithms can be used to solve (4) and (5).

**Upper-Confidence Reinforcement Learning (UCRL)**   The final exploration strategy we address is UCRL exploration (Jaksch et al., 2010), which optimizes jointly over policies and models inside the set $\mathcal{M}_t = \{\tilde{f} \mid |\tilde{f}(\mathbf{s}, \mathbf{a}) - \boldsymbol{\mu}_t(\mathbf{s}, \mathbf{a})| \leq \beta_t \boldsymbol{\sigma}_t(\mathbf{s}, \mathbf{a}) \, \forall \mathbf{s}, \mathbf{a} \in \mathcal{S} \times \mathcal{A}\}$ that contains all statistically-plausible models compatible with Assumption 2. The UCRL algorithm is

$$\pi_t^{\text{UCRL}} = \underset{\pi \in \Pi}{\operatorname{argmax}} \, \underset{\tilde{f} \in \mathcal{M}_t}{\max} J(\tilde{f}, \pi). \tag{6}$$

Instead of greedy exploitation, these algorithms optimize an optimistic policy that maximizes performance over all plausible models. Unfortunately, this joint optimization is in general *intractable* and algorithms designed for greedy exploitation (4) do *not* generally solve the UCRL objective (6).

## 3   Hallucinated Upper Confidence Reinforcement Learning (H-UCRL)

We propose a practical variant of the UCRL-exploration (6) algorithm. Namely, we reparameterize the functions $\tilde{f} \in \mathcal{M}_t$ as $\tilde{f} = \boldsymbol{\mu}_{t-1}(\mathbf{s}, \mathbf{a}) + \beta_{t-1}\boldsymbol{\Sigma}_{t-1}(\mathbf{s}, \mathbf{a})\eta(\mathbf{s}, \mathbf{a})$, for some function $\eta \colon \mathbb{R}^p \times \mathbb{R}^q \to [-1, 1]^p$. This transformation is similar in spirit to the re-parameterization trick from Kingma and Welling (2013), except that $\eta(\mathbf{s}, \mathbf{a})$ are functions. The key insight is that instead of optimizing over dynamics in $\tilde{f} \in \mathcal{M}_t$ as in UCRL, it suffices to optimize over the functions $\eta(\cdot)$. We call this algorithm H-UCRL, formally:

$$\pi_t^{\text{H-UCRL}} = \underset{\pi \in \Pi}{\operatorname{argmax}} \, \underset{\eta(\cdot) \in [-1,1]^p}{\max} J(\tilde{f}, \pi), \text{s.t. } \tilde{f}(\mathbf{s}, \mathbf{a}) = \boldsymbol{\mu}_{t-1}(\mathbf{s}, \mathbf{a}) + \beta_{t-1}\boldsymbol{\Sigma}_{t-1}(\mathbf{s}, \mathbf{a})\eta(\mathbf{s}, \mathbf{a}). \tag{7}$$

At a high level, the policy $\pi$ acts on the *inputs* (actions) of the dynamics and chooses the next-state distribution. In turn, the optimization variables $\eta$ act in the *outputs* of the dynamics to select the most-optimistic outcome from within the confidence intervals. We call the optimization variables the *hallucinated* controls as the agent hallucinates control authority to find the most-optimistic model.

The H-UCRL algorithm *does not explicitly propagate uncertainty* over the horizon. Instead, it does so *implicitly* by using the pointwise uncertainty estimates from the model to recursively plan an optimistic trajectory, as illustrated in Fig. 2. This has the practical advantage that the model only has to be well-calibrated for 1-step predictions and not $N$-step predictions. In practice, the parameter $\beta_t$ trades off between exploration and exploitation.

### 3.1   Solving the Optimization Problem

Problem (7) is still intractable as it requires to optimize over general functions. The *crucial* insight is that we can make the H-UCRL algorithm (7) practical by optimizing over a smaller class

---

**Algorithm 2** H-UCRL combining Optimistic Policy Search and Planning

---

**Inputs:** Mean $\boldsymbol{\mu}(\cdot, \cdot)$ and variance $\boldsymbol{\Sigma}^2(\cdot, \cdot)$, parametric policies $\pi_\theta(\cdot)$, $\eta_\theta(\cdot)$, parametric critic $Q_\vartheta(\cdot)$, horizon $N$, policy search algorithm `PolicySearch`, online planning algorithm `Plan`,

1: **for** $t = 1, 2, \ldots$ **do**
2: $\quad (\pi_{\theta,t}, \eta_{\theta,t}), Q_{\vartheta,t} \leftarrow \texttt{PolicySearch}(\boldsymbol{\mu}_{t-1}; \boldsymbol{\Sigma}^2_{t-1}; (\pi_{\theta,t-1}, \eta_{\theta,t-1}))$
3: $\quad$ **for** $n = 1, \ldots, N$ **do**
4: $\quad\quad (\mathbf{a}_{n-1,t}, \mathbf{a}'_{n-1,t}) = \texttt{Plan}(\mathbf{s}_{n-1,t}; \boldsymbol{\mu}_{t-1}; \boldsymbol{\Sigma}^2_{t-1}; (\pi_{\theta,t}, \eta_{\theta,t}), Q_\vartheta)$
5: $\quad\quad \mathbf{s}_{n,t} = f(\mathbf{s}_{n-1,t}, \mathbf{a}_{n-1,t}) + \boldsymbol{\omega}_{n-1,t}$
6: $\quad$ Update statistical dynamical model with the $N$ observed state transitions in $\mathcal{D}_t$.

---

of functions $\eta$. In Appendix E, we prove that it suffices to optimize over Lipschitz-continuous bounded functions instead of general bounded functions. Therefore, we can optimize jointly over policies and Lipschitz-continuous, bounded *functions* $\eta(\cdot)$. Furthermore, we can re-write $\eta(\tilde{\mathbf{s}}_n, \tilde{\mathbf{a}}_n) = \eta(\tilde{\mathbf{s}}_n, \pi(\tilde{\mathbf{s}}_{n,t})) = \eta(\tilde{\mathbf{s}}_{n,t})$. This allows to reduce the intractable optimistic problem (7) to *greedy exploitation* (4): We simply treat $\eta(\cdot) \in [-1, 1]^p$ as an additional *hallucinated* control input that has no associated control penalties and can exert as much control as the current *epistemic* uncertainty that the model affords. With this observation in mind, H-UCRL greedily exploits a *hallucinated* system with the extended dynamics $\tilde{f}$ in (7) and a corresponding augmented control policy $(\pi, \eta)$. This means that we can now use the *same* efficient MBRL approaches for optimistic exploration that were previously restricted to greedy exploitation and Thompson sampling (albeit on a slightly larger action space, since the dimension of the action space increases from $q$ to $q + p$).

In practice, if we have access to a greedy oracle $\pi = \texttt{GreedyOracle}(f)$, we simply access it using $\pi, \eta = \texttt{GreedyOracle}(\boldsymbol{\mu}_{t-1} + \beta_{t-1}\boldsymbol{\Sigma}_{t-1}\eta)$. Broadly speaking, greedy oracles are implemented using offline-policy search or online planning algorithms. Next, we discuss how to use these strategies independently to solve the H-UCRL planning problem (7). For a detailed discussion on how to augment common algorithms with hallucination, see Appendix C.

**Offline Policy Search** is any algorithm that optimizes a parametric policy to maximize performance of the current dynamical model. As inputs, it takes the dynamical model and a parametric family for the policy and the critic (the value function). It outputs the optimized policy and the corresponding critic of the optimized policy. These algorithms have fast inference time and scale to large dimensions but can suffer from model bias and inductive bias from the parametric policies and critics (van Hasselt et al., 2019).

**Online Planning** or Model Predictive Control (Morari and H. Lee, 1999) is a local planning algorithm that outputs the best action for the current state. This method solves the H-UCRL planning problem (7) in a receding-horizon fashion. The planning horizon is usually shorter than $N$ and the reward-to-go is bootstrapped using a terminal reward. In most cases, however, this terminal reward is unknown and must be learned (Lowrey et al., 2019). As the planner observes the *true* transitions during deployment, it suffers less from model errors. However, its running time is too slow for real-time implementation.

**Combining Offline Policy Search with Online Planning**    In Algorithm 2, we propose to combine the best of both worlds to solve the H-UCRL planning problem (7). In particular, Algorithm 2 takes as inputs a policy search algorithm and a planning algorithm. After each episode, it optimizes parametric (e.g. neural networks) control and hallucination policies $(\pi_\theta, \eta_\theta)$ using the policy search algorithm. As a by-product of the policy search algorithm we have the *learned* critic $Q_\vartheta$. At deployment, the planning algorithm returns the true and hallucinated actions $(a, a')$, and we only execute the true action $a$ to the true system. We initialize the planning algorithm using the learned policies $(\pi_\theta, \eta_\theta)$ and use the *learned* critic to bootstrap at the end of the prediction horizon. In this way, we achieve the best of both worlds. The policy search algorithm accelerates the planning algorithm by shortening the planning horizon with the learned critic and by using the learned policies to warm-start the optimization. The planning algorithm reduces the model-bias that a pure policy search algorithm has.

### 3.2    Theoretical Analysis

In this section, we analyze the H-UCRL algorithm (7). A natural quality criterion to evaluate exploration schemes is the *cumulative regret* $R_T = \sum_{t=1}^{T} |J(f, \pi^*) - J(f, \pi_t)|$, which is the

difference in performance between the optimal policy $\pi^*$ and $\pi_t$ on the true system $f$ over the run of the algorithm (Chowdhury and Gopalan, 2019). If we can show that $R_T$ is sublinear in $T$, then we know that the performance $J(f, \pi_t)$ of our chosen policies $\pi_t$ converges to the performance of the optimal policy $\pi^*$. We first introduce the final assumption for the results in this section to hold.

**Assumption 3** (Continuity). *The functions $\boldsymbol{\mu}_t$ and $\boldsymbol{\sigma}_t$ are $L_\mu$ and $L_\sigma$ Lipschitz continuous, any policy $\pi \in \Pi$ is $L_\pi$-Lipschitz continuous and the reward $r(\cdot, \cdot)$ is $L_r$-Lipschitz continuous.*

Assumption 3 is not restrictive. NN with Lipschitz-continuous non-linearities or GP with Lipschitz-continuous kernels output Lipschitz-continuous predictions (see Appendix G). Furthermore, we are free to choose the policy class $\Pi$, and most reward functions are either quadratic or tolerance functions (Tassa et al., 2018). Discontinuous reward functions are generally very difficult to optimize.

**Model complexity**   In general, we expect that $R_T$ depends on the complexity of the statistical model in Assumption 2. If we can quickly estimate the true model using a few data-points, then the regret would be lower than if the model is slower to learn. To account for these differences, we construct the following complexity measure over a given set $\mathcal{S}$ and $\mathcal{A}$,

$$I_T(\mathcal{S}, \mathcal{A}) = \max_{\mathcal{D}_1, \ldots, \mathcal{D}_T \subset \mathcal{S} \times \mathcal{S} \times \mathcal{A}, |\mathcal{D}_t| = N} \sum_{t=1}^{T} \sum_{\mathbf{s}, \mathbf{a} \in \mathcal{D}_t} \|\boldsymbol{\sigma}_{t-1}(\mathbf{s}, \mathbf{a})\|_2^2. \qquad (8)$$

While in general impossible to compute, this complexity measure considers the "worst-case" datasets $\mathcal{D}_1$ to $\mathcal{D}_T$, with $|\mathcal{D}_t| = N$ elements each, that we could collect at each iteration of Algorithm 1 in order to maximize the predictive uncertainty of our statistical model. Intuitively, if $\boldsymbol{\sigma}(\mathbf{s}, \mathbf{a})$ shrinks sufficiently quickly after observing a transition $(\cdot, \mathbf{s}, \mathbf{a})$ and if the model generalizes well over $\mathcal{S} \times \mathcal{A}$, then (8) will be small. In contrast, if our model does not learn or generalize at all, then $I_T$ will be $\mathcal{O}(TNp)$ and we cannot hope to succeed in finding the optimal policy. For the special case of Gaussian process (GP) models, we show that $I_T$ is indeed sublinear in the following.

**General regret bound**   The true sequence of states $\mathbf{s}_{n,t}$ at which we obtain data during our rollout in Line 5 of Algorithm 1 lies somewhere withing the light-gray shaded state distribution with epistemic uncertainty in Fig. 2. While this is generally difficult to compute, we can bound it in terms of the predictive variance $\boldsymbol{\sigma}_{t-1}(\mathbf{s}_{n,t}, \pi_t(\mathbf{s}_{n,t}))$, which is directly related to $I_T$. However, the optimistically planned trajectory instead depends on $\boldsymbol{\sigma}_{t-1}(\tilde{\mathbf{s}}_{n,t}, \pi(\tilde{\mathbf{s}}_{n,t}))$ in (7), which enables policy optimization without explicitly constructing the state distribution. How the predictive uncertainties of these two trajectories relate depends on the generalization properties of our statistical model; specifically on $L_\sigma$ in Assumption 3. We can use this observation to obtain the following bound on $R_T$:

**Theorem 1.** *Under Assumptions 1–3 let $\mathbf{s}_{n,t} \in \mathcal{S}$ and $\mathbf{a}_{n,t} \in \mathcal{A}$ for all $n, t > 0$. Then, for all $T \geq 1$, with probability at least $(1 - \delta)$, the regret of H-UCRL in (7) is at most $R_T \leq \mathcal{O}\left(L_\sigma^N \beta_{T-1}^N \sqrt{TN^3 \, I_T(\mathcal{S}, \mathcal{A})}\right)$.*

We provide a proof of Theorem 1 in Appendix D. The theorem ensures that, if we evaluate optimistic policies according to (7), we eventually achieve performance $J(f, \pi_t)$ arbitrarily close to the optimal performance of $J(f, \pi^*)$ if $I_T(\mathcal{S}, \mathcal{A})$ grows at a rate smaller than $T$. As one would expect, the regret bound in Theorem 1 depends on constant factors like the prediction horizon $N$, the relevant Lipschitz constants of the dynamics, policy, reward, and the predictive uncertainty. The dependence on the dimensionality of the state space $p$ is hidden inside $I_T$, while $\beta_t$ is a function of $\delta$.

**Gaussian Process Models**   For the bound in Theorem 1 to be useful, we must show that $I_T$ is sublinear. Proving this is impossible for general models, but can be proven for GP models. In particular, we show in Appendix H that $I_T$ is bounded by the worst-case mutual information (information capacity) of the GP model. Srinivas et al. (2012); Krause and Ong (2011) derive upper-bounds for the information capacity for commonly-used kernels. For example, when we use their results for independent GP models with squared exponential kernels for each component $[f(\mathbf{s}, \mathbf{a})]_i$, we obtain a regret bound $\mathcal{O}\left((1 + B_f)^N L_\sigma^N N^2 \sqrt{T} (p^2(p+q) \log(pTN))^{(N+1)/2}\right)$, where $B_f$ is a bound on the functional complexity of the function $f$. Specifically, $B_f$ is the norm of $f$ in the RKHS that corresponds to the kernel.

A similar optimistic exploration scheme was analyzed by Chowdhury and Gopalan (2019), but for an algorithm that is not implementable as we discussed at the beginning of Section 3. Their exploration scheme *depends* on the (generally unknown) Lipschitz constant of the value function, which corresponds to knowing $L_f$ *a priori* in our setting. While this is a restrictive and impractical requirement, we show in Appendix H.3 that under this assumption we can improve the dependence

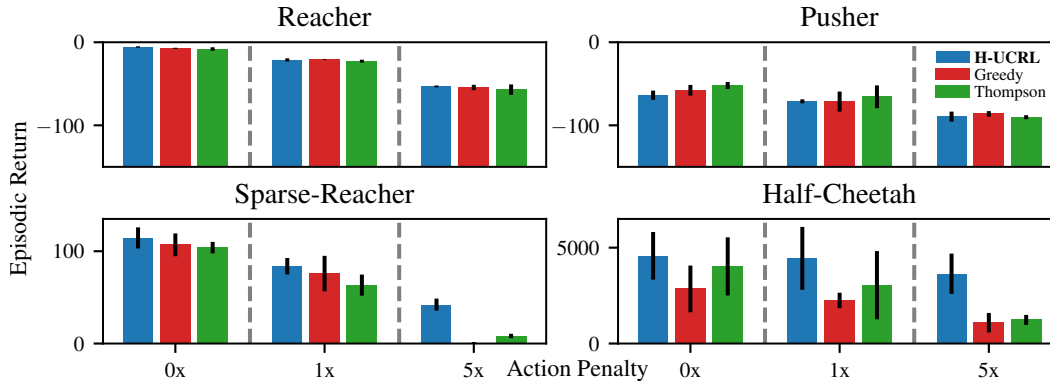

Figure 3: Mean final episodic returns on Mujoco tasks averaged over five different random seeds. For Reacher and Pusher (50 episodes), all exploration strategies perform equally. For Sparse-Reacher (50 episodes) and Half-Cheetah (250 episodes), H-UCRL outperforms other exploration algorithms.

on $L_\sigma^N \beta_T^N$ in the regret bound in Theorem 1 to $(L_f \beta_T)^{1/2}$. This matches the bounds derived by Chowdhury and Gopalan (2019) up to constant factors. Thus we can consider the regret term $L_\sigma^N \beta_T^N$ to be the additional cost that we have to pay for a practical algorithm.

**Unbounded domains**    We assume that the domain $\mathcal{S}$ is compact in order to bound $I_T$ for GP models, which enables a convenient analysis and is also used by Chowdhury and Gopalan (2019). However, it is incompatible with Assumption 1, which allows for potentially unbounded noise $\boldsymbol{\omega}$. While this is a technical detail, we formally prove in Appendix I that we can bound the domain with high probability within a norm-ball of radius $b_t = \mathcal{O}(L_f^N N p \log(N t^2))$. For GP models with a squared exponential kernel, we analyze $I_T$ in this setting and show that the regret bound only increases by a polylog factor.

## 4   Experiments

Throughout the experiments, we consider reward functions of the form $r(\mathbf{s}, \mathbf{a}) = r_{\text{state}}(\mathbf{s}) - \rho c_{\text{action}}(\mathbf{a})$, where $r_{\text{state}}(\mathbf{s})$ is the reward for being in a "good" state, and $\rho \in [0, \infty)$ is a parameter that scales the action costs $c_{\text{action}}(\mathbf{a})$. We evaluate how H-UCRL, greedy exploitation, and Thompson sampling perform for different values of $\rho$ in different Mujoco environments (Todorov et al., 2012). We expect greedy exploitation to struggle for larger $\rho$, whereas H-UCRL and Thompson sampling should perform well. As modeling choice, we use 5-head probabilistic ensembles as in Chua et al. (2018). For greedy exploitation, we sample the next-state from the ensemble mean and covariance (PE-DS algorithm in Chua et al. (2018)). We use ensemble sampling (Lu and Van Roy, 2017) to approximate Thompson sampling. For H-UCRL, we follow Lakshminarayanan et al. (2017) and use the ensemble mean and covariance as the next-state predictive distribution. For more experimental details and learning curves, see Appendix B. We provide an open-source implementation of our method, which is available at `http://github.com/sebascuri/hucrl`.

**Sparse Inverted Pendulum**    We first investigate a swing-up pendulum with sparse rewards. In this task, the policy must perform a complex maneuver to swing the pendulum to the upwards position. A policy that does not act obtains zero state rewards but suffers zero action costs. Slightly moving the pendulum still has zero state reward but the actions are penalized. Hence, a zero-action policy is locally optimal, but it fails to complete the task. We show the results in Fig. 1: With no action penalty, all exploration methods perform equally well – the randomness is enough to explore and find a quasi-optimal sequence. For $\rho = 0.1$, greedy exploitation struggles: sometimes it finds the swing-up sequence, which explains the large error bars. Finally, for $\rho = 0.2$ only H-UCRL is able to successfully swing up the pendulum.

**7-DOF PR2 Robot**    Next, we evaluate how H-UCRL performs in higher-dimensional problems. We start by comparing the Reacher and Pusher environments proposed by Chua et al. (2018). We plot the results in the upper left and right subplots in Fig. 3. The Reacher has to move the end-effector towards a goal that is randomly sampled at the beginning of each episode. The Pusher has to push an object towards a goal. The rewards and costs in these environments are quadratic. All exploration

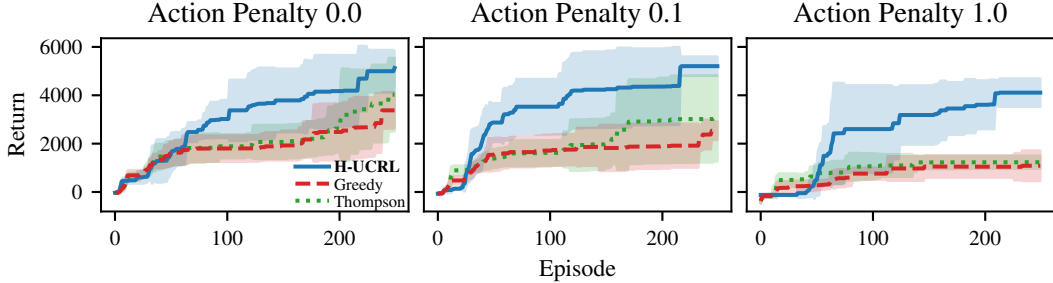

Figure 4: Learning curves in Half-Cheetah environment. For all action penalties, H-UCRL learns faster than greedy and Thompson sampling strategies. For larger action penalties, greedy and Thompson lead to insufficient exploration and get stuck in local optima with poor performance.

strategies achieve state-of-the-art performance, which seems to indicate that greedy exploitation is indeed sufficient for these tasks. Presumably, this is due to the over-actuated dynamics and the reward structure. This is in line with the theoretical results for linear-quadratic control by Mania et al. (2019).

To test this hypothesis, we repeat the Reacher experiment with a sparse reward function. We plot the results in the lower left plot of Fig. 3. The state reward has a positive signal when the end-effector is close to the goal and the action has a non-negative signal when it is close to zero. Here we observe that H-UCRL outperforms alternative methods, particularly for larger action penalties.

**Half-Cheetah**    Our final experiment demonstrates H-UCRL on a common deep-RL benchmark, the Half-Cheetah. The goal is to make the cheetah run forward as fast as possible. The actuators have to interact in a complex manner to achieve running. In Fig. 4, we can see a clear advantage of using H-UCRL at different action penalties, even at zero. This indicates that H-UCRL not only addresses action penalties, but also explores through complex dynamics. For the sake of completeness, we also show the final returns in the lower right plot of Fig. 3.

**H-UCRL vs. Thompson Sampling**    In Appendix B.4, we carry out extensive experiments to empirically evaluate why Thompson sampling fails in our setting. Phan et al. (2019) in the Bandit Setting and Kakade et al. (2020) in the RL setting also report that approximate Thompson sampling fails unless strong modelling priors are used. We believe that the poor performance of Thompson sampling relative to H-UCRL suggests that the models that we use are sufficient to construct well-calibrated 1-step ahead confidence intervals, but do not comprise a rich enough posterior distribution for Thompson sampling. As an example, in H-UCRL we use the five members of the ensemble to construct the 1-step ahead confidence interval at every time-step. On the other hand, in Thompson sampling we sample a *single* model from the *approximate* posterior for the full horizon. It is possible that in some regions of the state-space one member is more optimistic than others, and in a different region the situation reverses. This is not only a property of ensembles, but also other approximate models such as random-feature GP models (c.f. Appendix B.4.5) exhibit the same behaviour. This discussion highlights the advantage of H-UCRL over Thompson sampling using deep neural networks: H-UCRL only requires calibrated 1-step ahead confidence intervals, and we know how to construct them (c.f. Malik et al. (2019)). Instead, Thompson sampling requires posterior models that are calibrated throughout the full trajectory. Due to the multi-step nature of the problem, constructing *scalable* approximate posteriors that have enough variance to sufficiently explore is still an open problem.

## 5    Conclusions

In this work, we introduced H-UCRL: a practical optimistic-exploration algorithm for deep MBRL. The key idea is a reduction from (generally intractable) optimistic exploration to greedy exploitation in an augmented policy space. Crucially, this insight enables the use of highly effective standard MBRL algorithms that previously were restricted to greedy exploitation and Thompson sampling. Furthermore, we provided a theoretical analysis of H-UCRL and show that it attains sublinear regret for some models. In our experiments, H-UCRL performs as well or better than other exploration algorithms, achieving state-of-the-art performance on the evaluated tasks.

## Broader Impact

Improving sample efficiency is one of the key bottlenecks in applying reinforcement learning to real-world problems with potential major societal benefit such as personal robotics, renewable energy systems, medical decisions making, etc. Thus, algorithmic and theoretical contributions as presented in this paper can help decrease the cost associated with optimizing RL policies. Of course, the overall RL framework is so general that potential misuse cannot be ruled out.

## Acknowledgments and Disclosure of Funding

This project has received funding from the European Research Council (ERC) under the European Unions Horizon 2020 research and innovation program grant agreement No 815943. It was also supported by a fellowship from the Open Philanthropy Project.

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
