[Supplementary Material]

# Appendix

The following table provides an overview of the appendix.

## Table of Contents

## A Expected Performance for Exploration in the Bandit Setting

In practice, one of the most commonly used exploration strategies is to select $\boldsymbol{\theta}_t$ in order to maximize the expected performance over the aleatoric uncertainty *and epistemic uncertainty* induced by the Gaussian process model.

We consider the simplest possible case that still allows for nonlinear dynamics. That is, we consider a system with zero-mean noise, i.e., $\mathbb{E}[\boldsymbol{\omega}_n = \mathbf{0}]$ for all time steps $n \geq 0$. In addition, we consider a one-dimensional system, $p = 1$, with a linear (convex/concave) reward function $r(\mathbf{s}, \mathbf{a}) = \mathbf{s}$, a constant feedback policy $\pi(\mathbf{s}) = \boldsymbol{\theta}$ that is parameterized by some parameters $\boldsymbol{\theta}$, and a time horizon of one step, $N = 1$. With these simplifying assumptions, the performance estimate $J(f, \pi)$ in (2) reduces to

$$
\begin{aligned}
J(\tilde{f}, \pi) &= \mathbb{E}_{\boldsymbol{\omega}_{0:N-1}}\left[\sum_{n=0}^{N} r(\tilde{\mathbf{s}}_n, \pi(\tilde{\mathbf{s}}_n)) \,\Big|\, \mathbf{s}_0\right], \quad \text{s.t. } \tilde{\mathbf{s}}_{n+1} = \tilde{f}(\tilde{\mathbf{s}}_n, \pi(\tilde{\mathbf{s}}_n)) + \boldsymbol{\omega}_n, \\
&= \mathbb{E}_{\boldsymbol{\omega}_{0:N-1}}\left[\sum_{n=0}^{N} r(\tilde{\mathbf{s}}_n, \pi(\tilde{\mathbf{s}}_n)) \,\Big|\, \mathbf{s}_0\right], \quad \text{s.t. } \tilde{\mathbf{s}}_{n+1} = \tilde{f}(\tilde{\mathbf{s}}_n, \boldsymbol{\theta}) + \boldsymbol{\omega}_n, \qquad (\pi(\mathbf{s}) = \boldsymbol{\theta}) \\
&= \mathbb{E}_{\boldsymbol{\omega}_{0:N-1}}\left[\sum_{n=0}^{N} \tilde{\mathbf{s}}_n \,\Big|\, \mathbf{s}_0\right], \quad \text{s.t. } \tilde{\mathbf{s}}_{n+1} = \tilde{f}(\tilde{\mathbf{s}}_n, \boldsymbol{\theta}) + \boldsymbol{\omega}_n, \qquad (p = 1, r(\mathbf{s}, \mathbf{a}) = \mathbf{s}) \\
&= \mathbb{E}_{\boldsymbol{\omega}_0}\left[\mathbf{s}_0 + \tilde{\mathbf{s}}_1 \,\Big|\, \mathbf{s}_0\right], \quad \text{s.t. } \tilde{\mathbf{s}}_1 = \tilde{f}(\mathbf{s}_0, \boldsymbol{\theta}) + \boldsymbol{\omega}_0, \qquad (N = 1) \\
&= \mathbf{s}_0 + \tilde{f}(\mathbf{s}_0, \boldsymbol{\theta}) + \mathbb{E}_{\boldsymbol{\omega}_0}[\boldsymbol{\omega}_0], \\
&= \mathbf{s}_0 + \tilde{f}(\mathbf{s}_0, \boldsymbol{\theta}), \qquad (\mathbb{E}[\boldsymbol{\omega}] = 0)
\end{aligned}
\tag{9}
$$

so that the overall goal of model-based reinforcement learning in (3) becomes

$$
\boldsymbol{\theta}^* = \operatorname*{argmax}_{\pi_{\boldsymbol{\theta}}} J(f, \pi_{\boldsymbol{\theta}}), \tag{10}
$$

$$
= \operatorname*{argmax}_{\boldsymbol{\theta}} \mathbf{s}_0 + f(\mathbf{s}_0, \boldsymbol{\theta}), \tag{11}
$$

$$
= \operatorname*{argmax}_{\boldsymbol{\theta}} f(\mathbf{s}_0, \boldsymbol{\theta}). \tag{12}
$$

This is the simplest possible scenario and reduces the optimal control problem in (4) to the bandit problem, where want to maximize an unknown function $f$ that depends on parameters $\boldsymbol{\theta}$ together with a fixed context $\mathbf{s}_0$ that does not impact the solution of the problem.

Algorithms that model the unknown function $f$ in (10) with a probabilistic model $p(\tilde{f} \,|\, \mathcal{D}_{1:t})$ based on noisy observations in $\mathcal{D}_t$ are called Bayesian optimization algorithms (Brochu et al., 2010). In this special case of model-based reinforcement learning, the expected performance objective (4) reduces to

$$
\boldsymbol{\theta}_t = \operatorname*{argmax}_{\boldsymbol{\theta}} \mathbb{E}_{\tilde{f} \sim p(\tilde{f} \,|\, \mathcal{D}_{1:t})}\left[J(\tilde{f}, \pi_{\boldsymbol{\theta}})\right], \tag{13}
$$

$$
= \operatorname*{argmax}_{\boldsymbol{\theta}} \mathbb{E}_{\tilde{f} \sim p(\tilde{f} \,|\, \mathcal{D}_{1:t})}\left[\mathbf{s}_0 + \tilde{f}(\mathbf{s}_0, \boldsymbol{\theta})\right], \tag{14}
$$

$$
= \operatorname*{argmax}_{\boldsymbol{\theta}} \mathbb{E}_{\tilde{f} \sim p(\tilde{f} \,|\, \mathcal{D}_{1:t})}\left[\tilde{f}(\mathbf{s}_0, \boldsymbol{\theta})\right], \tag{15}
$$

$$
= \operatorname*{argmax}_{\boldsymbol{\theta}} \boldsymbol{\mu}_{t-1}(\mathbf{s}_0, \boldsymbol{\theta}). \tag{16}
$$

Thus the expected performance objective selects parameters $\boldsymbol{\theta}_t$ that maximize the posterior mean estimate of $f$ according to $p(\tilde{f} \,|\, \mathcal{D}_{1:t})$. This may seem natural, since the linear reward function encourages states that are as large as possible. However, in the Bayesian optimization literature (13) is equivalent to the UCB strategy with $\beta_t = 0$. This is a greedy algorithm that is well-known to get stuck in local optima (Srinivas et al., 2012).

This is illustrated in Fig. 5: We use a Gaussian process model for $f$ and use (13), which means we set $\beta = 0$ in the GP-UCB algorithm. As a result, we obtain optimization behaviors as in Fig. 5a. The

(a) $\beta_t = 0$.　　　　　　　　　　　　　　(b) $\beta_t = 2$.

Figure 5: Comparison of the GP-UCB algorithm with two different constants for $\beta_t$. The expected performance objective in (13) is equivalent setting to $\beta = 0$ in Fig. 5a. The algorithm gets stuck and repeatedly evaluates inputs (orange crosses) at a local optimum of the true objective function (black dashed). This is due to the mean function (blue line) achieving higher values than the prior expected performance of zero. In contrast, an optimistic algorithm with $\beta = 2$ in Fig. 5b determines close-to-optimal parameters after few evaluations.

first evaluation that achieves performance higher than the expected prior performance (in our case, zero), is evaluated repeatedly (orange crosses). However, this can correspond to a local optimum of the true, unknown objective function (black dashed). In contrast, if we use an optimistic algorithm and set $\beta = 2$, GP-UCB evaluates parameters with close-to-optimal performance.

As a consequence of this counter-example, it is clear that we cannot expect the expected performance exploration criterion in (4) to yield regret guarantees for exploration *in the general case*. However, under the additional assumption of linear dynamics, Mania et al. (2019) show that the algorithm is no-regret. More empirically, Deisenroth et al. (2014, Section 6.1) discuss how to choose specific reward functions that tend to encourage high-variance transitions and thus exploration. However, it is unclear how such an approach can be analyzed theoretically and we would prefer to avoid reward-shaping to encourage exploration.

## B　Extended Experiments

### B.1　Experimental Setup

**Models**　We consider ensembles of Probabilistic Neural Networks (PE) as in Chua et al. (2018) and Gaussian Process (GP) Models for the inverted pendulum as in Kamthe and Deisenroth (2018). For GPs, we use the predictive variance estimate as $\Sigma_{t-1}(\mathbf{s}, \mathbf{a})$ For Ensembles, we approximate the output of the ensemble with a Gaussian as suggested by Lakshminarayanan et al. (2017) and use its predictive mean and variance as $\boldsymbol{\mu}_{t-1}(\mathbf{s}, \mathbf{a})$ and $\Sigma_{t-1}(\mathbf{s}, \mathbf{a})$.

**Model Selection (Training)**　For GPs we do not optimize the Hyper-parameters as this is prone to getting stuck to local minima (Bull, 2011). Advanced methods to avoid this problem, such as those proposed by Berkenkamp et al. (2019), are left for future work. For Ensembles, we train each ensemble separately using Adam (Kingma and Ba, 2015). We assign a transition to each ensemble member sampling from a Poisson distribution $\text{Poi}(1)$ (Osband et al., 2016). This is an asymptotic approximation to the Bootstrap.

**Approximate Thompson Sampling**　We do not consider a Thompson sampling variant of Exact GPs due to the computational complexity. For PE, we sample at the beginning of each episode a head and use *only* this head for optimizing the policy as in Lu and Van Roy (2017).

**Trajectory Sampling**　For greedy exploitation, we propagate particles and the next-state distribution is given by the ensemble (or GP) output at the current particle location. This is the PE-DS

algorithm from Chua et al. (2018), which has comparable performance to PE-TS1 and PE-TS∞. We use this algorithm because it has the same predictive uncertainty used by H-UCRL.

**Policy Search and Planning Algorithm**    For experiments, we use a modification of MPO (Abdolmaleki et al., 2018) with Hallucinated Data Augmentation to simulate data and Hallucinated Value Expansion to compute targets as the `PolicySearch` algorithm. As the resulting algorithm is on-policy, we only learn a value function as critic. The planning algorithm is implemented using Dyna-MPC from Algorithm 7. We update the sampling distribution using the Cross-Entropy Method from Botev et al. (2013). We provide an open-source implementation of our method, which is available at `http://github.com/sebascuri/hucrl` that builds upon the RL-LIB library from Curi (2020), based on pytorch (Paszke et al., 2017).

## B.2    Environment Description and Learning Curves

### B.2.1    Swing-Up Inverted Pendulum

The pendulum has $p = 2$ and $q = 1$, with actions bounded in $[-1, 1]$ and each episode lasts 400 time steps.. We transform the angles to a quaternion representation via $[\sin(\theta), \cos(\theta)]$. The pendulum starts at $\theta_0 = \pi$, $\omega_0 = 0$ and the objective is to swing it up to $\theta_0 = 0$, $\omega_0 = 0$. The reward function is $r(\theta, \omega, \mathbf{a}) = r_\theta \cdot r_\omega + \rho r_{\mathbf{a}}$, where $r_\theta = \text{TOLERANCE}(\cos(\theta), \text{bounds} = (0.95, 1.), \text{margin} = 0.1)$, $r_\omega = \text{TOLERANCE}(\omega, \text{bounds} = (-0.5, 0.5), \text{margin} = 0.5)$, and $r_{\mathbf{a}} = \text{TOLERANCE}(\mathbf{a}, \text{bounds} = (-0.1, 0.1), \text{margin} = 0.1) - 1$. The TOLERANCE is defined in Tassa et al. (2018). In Fig. 6 we show the learning curve of the PE model for five different random seeds. H-UCRL finds quickly a swing-up maneuvere even with high action penalties.

Figure 6: Learning curves of the inverted pendulum. H-UCRL outperforms other algorithms during learning.

### B.2.2    Mujoco Cart Pole

We repeat the experiment in a easy environment, the Mujoco Cart Pole. The cart-pole has $p = 4$ and $q = 1$, with actions bounded in $[-3, 3]$ and each episode lasts 200 time steps. We transform the angles to a quaternion representation via $[\sin(\theta), \cos(\theta)]$. The cart-pole starts from $(0, 0, 0, 0) + \omega$, where $\omega$ is a zero-mean normal noise with $0.1$ standard deviation. The goal is to upswing and stabilize the end-effector at position $x = 0$. The reward is given by $r = e^{-\sum_{i=x,y} \text{ee}_i^2/0.6^2} - \rho \mathbf{a}^2$, where ee is vector of coordinates of the end-effector. Here we see again that, as the action penalty increases, expected and Thompson sampling do not find a swing-up maneuver. We plot the final results together with the learning curves in Fig. 7.

Figure 7: Top: Final episodic return in Cart-Pole environment. Bottom: Learning curves in Cart-Pole environment. For action penalty = 0.05, H-UCRL outperforms other algorithms. For action penalty=0.2 already after the fifth episode it finds a swing-up maneuver. Thompson sampling finds it in only one run after the thirtyfifth episode.

### B.2.3 Reacher

The Reacher is a 7DOF robot with $p = 14$ and $q = 7$, with actions bounded in $[-20, 20]^q$ and each episode lasts 150 time steps. The goal is sampled at location $(x, y, z) = (0, 0.25, 0) + \omega$, where $\omega$ is a zero-mean normal noise with 0.1 standard deviation. We transform the angles to a quaternion representation via $[\sin(\theta), \cos(\theta)]$. The goal is to move the end-effector towards the goal and the reward signal is given by $r = -\sum_{i=x,y,z}(\text{ee} - \text{goal})_i^2 - \rho \sum_{i=1}^{7} \mathbf{a}_i^2$, where $\text{ee} - \text{goal}$ is the vector that measures the distance between the end-effector and the goal. We show the results in Fig. 8. All algorithms perform equally for different action penalties.

Figure 8: Top: Final episodic return in Reacher environment. Bottom: Learning curves in Reacher environment. Greedy, Thompson sampling, and H-UCRL perform equally well.

### B.2.4 Pusher

The Pusher is also a 7DOF robot with $p = 14$ and $q = 7$, with action bounds in $[-2, 2]^q$ and each episode lasts 150 time steps. The object is free to move, introducing 3 more states to the environment. The robot starts with zero angles, an angular velocity sampled uniformly at random from $[-0.005, 0.005]$, the object is sampled from $(x, y) = (-0.25, 0.15) + \omega$, where $\omega$ is a zero-mean normal noise with 0.025 standard deviation. The objective is to push the object towards the goal at $(x, y) = (0, 0)$. The reward signal is given by $r = -0.5 \sum_{i=x,y,z} (\text{ee} - \text{obj})_i^2 - 1.25 \sum_{i=x,y,z} (\text{obj} - \text{goal})_i^2 - \rho \sum_{i=1}^{7} \mathbf{a}_i^2$, where $\text{ee} - \text{obj}$ is the distance between the end-effector and the object and $\text{obj} - \text{goal}$ is the distance between the object and the goal. We show the results in Fig. 9. All algorithms perform equally for different action penalties.

Figure 9: Top: Final episodic return in Pusher environment. Bottom: Learning curves in Pusher environment. Greedy, Thompson sampling, and H-UCRL perform equally well.

### B.2.5 Sparse Reacher

The sparse Reacher is the same 7DOF robot as the Reacher with $p = 14$ and $q = 7$, with actions bounded in $[-20, 20]^q$ and each episode lasts 150 time steps. The sole difference arises in the reward function, which is given by $r = e^{-\sum_{i=x,y,z}(\text{ee} - \text{goal})_i^2 / 0.45^2} + \rho (e^{-\sum_{i=1}^{7} \mathbf{a}_i^2} - 1)$. We show the results in Fig. 10. H-UCRL performs better than Greedy and Thompson, particularly for larger action penalties.

Figure 10: Top: Final episodic return in sparse Reacher environment. Bottom: Learning curves in sparse Reacher environment. H-UCRL outperforms greedy and Thompson sampling, particularly when the action penalty increases.

### B.2.6 Half-Cheetah

The Half-Cheetah is a mobile robot with $p = 17$ and $q = 6$, with actions bounded in $[-2, 2]^q$ and each episode lasts 1000 time steps. The objective is to make the cheetah run as fast as possible forwards up to a maximum of 10m/s. The reward function is given by $r = \max(v, 10)$. We show the results in Fig. 11. H-UCRL performs finds quicker policies with higher returns and, when the action penalty is 1, it outperforms greedy and Thompson sampling considerably.

Figure 11: Top: Final episodic return in Half-Cheetah environment. Bottom: Learning curves in Half-Cheetah environment. H-UCRL outperforms greedy and Thompson sampling, particularly when the actoin penalty increases.

## B.3  Visualization of Real and Simulated Trajectories for Inverted Pendulum

In this section, we visualize the optimistic trajectory for the inverted pendulum problem. We plot the real and simulated trajectories using H-UCRL in Figs. 12–14 with increasing action penalties.

### B.3.1  H-UCRL Trajectories

Already in the first episode, the H-UCRL finds an optimistic trajectory to reach the goal (0, 0) position. With more episodes, it learns the dynamics and simulated and real trajectories match. As the action penalty increases, the action magnitude decreases and it takes longer for the algorithm to find a swing-up trajectory.

Figure 12: Real and simulated trajectories for first 6 episodes with H-UCRL (0 action penalty). We plot the trajectory in phase space, and use color coding to denote the action magnitude.

Figure 13: Real and simulated trajectories for first 6 episodes with H-UCRL (0.1 action penalty). We plot the trajectory in phase space, and use color coding to denote the action magnitude.

Figure 14: Real and simulated trajectories for first 6 episodes with H-UCRL (0.2 action penalty). We plot the trajectory in phase space, and use color coding to denote the action magnitude.

## B.4 Further Experiments on Thompson Sampling

We found surprising that Thompson Sampling under-performs compared to optimistic exploration. To understand better why this happens, we perform different experiments in this section.

### B.4.1 Can the sampled models solve the task?

One possibility is that, when doing posterior sampling, the agent learns a model for the sampled model, which might be biased. If this was the case, we would expect to see the *simulated* returns, i.e., the returns of the optimal policy in the sampled system $\tilde{f}_i$ large.

In Fig. 15 we show the returns of the last simulated trajectory starting from the bottom position of each episode. This figure indicates that there is no model bias, i.e., the simulated returns for Thompson sampling are also low. We conclude that it is not over-fitting to the sampled model, but rather the algorithm cannot solve the task with the sampled model.

Figure 15: Total return from last simulated trajectory with the same initial state as the environment initial state. H-UCRL has higher simulated returns than Greedy and Thompson as the action penalty increases.

### B.4.2 Is it variance starvation?

Another possibility is Thompson Sampling suffers variance starvation, i.e., all ensemble members' predictions are identical. Variance starvation means that the approximate posterior variance is smaller than the true posterior variance. When this happens, (approximate) Thompson Sampling fails because of lack of exploration (Wang et al., 2018). In contrast to UCRL-stye algorithms where the optimism is implemented *deterministically*, Thompson sampling implements optimism *stochastically*. Thus, it is crucial that the variance is not underestimated.

If there was variance starvation, we would expect to see the epistemic variance along simulated trajectories shrink. In Fig. 16 we show the average simulated uncertainty during training, considered as the predictive variance of the ensemble. To summarize the predictive uncertainty into a scalar, we consider the trace of the Cholesky factorization of the covariance matrix. From the figure, we see that H-UCRL starts with the same predictive uncertainty as greedy and Thompson sampling. Furthermore, the variance of Thompson sampling does not shrink. We conclude that there is no variance starvation in the one-step ahead predictions.

Figure 16: Epistemic model uncertainty along simulated trajectories. Thompson and Greedy have the same or more uncertainty than H-UCRL.

### B.4.3 Is the number of ensemble members enough?

In order to verify this hypothesis, we ran the same experiments with 5, 10, 20, 50, and 100 ensemble members. All models swing-up the pendulum with 0 action penalty. With 0.1 action penalty, the 20, 50, and 100 ensembles find a swing up in only one run out of five. With 0.2 action penalty, no model finds a swing-up strategy. This suggests that having larger ensembles could help, but it is not convincing. Furthermore, the model training computational complexity increases linearly with the number of ensemble members, which limits the practicality of larger ensembles.

Figure 17: Episodic returns using Thompson Sampling for different number of ensemble members

### B.4.4 Is it the bootstrapping procedure during Training?

Yet another possibility is that the bootstrap procedure yields inconsistent models for Thompson sampling. To simulate bootstrapping, for each transition and ensemble member, we sample a mask from a Poisson distribution (Osband et al., 2016). Then, we train using the loss of each transition multiplied by this mask. This yields correct one-step ahead confidence intervals. However, the model is used for multi-step ahead predictions. To test if this is the reason of the failure we repeat the experiment *without* bootstrapping the transitions. The only source of discrepancy between the models comes from the initialization of the model. This is how Chua et al. (2018) train their probabilistic models and the models learn from *consistent* trajectories.

In Fig. 18 we show the results when training without bootstrapping. The learning curves closely follow those with bootstrapping in Fig. 6. We conclude that the bootstrapping procedure is likely not the cause of the failure of Thompson Sampling.

Figure 18: Episodic Returns in inverted pendulum *without* bootstrapping data while learning the model.

### B.4.5 Are probabilistic ensembles not a good approximation to the posterior in Thompson sampling?

We next investigate the possibility that Probabilistic Ensembles are not a good approximation for $p(\tilde{f} \,|\, \mathcal{D}_{1:t})$. To this end, we consider the Random Fourier Features (RFF) proposed by Rahimi and Recht (2008) for GP Models. To sample a posterior, we sample a set of random features and use the same features throughout the episodes as required by theoretical results for Thompson sampling and suggested by Hewing et al. (2019) to simulate trajectories. RFFs, however, are known to suffer from variance starvation. We also consider Quadrature Fourier Features (QFF) proposed by Mutny and Krause (2018). QFFs have provable no-regret guarantees in the Bandit setting as well as a uniform approximation bound.

In Fig. 19, we show the results for both RFF (1296 features), and QFFs (625 features). Neither QFFs nor RFFs find a swing-up maneuver for action penalties larger than zero, whereas optimistic exploration with both QFFs and RFFs do. For 0 action penalty, optimistic exploration with RFFs underperforms compared to greedy exploitation and Thompson sampling. This might be due to variance starvation of RFFs because we do not see the same effect on QFFs. We conclude that PE are as good as other approximate posterior methods such as random feature models.

Figure 19: Episodic Returns in inverted pendulum using Random Fourier Features (RFF) and Quadrature Fourier Features (QFF).

### B.4.6 Is it the optimization procedure?

The final and perhaps most enlightening experiment is the following. We run optimistic exploration with five ensemble heads and save snapshots of the models after the first, fifth and tenth episode. Then, we optimize a different policy for each of the models separately. In Fig. 20 we compare the simulated returns using optimistic exploration on the ensemble at each episode against the *maximum* return obtained by the best head.

After the first episode, the simulated returns using optimistic exploration always find an optimistic swing-up trajectory, whereas the best-head always returns zero. This indicates that, when the uncertainty is large, optimistic exploration finds a better policy than approximate Thompson sampling. Without action penalty, the best head return quickly catches up to the simulated ones with optimistic exploration. For an action penalty of 0.1, after five episodes the best head is not able to find a swing-up trajectory. However, after ten episodes it does. This shows that the optimization algorithm is able to find the policy that swings-up a single model. However, when Thompson sampling is used to collect data, the optimization does not find such a policy. This indicates that the models learned using H-UCRL better reduce the uncertainty around the high-reward region and each member of the ensemble has *sharper* predictions. For 0.2 action penalty, the best head never finds a swing-up policy in ten episodes.

Figure 20: Simulated Returns using H-UCRL vs. Maximum simulated return over all ensemble members using the same model as H-UCRL.

### B.4.7 Conclusions

We believe that the poor performance of Thompson sampling relative to H-UCRL suggests that a probabilistic ensemble with five members is sufficient to construct reasonable confidence intervals (hence H-UCRL finds good policies), but does not comprise a rich enough posterior distribution

for Thompson Sampling. We suspect that this effect is inherent to the multi-step RL setting. It seems to be the case that an approximate posterior model whose variance is rich enough for one-step predictions does not sufficiently represent/cover the diversity of plausible trajectories in the multi-step setting. Thompson sampling implements optimism *stochastically*: for it to work, we must be able to sample a model that solves the task using multi-step predictions. Designing *tractable* approximate posteriors with *sufficient* variance for multi-step prediction is still a challenging problem. For instance, an ensemble model with $B$ members that has sufficient variance for 1-step predictions, requires $B^N$ members for N-step predictions, this quickly becomes intractable.

Compared to Thompson sampling, UCRL algorithms in general, and H-UCRL in particular, only require one-step ahead calibrated predictive uncertainties in order to successfully implement optimism. This is because the optimism is implemented *deterministically* and it can be used recursively in a computationally efficient way. Furthermore, we know how to train (and calibrate) models to capture the uncertainty. This hints that optimism might be better suited than approximate Thompson sampling in model-based reinforcement learning.

## C Solving the Augmented Greedy Exploitation Program

In this section, we discuss how to practically solve the greedy exploitation problem with the augmented hallucination variables. In Section 3.1 we showed that the optimization program is a stochastic optimal-control problem for the hallucinated model $\tilde{f}$. There are two common ways to solve this stochastic optimal-control problem: off-line policy search and on-line planning. In Appendix C.1, we describe offline policy search algorithms, in Appendix C.2 we present online planning algorithms, and in Appendix C.3 we show how to combine these algorithms.

### C.1 Offline Policy Search

Off-line policy search usually parameterize a policy $\pi(\cdot; \theta)$ using a function approximation method (e.g., neural networks), and then uses the policy $\pi(\cdot; \theta)$ to interact with the environment. We parameterize both the true and hallucinated policies with neural network $\pi(\cdot; \theta), \eta(\cdot; \theta)$. Next, we describe how to augment common policy-search algorithms with hallucinated policies. Any of such algorithms can be used as the `PolicySearch` method in Algorithm 2.

**Imagined Data Augmentation** consists of using the model to simulate data and then use these data to learn a policy using a model-free RL method. For example, the celebrated Dyna algorithm from Sutton (1990), DAD from Venkatraman et al. (2016), IB from Kalweit and Boedecker (2017), and I2A Racanière et al. (2017) generate data by sampling from expected models. In Algorithm 3, we show HDA (for Hallucinated Data Augmentation). In HDA, we generate data using the optimistic dynamics in (4) and then call any model-free RL algorithm such as SAC (Haarnoja et al., 2018), MPO (Abdolmaleki et al., 2018), TD3 (Fujimoto et al., 2018), TRPO (Schulman et al., 2015), or PPO (Schulman et al., 2017). Furthermore, the initial state distribution where hallucinated trajectories start from might be any exploratory distribution. This greatly simplifies the task of the `ModelFree` algorithm. Usually these strategies combine true with hallucinated data buffers. To match dimensions between these, we augment the action space of the true data buffer with samples of a standard normal. This strategy usually suffers from model-bias as model errors compound throughout a trajectory, yielding highly biased estimates that hinder the policy optimization (van Hasselt et al., 2019).

---

**Algorithm 3** Hallucinated Data Augmentation

---

**Inputs:** Calibrated dynamical model $(\boldsymbol{\mu}, \boldsymbol{\Sigma})$, reward function $r(\mathbf{s}, \mathbf{a})$, horizon $N$, initial state distribution $d(\mathbf{s}_0)$, number of iterations $N_{\text{iter}}$, number of data points $N_{\text{data}}$, initial parameters $\theta_{t-1}, \vartheta_{t-1}$, model-free algorithm `ModelFree`.

1: Initialize $\theta_{t,0} \leftarrow \theta_{t-1}, \vartheta_{t,0} \leftarrow \vartheta_{t-1}$
2: **for** $i = 1, \ldots, N_{\text{iter}}$ **do**
3:     /* Simulate Data */
4:     Initialize hallucinated data buffer $\mathcal{D}_{\text{h}} = \{\emptyset\}$.
5:     **for** $i = 1, \ldots, N_{\text{data}}$ **do**
6:         Start from initial state distribution $\hat{\mathbf{s}}_0 \sim d(\mathbf{s}_0)$.
7:         **for** $n = 0, \ldots, N-1$ **do**
8:             Compute action $\hat{\mathbf{a}}_n \sim \pi(\hat{\mathbf{s}}_n; \theta_{t,i}), \hat{\mathbf{a}}'_n \sim \eta(\hat{\mathbf{s}}_n; \theta_{t,i})$
9:             Sample next state $\hat{\mathbf{s}}_{n+1} \sim \boldsymbol{\mu}_t(\hat{\mathbf{s}}_n, \hat{\mathbf{a}}_n) + \beta_t \boldsymbol{\Sigma}_t(\hat{\mathbf{s}}_n, \hat{\mathbf{a}}_n) \hat{\mathbf{a}}'_n + \boldsymbol{\omega}_n$.
10:             Append transition to buffer $\mathcal{D}_{\text{h}} \leftarrow \mathcal{D}_{\text{h}} \cup \{(\hat{\mathbf{s}}_n, \hat{\mathbf{s}}_{n+1}, \hat{\mathbf{a}}_n, \hat{\mathbf{a}}'_n, r(\hat{\mathbf{s}}_n, \hat{\mathbf{a}}_n))\}$.
11:     /* Optimize Policy */
12:     $\theta_{t,i+1}, \vartheta_{t,i+1} \leftarrow \texttt{ModelFree}(\mathcal{D}_{\text{h}}, \theta_{t,i}, \vartheta_{t,i})$
**Outputs:** Final policy and critic $\theta_t = \theta_{t,N_{\text{iter}}}, \vartheta_t = \vartheta_{t,N_{\text{iter}}}$

---

**Back-Propagation Through Time** is an algorithm that updates the policy parameters by computing the derivatives of the performance w.r.t. the parameters directly. For instance, PILCO from Deisenroth and Rasmussen (2011) and MBAC from Clavera et al. (2020) are different examples of practical algorithms that use a greedy policy (4) using GPs and ensembles of neural networks, respectively. In Algorithm 4, we show how to adapt BPTT to hallucinated control. Like in BPTT it samples the trajectories in a differentiable way, i.e., using the reparameterization trick (Kingma and Welling, 2013). Under some assumptions (such as moment matching), the sampling step in Line 8 of Algorithm 4 can be replaced by exact integration as in PILCO (Deisenroth and Rasmussen, 2011). While performing the rollout, it computes the performance and at the end it bootstrapped with a critic. This critic is

learned using a policy evaluation `PolEval` algorithm such as Fitted Value Iteration (Antos et al., 2008). This strategy usually suffers from high variance due to the stochasticity of the sampled trajectories and the compounding of gradients (McHutchon, 2014). Interestingly, Parmas et al. (2018) propose a method to combine the model-free gradients given by any HDA strategy together with the model-based gradients given by HBPTT, but we leave this for future work. We found that limiting the KL-divergence between the policies in different episodes as suggested by Schulman et al. (2015) helps to control this variance by regularization.

---

**Algorithm 4** Hallucinated Back-Propagation Through Time

---

**Inputs:** Calibrated dynamical model $(\boldsymbol{\mu}, \boldsymbol{\Sigma})$, reward function $r(\mathbf{s}, \mathbf{a})$, horizon $N$, initial state distribution $d(\mathbf{s}_0)$, number of iterations $N_{\text{iter}}$, initial parameters $\theta_{t-1}, \vartheta_{t-1}$, learning rate $eta$, policy evaluation algorithm `PolEval`, regularization $\lambda$.

1:  Initialize $\theta_{t,0} \leftarrow \theta_{t-1}, \vartheta_{t,0} \leftarrow \vartheta_{t-1}$
2:  **for** $i = 1, \ldots, N_{\text{iter}}$ **do**
3:      /* Simulate Data */
4:      Start from initial state distribution $\hat{\mathbf{s}}_0 \sim d(\mathbf{s}_0)$.
5:      Restart $J \leftarrow 0$
6:      **for** $n = 0, \ldots, N - 1$ **do**
7:          Compute action $\hat{\mathbf{a}}_n \sim \pi(\hat{\mathbf{s}}_n; \theta_{t,i}), \hat{\mathbf{a}}'_n \sim \eta(\hat{\mathbf{s}}_n; \theta_{t,i})$
8:          Sample next state $\hat{\mathbf{s}}_{n+1} \sim \boldsymbol{\mu}_t(\hat{\mathbf{s}}_n, \hat{\mathbf{a}}_n) + \beta_t \boldsymbol{\Sigma}_t(\hat{\mathbf{s}}_n, \hat{\mathbf{a}}_n)\hat{\mathbf{a}}'_n + \boldsymbol{\omega}_n$ .
9:          Accumulate $J \leftarrow J + \gamma^n r(\hat{\mathbf{s}}_n, \hat{\mathbf{a}}_n) - \lambda \text{KL}(\pi(\hat{\mathbf{s}}_n; \theta_{t,i}) || \pi(\hat{\mathbf{s}}_n; \theta_{t-1}))$.
10:     Bootstrap $J \leftarrow J + \gamma^N Q(\hat{\mathbf{s}}_N, \pi(\hat{\mathbf{s}}_N; \theta_{t,i}), \eta(\hat{\mathbf{s}}_N; \theta_{t,i}); \vartheta_{t,i})$
11:     /* Optimize Policy */
12:     Compute gradient $\partial J / \partial \theta_t$ with back-propagation through time.
13:     Do gradient step $\theta_{t,i+1} \leftarrow \theta_{t,i} + \eta \partial J / \partial \theta_t$
14:     Update Critic $\vartheta_{t,i+1} \leftarrow \texttt{PolEval}(\theta_{t,i+1})$

**Outputs:** Final policy and critic $\theta_t = \theta_{t,N_{\text{iter}}}, \vartheta_t = \vartheta_{t,N_{\text{iter}}}$

---

**Model-Based Value Expansion** is an Actor-Critic approach that uses the model to compute the next-states for the Bellman target when learning the action-value function. It then uses pathwise derivatives (Mohamed et al., 2019) through the learned action-value function. For example MVE from (Feinberg et al., 2018) and STEVE from Buckman et al. (2018) use such strategy. In Algorithm 5, we show H-MVE (Hallucinated-Model Based Value Expansion). Here we use optimistic trajectories only to learn the Bellman target. In turn, the learned action-values functions are optimistic and so are the pathwise gradients computed through them. This strategy is usually less data efficient than BPTT or IDA as it uses the model only to compute targets, but suffers less from model bias. To address data efficiency, one can combine HVE and HDA to compute optimistic value functions as well as simulating optimistic data.

### C.2   Online Planning

An alternative approach is to consider non-parametric policies and directly optimize the true and hallucinated actions as $\mathbf{a}_{n,t} \in [-1, 1]^q, \mathbf{a}'_{n,t} \in [-1, 1]^p$. This is usually called Model-Predictive Control (MPC) and it is implemented in a receding horizon fashion (Morari and H. Lee, 1999). That means that for each new state encounter online the HUCRL planning problem (7) is solved using the actions as decission variables. This addresses model errors compounding as the trajectories are evaluated through the real trajectories, but it comes at high online computational costs, which limit the applicability of such algorithms to simulations.

GP-MPC Kamthe and Deisenroth (2018) and PETS Chua et al. (2018) are MPC-based methods that use the greedy policy (4) using GP and neural networks ensembles, respectively. Other MPC solvers such as POPLIN Wang and Ba (2019) or POLO (Lowrey et al., 2019) are also compatible with such dynamical models. In H-MPC (Hallucinated-MPC), we directly optimize both the control and hallucinated inputs jointly and any of the previous methods can be used as the MPC solver. Moldovan et al. (2015) also use MPC to solve an optimistic exploration scheme but only on linear models and, like other on-line planning methods, are extremely slow for real-time deployment.

---

**Algorithm 5** Hallucinated Value Expansion

---

**Inputs:** Calibrated dynamical model $(\boldsymbol{\mu}, \boldsymbol{\Sigma})$, reward function $r(\mathbf{s}, \mathbf{a})$, number of steps $N$, number of iterations $N_{\text{iter}}$, initial parameters $\theta_{t-1}, \vartheta_{t-1}$, true data buffer $\mathcal{D}_{\text{r}}$, learning rate $\eta$, polyak parameter $\tau$.

1:  Initialize $\theta_{t,0} \leftarrow \theta_{t-1}, \vartheta_{t,0} \leftarrow \vartheta_{t-1}, \bar{\vartheta}_{t,0} \leftarrow \vartheta_{t-1}$
2:  **for** $i = 1, \ldots, N_{\text{iter}}$ **do**
3:      /* Simulate Data */
4:      Start from buffer $\hat{\mathbf{s}}_0 \sim \mathcal{D}_{\text{r}}$.
5:      Initialize target $Q_{\text{target}} \leftarrow 0$.
6:      Compute prediction $Q_{\text{pred}} = Q(\hat{\mathbf{s}}_0; \vartheta_{t,i})$.
7:      **for** $n = 0, \ldots, N-1$ **do**
8:          Compute action $\hat{\mathbf{a}}_n \sim \pi(\hat{\mathbf{s}}_n; \theta_{t,i}), \hat{\mathbf{a}}'_n \sim \eta(\hat{\mathbf{s}}_n; \theta_{t,i})$
9:          Sample next state $\hat{\mathbf{s}}_{n+1} \sim \boldsymbol{\mu}_t(\hat{\mathbf{s}}_n, \hat{\mathbf{a}}_n) + \beta_t \boldsymbol{\Sigma}_t(\hat{\mathbf{s}}_n, \hat{\mathbf{a}}_n)\hat{\mathbf{a}}'_n + \boldsymbol{\omega}_n$ .
10:         Accumulate target $Q_{\text{target}} \leftarrow \gamma^n r(\hat{\mathbf{s}}_n, \hat{\mathbf{a}}_n)$.
11:     Bootstrap $Q_{\text{target}} \leftarrow Q_{\text{target}} + \gamma^N Q(\hat{\mathbf{s}}_N, \pi(\hat{\mathbf{s}}_N; \theta_{t,i}), \eta(\hat{\mathbf{s}}_N; \theta_{t,i}); \bar{\vartheta}_{t,i})$
12:     /* Optimize Critic */
13:     $\vartheta_{t,i+1} \leftarrow \vartheta_{t,i} - \eta \nabla_\vartheta (Q_{\text{pred}} - Q_{\text{target}})^2$
14:     Update target parameters $\bar{\vartheta}_{t,i+1} \leftarrow \tau \bar{\vartheta}_{t,i} + (1 - \tau)\vartheta_{t,i+1}$
15:     /* Optimize Policy */
16:     $\theta_{t,i+1} \leftarrow \theta_{t,i} + \eta \nabla_{\theta_{t,i}} Q(\hat{\mathbf{s}}_0; \vartheta_{t,i})$

**Outputs:** Final policy $\theta_t = \theta_{t,\theta_t}$.

---

To solve the optimization problem, approximate local solvers are usually used that rely either on sampling or on linearization. We discuss how to use both of them with hallucinated inputs. These algorithms can be used as the `Plan` method in Algorithm 2.

**Random Sampling Methods**    An approximate way of solving MPC problems is to exhaustively sample the decision variables. Shooting methods sample the actions and then propagate the trajectory through the model whereas collocation methods sample both the states and the actions. For simplicity, we only consider shooting methods. This method initializes particles at the current state. For each particle, it samples a sequence of actions from a proposal distribution and rollouts each particle independently, computing the returns of such sequence. This process is repeated updating the proposal distribution. Random Shooting (Richards and How, 2006), the Cross-Entropy Method (Botev et al., 2013), and Model-Predictive Path Integral Control (Williams et al., 2016) differ in the ways to select the elite actions between iterations and how to update the sampling distributions. All these methods maintain a distribution over the actions. POPLIN from Wang and Ba (2019) instead maintains a distribution over the weights of a policy network and samples different policies. The main advantage of this method is that it correlates the random samples through the dynamics, possibly scaling to higher dimensions. Any of these methods can be used with hallucination. We show in Algorithm 6 the pseudo-code for a meta-Hallucinated shooting algorithm.

**Differential Dynamic Programming (DDP)**    DDP can be interpreted as a second-order shooting method Jacobson (1968) for dynamical systems. For linear dynamical models with quadratic costs, problem (4) is a quadratic program (QP) that enjoys a closed form solution (Morari and H. Lee, 1999). To address non-linear systems and other cost functions, a common strategy is to use a variant of iLQR Li and Todorov (2004); Todorov and Li (2005); Tassa et al. (2012) which linearizes the system and uses a second order approximation to the cost function to solve sequential QPs (SQP) that approximate the original problem. When the rewards and dynamical model are differentiable, this method is faster to sampling methods as it uses the problem structure to update the sampling distribution.

## C.3  Combining Offline Policy Search with Online Planning

MPC methods suffer less from model bias, but typically require substantial computation. Furthermore, they are limited to the planning horizon unless a *learned* terminal reward is used to approximate the reward-to-go (Lowrey et al., 2019). On the other hand, off-policy search approaches yield policies and value function estimates (critics) that are fast to evaluate, but suffer from bias (van Hasselt et al.,

---
**Algorithm 6** Hallucinated Shooting Method
---
**Inputs:** Calibrated dynamical model $(\boldsymbol{\mu}, \boldsymbol{\Sigma})$, terminal reward $V$, reward function $r(\mathbf{s}, \mathbf{a})$, horizon $N$, current state $\mathbf{s}_n$, number of particles $n_{\text{particle}}$, number of iterations $n_{\text{iter}}$, number of elite particles $n_{\text{elite}}$. initial sampling distribution $d(\cdot)$, algorithm to evaluate actions `EliteActions`, algorithm to update distribution `UpdateDistribution`.

1: **for** $i = 1, \ldots, n_{\text{iter}}$ **do**
2:    /* Simulate Data */
3:    Initialize $n_{\text{particle}}$ at the current state $\hat{\mathbf{s}}_0^{(i)} = \mathbf{s}_n$
4:    Initialize $J^{(i)} \leftarrow 0$
5:    **for** $n = 0, \ldots, N - 1$ **do**
6:       Sample action $\hat{\mathbf{a}}_n^{(i)}, \hat{\mathbf{a}}_n'^{(i)} \sim d(\cdot)$
7:       Sample next state $\hat{\mathbf{s}}_{n+1}^{(i)} \sim \boldsymbol{\mu}_n(\hat{\mathbf{s}}_n^{(i)}, \hat{\mathbf{a}}_n^{(i)}) + \beta_t \boldsymbol{\Sigma}_n(\hat{\mathbf{s}}_n^{(i)}, \hat{\mathbf{a}}_n^{(i)}) \hat{\mathbf{a}}_n'^{(i)} + \boldsymbol{\omega}_n$.
8:       Accumulate $J^{(i)} \leftarrow J^{(i)} + \gamma^n r(\hat{\mathbf{s}}_n^{(i)}, \hat{\mathbf{a}}_n^{(i)})$
9:    Bootstrap $J^{(i)} \leftarrow J^{(i)} + \gamma^N V(\hat{\mathbf{s}}_N^{(i)})$.
10:   $a, a' \leftarrow$ `EliteActions`$(J^{(i)}, \hat{\mathbf{a}}_{0:N-1}^{(i)}, \hat{\mathbf{a}}_{0:N-1}'^{(i)}, n_{\text{elite}})$
11:   /* Optimize Policy */
12:   Update proposal distribution $d(\cdot) \leftarrow$ `UpdateDistribution`$(a, a')$.
**Outputs:** Return best action $a, a' \leftarrow$ `EliteActions`$(J^{(i)}, \hat{\mathbf{a}}_{0:N-1}^{(i)}, \hat{\mathbf{a}}_{0:N-1}'^{(i)}, 1)$.
---

---
**Algorithm 7** Dyna-MPC with Hallucinated Models
---
**Inputs:** Calibrated dynamical model $(\boldsymbol{\mu}, \boldsymbol{\Sigma})$, learned policies $\pi(\cdot; \theta), \eta(\cdot; \theta)$ learned critic $Q(\cdot; \vartheta)$, reward function $r(\mathbf{s}, \mathbf{a})$, horizon $N$, current state $\mathbf{s}_n$, number of particles $n_{\text{particle}}$, number of iterations $n_{\text{iter}}$, number of elite particles $n_{\text{elite}}$. initial sampling distribution $d(\cdot)$, algorithm to evaluate actions `EliteActions`, algorithm to update distribution `UpdateDistribution`.

1: **for** $i = 1, \ldots, n_{\text{iter}}$ **do**
2:    /* Simulate Data */
3:    Initialize $n_{\text{particle}}$ at the current state $\hat{\mathbf{s}}_0^{(i)} = \mathbf{s}_n$
4:    Initialize $J^{(i)} \leftarrow 0$
5:    **for** $n = 0, \ldots, N - 1$ **do**
6:       Sample action $\hat{\mathbf{a}}_n^{(i)}, \hat{\mathbf{a}}_n'^{(i)} \sim (\pi(\hat{\mathbf{s}}_n^{(i)}; \theta), \eta(\hat{\mathbf{s}}_n^{(i)}; \theta)) + d(\cdot)$
7:       Sample next state $\hat{\mathbf{s}}_{n+1}^{(i)} \sim \boldsymbol{\mu}_n(\hat{\mathbf{s}}_n^{(i)}, \hat{\mathbf{a}}_n^{(i)}) + \beta_t \boldsymbol{\Sigma}_n(\hat{\mathbf{s}}_n^{(i)}, \hat{\mathbf{a}}_n^{(i)}) \hat{\mathbf{a}}_n'^{(i)} + \boldsymbol{\omega}_n$.
8:       Accumulate $J^{(i)} \leftarrow J^{(i)} + \gamma^n r(\hat{\mathbf{s}}_n^{(i)}, \hat{\mathbf{a}}_n^{(i)})$
9:    Bootstrap $J^{(i)} \leftarrow J^{(i)} + \gamma^N Q(\hat{\mathbf{s}}_N^{(i)}, \hat{\mathbf{a}}_N^{(i)}, \hat{\mathbf{a}}_N'^{(i)}; \vartheta)$.
10:   $a, a' \leftarrow$ `EliteActions`$(J^{(i)}, \hat{\mathbf{a}}_{0:N-1}^{(i)}, \hat{\mathbf{a}}_{0:N-1}'^{(i)}, n_{\text{elite}})$
11:   /* Optimize Policy */
12:   Update proposal distribution $d(\cdot) \leftarrow$ `UpdateDistribution`$(a, a')$.
**Outputs:** Return best action $a, a' \leftarrow$ `EliteActions`$(J^{(i)}, \hat{\mathbf{a}}_{0:N-1}^{(i)}, \hat{\mathbf{a}}_{0:N-1}'^{(i)}, 1)$.
---

2019). We propose to combine these methods to get the best of both worlds: First, we learn parametric policies $\pi$ and $\eta$ using a policy search algorithm. Then, we use such policies as a warm-start for the sampling distributions of the planning algorithm. We name this planning algorithm Dyna-MPC, as it resembles the Dyna architecture proposed by Sutton (1990) and we show the pseudo-code for hallucinated models in Algorithm 7.

Closely related to Dyna-MPC is POPLIN (Wang and Ba, 2019). We also use a policy to initialize actions and and then refine them with a shooting method. Nevertheless, we use a policy search algorithm to optimize the policy parameters instead of the cross-entropy method. Hong et al. (2019) also uses MPC to refine an off-line learned policy. However, they use a model-free algorithm directly form real data instead of model-based policy search.

# D    Proofs for Exploration Regret Bound

In this section, we prove the main theorem.

## D.1    Notation

In the following, we implicitly denote with $\mathbf{s}_{n,t}$ the states visited under the true dynamics $f$ in (1) and with $\tilde{\mathbf{s}}_n$ the states visited under $\pi_t$ but the optimistic dynamics $\tilde{f}_t(\mathbf{s}, \mathbf{a}) = \boldsymbol{\mu}_{t-1}(\mathbf{s}, \mathbf{a}) + \boldsymbol{\Sigma}_{t-1}(\mathbf{s}, \mathbf{a})\eta_t(\mathbf{s}, \mathbf{a})$,

$$\mathbf{s}_{n+1,t} = f(\mathbf{s}_{n,t}, \mathbf{a}_{n,t}) + \boldsymbol{\omega}_{n,t} \tag{17a}$$

$$\mathbf{a}_{n,t} = \pi_t(\mathbf{s}_{n,t}) \tag{17b}$$

and

$$\tilde{\mathbf{s}}_{n+1,t} = \tilde{f}_t(\mathbf{s}_{n,t}, \tilde{\mathbf{a}}_{n,t}) + \boldsymbol{\omega}_{n,t} \tag{17c}$$

$$= \boldsymbol{\mu}_{t-1}(\mathbf{s}_{n,t}, \tilde{\mathbf{a}}_{n,t}) + \boldsymbol{\Sigma}_{t-1}(\mathbf{s}_{n,t}, \tilde{\mathbf{a}}_{n,t})\eta_t(\mathbf{s}_{n,t}, \tilde{\mathbf{a}}_{n,t}) + \boldsymbol{\omega}_{n,t} \tag{17d}$$

$$\tilde{\mathbf{a}}_{n,t} = \pi_t(\tilde{\mathbf{s}}_{n,t}). \tag{17e}$$

Since the control actions $\mathbf{a}_{n,t} = \pi_t(\mathbf{s}_{n,t})$ and $\tilde{\mathbf{a}}_{n,t} = \pi_t(\tilde{\mathbf{s}}_{n,t})$ are fixed given $\pi_t$, we generally drop the dependence on $u$ and write $f(\mathbf{s}) = f(\mathbf{s}, \pi_t(\mathbf{s}))$, $\boldsymbol{\mu}(\mathbf{s}, \pi_t(\mathbf{s}))$, etc. We also drop the subscript $t$ from $\mathbf{s}_{n,t}$ whenever it is clear that we refer to the $t$th episode. Lastly, when no norm is specified, $\|\cdot\| = \|\cdot\|_2$ refers to the two-norm.

We start by clarifying that as a consequence of Assumptions 1 and 3 the closed-loop dynamics are Lipschitz continuous too.

**Corollary 1.** *As in Assumption 6, let the open-loop dynamics $f$ in (1) be $L_f$-Lipschitz continuous and the policy $\pi \in \Pi$ be $L_\pi$-Lipschitz continuous w.r.t. to the 2-norm. Then the closed-loop system is $L_{\mathrm{fc}}$-Lipschitz continuous with $L_{\mathrm{fc}} = L_f\sqrt{1 + L_\pi}$.*

*Proof.*

$$\|f(\mathbf{s}, \pi(\mathbf{s})) - f(\mathbf{s}', \pi(\mathbf{s}'))\|_2 \leq L_f \|(\mathbf{s} - \mathbf{s}', \pi(\mathbf{s}) - \pi(\mathbf{s}'))\|_2 \tag{18}$$

$$= L_f \sqrt{\|(\mathbf{s} - \mathbf{s}'\|_2^2 + \|\pi(\mathbf{s}) - \pi(\mathbf{s}'))\|_2^2} \tag{19}$$

$$\leq L_f \sqrt{\|(\mathbf{s} - \mathbf{s}'\|_2^2 + L_\pi\|\mathbf{s} - \mathbf{s}'))\|_2^2} \tag{20}$$

$$= \underbrace{L_f \sqrt{1 + L_\pi}}_{:=L_{\mathrm{fc}}}\|\mathbf{s} - \mathbf{s}'\|_2 \tag{21}$$

$\square$

## D.2    Bounding the Regret

We start by bounding the cumulative regret in terms of the predictive variance of the states/actions on the true trajectory (the one that we will later collect data one).

**Lemma 1.** *Under Assumption 2, for any sequence $\mathbf{s}_{n,t}$ generated by the true system (1), there exists a function $\eta\colon \mathbb{R}^p \to [-1, 1]^p$ such that $\mathbf{s}_{n,t} = \tilde{\mathbf{s}}_{n,t}$ if $\boldsymbol{\omega} = \tilde{\boldsymbol{\omega}}$.*

*Proof.* By Assumption 2 we have $|f(\mathbf{s}) - \boldsymbol{\mu}(\mathbf{s})| \leq \beta\boldsymbol{\sigma}(\mathbf{s})$ elementwise. Thus for each $\mathbf{s}, \mathbf{a}$ there exists a vector $\boldsymbol{\eta}$ with values in $[-1, 1]^p$ such that $f(\mathbf{s}, \mathbf{a}) = \mu(\mathbf{s}, \mathbf{a}) + \boldsymbol{\Sigma}(\mathbf{s}, \mathbf{a})\boldsymbol{\eta}$. Let the function $\eta(\cdot)$ return this vector for each state and action, then the result follows. $\square$

**Lemma 2.** *Under Assumption 2, with probability at least $(1 - \delta)$ we have for all $t \geq 0$ that the regret $r_t$ is bounded by*

$$r_t = J(f, \pi^*) - J(f, \pi_t) \leq J(\tilde{f}_t, \pi_t) - J(f, \pi_t) \tag{22}$$

*Proof.* By Assumption 2, we know from Lemma 1 that the true dynamics are contained within the feasible region of (7); that is, there exists an $\eta(\cdot)\colon \mathbb{R}^p \times \mathbb{R}^q \to [-1,1]^p$ such that with $\tilde{f}(\mathbf{s}) = \boldsymbol{\mu}(\mathbf{s}) + \boldsymbol{\Sigma}(\mathbf{s})\eta(\mathbf{s})$ we have $J(f,\pi^*) = \tilde{J}(\tilde{f},\pi^*)$. As a consequence, we have $J(f,\pi^*) \leq J(\tilde{f}_t,\pi_t)$ and the result follows. $\qquad\square$

Thus, to bound the instantaneous regret $r_t$, we must bound the difference between the optimistic value estimate $J(\tilde{f}_t,\pi_t)$ and the true value $J(f,\pi_t)$. We can use the Lipschitz continuity properties to obtain

**Lemma 3.** *Based on Assumption 3 we have*

$$|J(\tilde{f}_t,\pi_t) - J(f,\pi_t)| \leq L_r\sqrt{1+L_\pi}\sum_{n=0}^{N}\mathbb{E}_{\boldsymbol{\omega}=\tilde{\boldsymbol{\omega}}}[\|\mathbf{s}_{n,t} - \tilde{\mathbf{s}}_{n,t}\|_2] \qquad (23)$$

*Proof.*

$$|J(\tilde{f}_t,\pi_t) - J(f,\pi_t)| = \left|\mathbb{E}_{\tilde{\boldsymbol{\omega}}}\left[\sum_{n=0}^{N} r(\tilde{\mathbf{s}}_n,\pi_t(\tilde{\mathbf{s}}_n))\right] - \mathbb{E}_{\boldsymbol{\omega}}\left[\sum_{n=0}^{N} r(\mathbf{s}_n,\pi_t(\mathbf{s}_n))\right]\right| \qquad (24)$$

$$= \left|\mathbb{E}_{\boldsymbol{\omega}=\tilde{\boldsymbol{\omega}}}\left[\sum_{n=0}^{N} r(\tilde{\mathbf{s}}_n,\pi_t(\tilde{\mathbf{s}}_n)) - r(\mathbf{s}_n,\pi_t(\mathbf{s}_n))\right]\right| \qquad (25)$$

$$\leq L_r\sqrt{1+L_\pi}\sum_{n=0}^{N}\mathbb{E}_{\boldsymbol{\omega}=\tilde{\boldsymbol{\omega}}}[\|\tilde{\mathbf{s}}_n - \mathbf{s}_n\|_2], \qquad (26)$$

where $\mathbb{E}_{\boldsymbol{\omega}=\tilde{\boldsymbol{\omega}}}[\cdot]$ means in expectation over $\boldsymbol{\omega}$ and with $\tilde{\boldsymbol{\omega}} = \boldsymbol{\omega}$; that is, $\tilde{\boldsymbol{\omega}}$ and $\boldsymbol{\omega}$ are the same random variable. $\qquad\square$

Figure 21: Illustrative comparison of the true state trajectory $\mathbf{s}_n$ under the policy $\pi_{\boldsymbol{\theta}}$ and the optimistic trajectory $\tilde{\mathbf{s}}_n$ from (7). After one step, $\mathbf{s}_1$ is contained within the confidence intervals (grey bars). The optimistic dynamics are chosen within this confidence interval to maximize performance. Since the optimistic dynamics are constructed iteratively based on the previous state $\tilde{\mathbf{s}}_n$, beyond one step the true dynamics are not contained in the confidence intervals.

What remains is to bound the deviation of the optimistic and the true trajectory. We show a different perspective of Fig. 2 in Fig. 21, where we explicitly show the "real" state trajectory under a policy and for a given noise realisation the the optimistic trajectory with its one-step uncertainty estimates as in (7). We exploit the Lipschitz continuity of $\boldsymbol{\sigma}$ from Assumption 3 in order to bound the deviation in terms of $\boldsymbol{\sigma}_{t-1}$ at states of the "real" trajectory.

**Lemma 4.** *Under Assumptions 1–3, let $\bar{L}_f = 1 + L_{\text{fc}} + 2\beta_{t-1}L_\sigma\sqrt{1+L_\pi}$. Then, for all iterations $t > 0$, any function $\eta\colon \mathbb{R}^p \times \mathbb{R}^q \to [-1,1]^p$ and any sequence of $\boldsymbol{\omega}_n$ with $\tilde{\boldsymbol{\omega}}_n = \boldsymbol{\omega}_n$, $\pi \in \Pi$ with $1 \leq n \leq N$ we have that*

$$\|\mathbf{s}_{n,t} - \tilde{\mathbf{s}}_{n,t}\| \leq 2\beta_{t-1}\bar{L}_f^{N-1}\sum_{i=0}^{n-1}\|\boldsymbol{\sigma}_{t-1}(\mathbf{s}_{i,t})\| \qquad (27)$$

*Proof.* We start by showing that, for any $n \geq 1$ we have

$$\|\mathbf{s}_{n,t} - \tilde{\mathbf{s}}_{n,t}\| \leq 2\beta_{t-1} \sum_{i=0}^{n-1}(L_{\text{fc}} + 2\beta_{t-1}L_\sigma\sqrt{1+L_\pi})^{n-1-i}\|\boldsymbol{\sigma}_{t-1}(\mathbf{s}_{i,t})\| \tag{28}$$

by induction. For the base case we have $\tilde{\mathbf{s}}_0 = \mathbf{s}_0$. Consequently, at iteration $t$ we have

$$\|\mathbf{s}_{1,t} - \tilde{\mathbf{s}}_{1,t}\| = \|f(\mathbf{s}_0) + \boldsymbol{\omega}_0 - \boldsymbol{\mu}_{t-1}(\mathbf{s}_0) - \beta_{t-1}\boldsymbol{\Sigma}_{t-1}(\mathbf{s}_0)\eta(\mathbf{s}_0) - \tilde{\boldsymbol{\omega}}_0\| \tag{29}$$

$$\leq \|f(\mathbf{s}_0) - \boldsymbol{\mu}_{t-1}(\mathbf{s}_0)\| + \beta_{t-1}\|\boldsymbol{\Sigma}_{t-1}(\mathbf{s}_0)\eta(\mathbf{s}_0)\| \tag{30}$$

$$\leq \beta_{t-1}\|\boldsymbol{\sigma}_{t-1}(\mathbf{s}_0)\| + \beta_{t-1}\|\boldsymbol{\sigma}_{t-1}(\mathbf{s}_0)\| \tag{31}$$

$$= 2\beta_{t-1}\|\boldsymbol{\sigma}_{t-1}(\mathbf{s}_0)\| \tag{32}$$

For the induction step assume that (28) holds at time step $n$. Subsequently we have at iteration $t$ that

$$\|\mathbf{s}_{n+1,t} - \tilde{\mathbf{s}}_{n+1,t}\|$$
$$= \|f(\mathbf{s}_n) + \boldsymbol{\omega}_n - \boldsymbol{\mu}_{t-1}(\tilde{\mathbf{s}}_n) - \beta_{t-1}\boldsymbol{\Sigma}_{t-1}(\tilde{\mathbf{s}}_n)\eta(\tilde{\mathbf{s}}_n) - \tilde{\boldsymbol{\omega}}_n\|$$
$$= \|f(\mathbf{s}_n) - \boldsymbol{\mu}_{t-1}(\tilde{\mathbf{s}}_n) - \beta_{t-1}\boldsymbol{\Sigma}_{t-1}(\tilde{\mathbf{s}}_n)\eta(\tilde{\mathbf{s}}_n) + f(\tilde{\mathbf{s}}_n) - f(\tilde{\mathbf{s}}_n)\|$$
$$= \|f(\tilde{\mathbf{s}}_n) - \boldsymbol{\mu}_{t-1}(\tilde{\mathbf{s}}_n) - \beta_{t-1}\boldsymbol{\Sigma}_{t-1}(\tilde{\mathbf{s}}_n)\eta(\tilde{\mathbf{s}}_n) + f(\mathbf{s}_n) - f(\tilde{\mathbf{s}}_n)\|$$
$$= \|f(\tilde{\mathbf{s}}_n) - \boldsymbol{\mu}_{t-1}(\tilde{\mathbf{s}}_n)\| + \|\beta_{t-1}\boldsymbol{\Sigma}_{t-1}(\tilde{\mathbf{s}}_n)\eta(\tilde{\mathbf{s}}_n)\| + \|f(\mathbf{s}_n) - f(\tilde{\mathbf{s}}_n)\|$$
$$\leq \beta_{t-1}\|\boldsymbol{\sigma}_{t-1}(\tilde{\mathbf{s}}_n)\| + \beta_{t-1}\|\boldsymbol{\sigma}_{t-1}(\tilde{\mathbf{s}}_n)\| + L_{\text{fc}}\|\mathbf{s}_n - \tilde{\mathbf{s}}_n\|$$
$$= 2\beta_{t-1}\|\boldsymbol{\sigma}_{t-1}(\tilde{\mathbf{s}}_n)\| + L_{\text{fc}}\|\mathbf{s}_n - \tilde{\mathbf{s}}_n\|$$
$$= 2\beta_{t-1}\|\boldsymbol{\sigma}_{t-1}(\mathbf{s}_n) + \boldsymbol{\sigma}_{t-1}(\tilde{\mathbf{s}}_n) - \boldsymbol{\sigma}_{t-1}(\mathbf{s}_n)\| + L_{\text{fc}}\|\mathbf{s}_n - \tilde{\mathbf{s}}_n\|$$
$$\leq 2\beta_{t-1}\left(\|\boldsymbol{\sigma}_{t-1}(\mathbf{s}_n)\| + L_\sigma\sqrt{1+L_\pi}\|\mathbf{s}_n - \tilde{\mathbf{s}}_n\|\right) + L_{\text{fc}}\|\mathbf{s}_n - \tilde{\mathbf{s}}_n\|$$
$$= 2\beta_{t-1}\|\boldsymbol{\sigma}_{t-1}(\mathbf{s}_n)\| + (L_{\text{fc}} + 2\beta_{t-1}L_\sigma\sqrt{1+L_\pi})\|\mathbf{s}_n - \tilde{\mathbf{s}}_n\|$$
$$\leq 2\beta_{t-1}\|\boldsymbol{\sigma}_{t-1}(\tilde{\mathbf{s}}_n)\| + (L_{\text{fc}} + 2\beta_{t-1}L_\sigma\sqrt{1+L_\pi})2\beta_{t-1}\sum_{i=0}^{n-1}(L_{\text{fc}} + 2\beta_{t-1}L_\sigma\sqrt{1+L_\pi})^{n-1-i}\|\boldsymbol{\sigma}_{t-1}(\mathbf{s}_i)\|$$
$$= 2\beta_{t-1}\sum_{i=0}^{(n+1)-1}(L_{\text{fc}} + 2\beta_{t-1}L_\sigma\sqrt{1+L_\pi})^{(n+1)-1-i}\|\boldsymbol{\sigma}_{t-1}(\mathbf{s}_i)\|$$

Thus (28) holds. Now since $n \leq N$ we have

$$\|\mathbf{s}_{n,t} - \tilde{\mathbf{s}}_{n,t}\| \leq 2\beta_{t-1}\sum_{i=0}^{n-1}(L_{\text{fc}} + 2\beta_{t-1}L_\sigma\sqrt{1+L_\pi})^{n-1-i}\|\boldsymbol{\sigma}_{t-1}(\mathbf{s}_{i,t})\| \tag{33}$$

$$\leq 2\beta_{t-1}\sum_{i=0}^{n-1}(1 + L_{\text{fc}} + 2\beta_{t-1}L_\sigma\sqrt{1+L_\pi})^{n-1-i}\|\boldsymbol{\sigma}_{t-1}(\mathbf{s}_{i,t})\| \tag{34}$$

$$\leq 2\beta_{t-1}\underbrace{(1 + L_{\text{fc}} + 2\beta_{t-1}L_\sigma\sqrt{1+L_\pi})}_{:=\bar{L}_f}{}^{N-1}\sum_{i=0}^{n-1}\|\boldsymbol{\sigma}_{t-1}(\mathbf{s}_{i,t})\| \tag{35}$$

$$\tag{36}$$

$\square$

**Corollary 2.** *Under the assumptions of Lemma 4, for any sequence of $\eta_n \in [-1,1]$, $\boldsymbol{\theta} \in \mathcal{D}$, and $n \geq 1$, $t \geq 1$ we have that*

$$\mathbb{E}_{\boldsymbol{\omega}=\tilde{\boldsymbol{\omega}}}[\|\mathbf{s}_{n,t} - \tilde{\mathbf{s}}_{n,t}\|] \leq 2\beta_{t-1}\bar{L}_f^{N-1}\mathbb{E}_{\boldsymbol{\omega}}\left[\sum_{i=0}^{n-1}\|\boldsymbol{\sigma}_{t-1}(\mathbf{s}_{i,t})\|\right] \tag{37}$$

*Proof.* This is a direct consequence of Lemma 4. $\square$

As a direct consequence of these lemmas, we can bound the regret in terms of the predictive uncertainty of our statistical model in expectation over the states visited under the true dynamics.

**Lemma 5.** *Under Assumptions 2–3, let $L_J = 2L_r\sqrt{1 + L_\pi}\beta_{t-1}\bar{L}_f^{N-1}$. Then, with probability at least $(1 - \delta)$ it holds for all $t \geq 0$ that*

$$r_t^2 \leq L_J^2 N^3 \mathbb{E}_{\boldsymbol{\omega}}\left[\sum_{n=0}^{N-1} \|\boldsymbol{\sigma}_{t-1}(\mathbf{s}_{n,t})\|_2^2\right] \tag{38}$$

*Proof.*

$$r_t \leq J(\tilde{f}_t, \pi_t) - J(f, \pi_t) \tag{39}$$

$$\leq L_r\sqrt{1 + L_\pi}\sum_{n=0}^{N} \mathbb{E}_{\boldsymbol{\omega}=\tilde{\boldsymbol{\omega}}}[\|\mathbf{s}_{n,t} - \tilde{\mathbf{s}}_{n,t}\|_2] \tag{40}$$

$$\leq 2L_r\sqrt{1 + L_\pi}\beta_{t-1}\bar{L}_f^{N-1}\sum_{n=0}^{N} \mathbb{E}_{\boldsymbol{\omega}}\left[\sum_{i=0}^{n-1} \|\boldsymbol{\sigma}_{t-1}(\mathbf{s}_{i,t})\|_2\right] \tag{41}$$

$$\leq 2L_r\sqrt{1 + L_\pi}\beta_{t-1}\bar{L}_f^{N-1} N\mathbb{E}_{\boldsymbol{\omega}}\left[\sum_{n=0}^{N-1} \|\boldsymbol{\sigma}_{t-1}(\mathbf{s}_{n,t})\|_2\right] \tag{42}$$

where the third inequality follows from Corollary 2. Now, let $L_J = 2L_r\sqrt{1 + L_\pi}\beta_{t-1}\bar{L}_f^{N-1}$, so that

$$r_t \leq L_J N\mathbb{E}_{\boldsymbol{\omega}}\left[\sum_{n=0}^{N-1} \|\boldsymbol{\sigma}_{t-1}(\mathbf{s}_{n,t})\|_2\right] \tag{43}$$

$$r_t^2 \leq L_J^2 N^2 \left(\mathbb{E}_{\boldsymbol{\omega}}\left[\sum_{n=0}^{N-1} \|\boldsymbol{\sigma}_{t-1}(\mathbf{s}_{n,t})\|_2\right]\right)^2 \tag{44}$$

$$\leq L_J^2 N^2 \mathbb{E}_{\boldsymbol{\omega}}\left[\left(\sum_{n=0}^{N-1} \|\boldsymbol{\sigma}_{t-1}(\mathbf{s}_{n,t})\|_2\right)^2\right] \tag{45}$$

$$\leq L_J^2 N^3 \mathbb{E}_{\boldsymbol{\omega}}\left[\sum_{n=0}^{N-1} \|\boldsymbol{\sigma}_{t-1}(\mathbf{s}_{n,t})\|_2^2\right] \tag{46}$$

$\square$

**Lemma 6.** *Under the assumption of Assumptions 1–3, with probability at least $(1 - \delta)$ it holds for all $t \geq 0$ that*

$$R_T^2 \leq T L_J^2 N^3 \sum_{t=1}^{T} \mathbb{E}_{\boldsymbol{\omega}}\left[\sum_{n=0}^{N-1} \|\sigma_{t-1}(\mathbf{s}_{n,t}, \mathbf{a}_{n,t})^2\|_2^2\right] \tag{47}$$

*Proof.*

$$R_T^2 = \left(\sum_{t=1}^{T} r_t\right)^2 \tag{48}$$

$$\leq T\sum_{t=1}^{T} r_t^2 \qquad\qquad \text{Jensen's} \tag{49}$$

$$\leq T L_J^2 N^3 \sum_{t=1}^{T} \mathbb{E}_{\boldsymbol{\omega}}\left[\sum_{n=0}^{N-1} \|\boldsymbol{\sigma}_{t-1}(\mathbf{s}_{n,t}, \mathbf{a}_{n,t})^2\|_2^2\right] \qquad\qquad \text{Lemma 5} \tag{50}$$

$\square$

That is, at every iteration $t$ the regret bound increases by the sum of predictive uncertainties in expectation over the true states that we may visit. This is an instance-dependent bound, since it depends on specific data collected up to iteration $t$ within $\sigma_{t-1}$. We will replace this with a worst-case bound in the following.

**Lemma 7.** *Under the assumption of Assumptions 1–3, let* $\mathbf{s}_{n,t} \in \mathcal{S}_t$, $\mathcal{S}_{t-1} \subseteq \mathcal{S}_t$, *and* $\mathbf{a}_{n,t} \in \mathcal{A}$ *for all* $n, t > 0$ *with compact sets* $\mathcal{S}_t$ *and* $\mathcal{A}$. *Then, with probability at least* $(1 - \delta)$ *it holds for all* $t \geq 0$ *that*

$$R_T^2 \leq T L_J^2 N^3 I_T(\mathcal{S}_t, \mathcal{A}) \tag{51}$$

*where*

$$I_T(\mathcal{S}, \mathcal{A}) = \max_{\mathcal{D}_1, \dots, \mathcal{D}_T \subset \mathcal{S} \times \mathcal{S} \times \mathcal{A}, |\mathcal{D}_i| = N} \sum_{t=1}^{T} \sum_{\mathbf{s}, \mathbf{a} \in \mathcal{D}_t} \|\boldsymbol{\sigma}_{t-1}(\mathbf{s}, \mathbf{a})\|_2^2 \tag{52}$$

*Proof.* As a consequence of $\mathbf{s}_{n,t} \in \mathcal{S}_t$ we have

$$\sum_{t=1}^{T} \mathbb{E}_{\boldsymbol{\omega}} \left[ \sum_{n=0}^{N-1} \|\boldsymbol{\sigma}_{t-1}(\mathbf{s}_{n,t}, \mathbf{a}_{n,t})^2\|_2^2 \right] \leq I_T(\mathcal{S}_t, \mathcal{A}) \tag{53}$$

and thus

$$R_T^2 \leq T L_J^2 N^3 I_T(\mathcal{S}_t, \mathcal{A}). \tag{54}$$

$\square$

**Theorem 2.** *Under Assumptions 1–3 let* $\mathbf{s}_{n,t} \in \mathcal{S}_t$, $\mathcal{S}_{t-1} \subseteq \mathcal{S}_t$, *and* $\mathbf{a}_{n,t} \in \mathcal{A}$ *for all* $n, t > 0$. *Then, for all* $T \geq 1$, *with probability at least* $(1 - \delta)$, *the regret of H-UCRL in* (7) *is at most* $R_T \leq \mathcal{O}\left(\beta_{T-1}^N L_\sigma^N \sqrt{T N^3 I_T(\mathcal{S}_T, \mathcal{A})}\right)$.

*Proof.* From Lemma 7 we have

$$R_T^2 \leq T L_J^2 N^3 I_T(\mathcal{S}_t, \mathcal{A}) \tag{55}$$

$$R_T \leq L_J \sqrt{N^3 I_T(\mathcal{S}_t, \mathcal{A})} \tag{56}$$

where $L_J = 2 L_r \sqrt{1 + L_\pi} \beta_{t-1} \bar{L}_f^{N-1}$ from Lemma 5 and $\bar{L}_f = 1 + L_f + 2\beta_{t-1} L_\sigma \sqrt{1 + L_\pi}$ from Lemma 4. Plugging in we get $L_J = 2 L_r \sqrt{1 + L_\pi} \beta_{t-1} (1 + L_f + 2\beta_{t-1} L_\sigma \sqrt{1 + L_\pi})^{N-1} = \mathcal{O}\left(\beta_{t-1}^N L_\sigma^N\right)$ so that

$$R_T \leq \mathcal{O}\left(\beta_{t-1}^N L_\sigma^N \sqrt{N^3 I_T(\mathcal{S}_t, \mathcal{A})}\right) \tag{57}$$

$\square$

**Theorem 1.** *Under Assumptions 1–3 let* $\mathbf{s}_{n,t} \in \mathcal{S}$ *and* $\mathbf{a}_{n,t} \in \mathcal{A}$ *for all* $n, t > 0$. *Then, for all* $T \geq 1$, *with probability at least* $(1 - \delta)$, *the regret of H-UCRL in* (7) *is at most* $R_T \leq \mathcal{O}\left(L_\sigma^N \beta_{T-1}^N \sqrt{T N^3 I_T(\mathcal{S}, \mathcal{A})}\right)$.

*Proof.* A direct consequence of Theorem 2. $\square$

# E   Properties of the Functions $\eta(\cdot)$

So far, we have considered general functions $\eta \colon \mathbb{R}^p \times \mathbb{R}^q \to [-1, 1]^p$, which can potentially be discontinuous. However, as long as Lemma 1 holds and the true dynamics are feasible in (7), we can use any more restrictive function class. In this section, we investigate properties of $\eta$.

It is clear, that it is sufficient to consider functions $\eta$ such that $\boldsymbol{\Sigma}_t(\mathbf{s})\eta(\mathbf{s})$ is Lipschitz continuous, since it aims to approximate a Lipschitz continuous function $f$:

**Lemma 8.** *With Assumptions 1–3 let* $\eta(\cdot)$ *be a function such that* $f(\mathbf{s}) - \boldsymbol{\mu}_t(\mathbf{s}) = \beta_t \boldsymbol{\Sigma}_t(\mathbf{s})\eta(\mathbf{s})$ *as in Lemma 1. Then* $\boldsymbol{\Sigma}_t(\mathbf{s})\eta(\mathbf{s})$ *is Lipschitz continuous.*

*Proof.*

$$\|\boldsymbol{\Sigma}_t(\mathbf{s})\eta(\mathbf{s}) - \boldsymbol{\Sigma}_t(\mathbf{s}')\eta(\mathbf{s}')\| \leq \|f(\mathbf{s}) - \boldsymbol{\mu}_t(\mathbf{s}) - (f(\mathbf{s}') - \boldsymbol{\mu}_t(\mathbf{s}'))\| \tag{58}$$

$$\leq (L_f + L_\mu)\|\mathbf{s} - \mathbf{s}'\| \tag{59}$$

$\square$

Unfortunately, the same is not true for $\eta$ on its own in general. However, if the predictive standard deviation $\sigma$ does not decay to zero, this holds.

**Lemma 9.** *Under the assumptions of Lemma 8 let $0 < \sigma_{\min} \le \boldsymbol{\sigma}(\mathbf{s}, \mathbf{a}) \le \sigma_{\max}$ elementwise for all $\mathbf{s}, \mathbf{a} \in \mathcal{S} \times \mathcal{A}$. Then, with probability at least $(1 - \delta)$, there exists a Lipschitz-continuous function $\eta(\cdot)$ with $\|\eta(\cdot)\|_\infty = 1$ such that $f(\mathbf{s}) - \boldsymbol{\mu}_t(\mathbf{s}) = \beta_t \boldsymbol{\Sigma}_t(\mathbf{s}) \eta(\mathbf{s})$ for all $\mathbf{s} \in \mathbb{R}^p$.*

*Proof.* By contradiction. Let $\eta(\cdot)$ be a function that is not Lipschitz continuous such that $f(\mathbf{s}) - \boldsymbol{\mu}(\mathbf{s}) = \beta \boldsymbol{\Sigma}(\mathbf{s}) \eta(\mathbf{s})$. By assumption we know that $\sigma_t(\mathbf{s})$ is strictly larger than zero and bounded element-wise from above by some constant. As a consequence, $\boldsymbol{\Sigma}^{-1}(\mathbf{s})$ exists and is $L_\sigma / \sigma_{\min}^2$-Lipschitz continuous w.r.t. the Frobenius norm. Thus, we have

$$\|\eta(\mathbf{s}) - \eta(\mathbf{s}')\|_2$$

$$= \|\frac{1}{\beta} \boldsymbol{\Sigma}^{-1}(\mathbf{s})(f(\mathbf{s}) - \boldsymbol{\mu}(\mathbf{s})) - \frac{1}{\beta} \boldsymbol{\Sigma}^{-1}(\mathbf{s}')(f(\mathbf{s}') - \boldsymbol{\mu}(\mathbf{s}'))\|_2$$

$$\le |\frac{1}{\beta}| \|\boldsymbol{\Sigma}^{-1}(\mathbf{s})((f(\mathbf{s}) - \boldsymbol{\mu}(\mathbf{s})) - (f(\mathbf{s}') - \boldsymbol{\mu}(\mathbf{s}')))\|_2 + |\frac{1}{\beta}| \|\left(\boldsymbol{\Sigma}^{-1}(\mathbf{s}) - \boldsymbol{\Sigma}^{-1}(\mathbf{s}')\right)(f(\mathbf{s}') - \boldsymbol{\mu}(\mathbf{s}'))\|_2$$

$$\le |\frac{1}{\beta}| \|\boldsymbol{\Sigma}^{-1}(\mathbf{s})\|_{\mathrm{F}} \|(f(\mathbf{s}) - \boldsymbol{\mu}(\mathbf{s})) - (f(\mathbf{s}') - \boldsymbol{\mu}(\mathbf{s}'))\|_2 + |\frac{1}{\beta}| \|f(\mathbf{s}') - \boldsymbol{\mu}(\mathbf{s}')\|_2 \|\boldsymbol{\Sigma}^{-1}(\mathbf{s}) - \boldsymbol{\Sigma}^{-1}(\mathbf{s}')\|_{\mathrm{F}}$$

$$\le |\frac{1}{\beta}| \|\boldsymbol{\Sigma}^{-1}(\mathbf{s})\|_{\mathrm{F}} (L_{\mathrm{fc}} + L_\mu \sqrt{1 + L_\pi}) \|\mathbf{s} - \mathbf{s}'\|_2 + |\frac{1}{\beta}| \|\beta \boldsymbol{\sigma}(\mathbf{s}')\|_2 \|\boldsymbol{\Sigma}^{-1}(\mathbf{s}) - \boldsymbol{\Sigma}^{-1}(\mathbf{s}')\|_{\mathrm{F}}$$

$$\le \frac{\sqrt{p}}{\beta \sigma_{\min}} (L_{\mathrm{fc}} + L_\mu \sqrt{1 + L_\pi}) \|\mathbf{s} - \mathbf{s}'\|_2 + \frac{\sqrt{p} \sigma_{\max}}{\sigma_{\min}^2} L_\sigma \sqrt{1 + L_\pi} \|\mathbf{s} - \mathbf{s}'\|_2$$

Since $\beta_t > 0$ we have that $\eta(\mathbf{s})$ is Lipschitz continuous, which is a contradiction. $\square$

Thus, it is generally sufficient to optimize over Lipschitz continuous functions in order to obtain the same regret bounds as in the optimistic case. However, it is important to note that the complexity of the function (i.e., its Lipschitz constant) will generally increase as the predictive variance decreases. It is easy to construct cases where $\boldsymbol{\sigma}(\cdot) = 0$ implies that $\eta$ has to be discontinuous. However, at least in theory $\boldsymbol{\sigma}(\cdot) = 0$ is impossible with finite data when the system is noisy ($\sigma > 0$). Also note that as $\boldsymbol{\sigma}$ decreases, the effect of $\eta$ on the dynamics also decreases.

This might also motivate optimizing over a function that model $\boldsymbol{\Sigma}_{t-1}(\mathbf{s}, \mathbf{a}) \eta(\mathbf{s}, \mathbf{a})$ jointly, since that one is regular even for $\boldsymbol{\sigma}(\cdot) = 0$. However, this would require regularizing the resulting function to be bounded by $\beta_t \boldsymbol{\sigma}_t(\mathbf{s}, \mathbf{a})$ and might lead to difficulties with policy optimization, since the resulting hallucinated actions are no longer normalized to $[-1, 1]^p$. We leave it as an avenue for future research.

## F  Background on Gaussian Processes

Gaussian processes are a nonparametric Bayesian model that has a tractable, closed-form posterior distribution (Rasmussen and Williams, 2006). The goal of Gaussian process inference is to infer a posterior distribution over a nonlinear map $f'(\mathbf{x}) : \mathcal{X} \to \mathbb{R}$ from an input vector $\mathbf{x} \in \mathcal{X}$ with $\mathcal{X} \subseteq \mathbb{R}^d$ to the function value $f'(\mathbf{x})$. This is accomplished by assuming that the function values $f'(\mathbf{x})$, associated with different values of $\mathbf{x}$, are random variables and that any finite number of these random variables have a *joint* normal distribution (Rasmussen and Williams, 2006).

A Gaussian process distribution is parameterized by a prior mean function and a covariance function or kernel $k(\mathbf{x}, \mathbf{x}')$, which defines the covariance of any two function values $f(\mathbf{x})$ and $f(\mathbf{x}')$ for $\mathbf{x}, \mathbf{x}' \in \mathcal{X}$. In this work, the mean is assumed to be zero without loss of generality. The choice of kernel function is problem-dependent and encodes assumptions about the unknown function. A review of potential kernels can be found in (Rasmussen and Williams, 2006).

We can condition a Gaussian process on the observations $\mathbf{y}_t$ at input locations $\mathcal{X}_t$. The Gaussian process model assumes that observations are noisy measurements of the true function value with Gaussian noise, $\omega \sim \mathcal{N}(0, \sigma^2)$. The posterior distribution is again a Gaussian process with mean $\mu_t$,

covariance $k_t$, and variance $\sigma_t$, where

$$\mu_t(\mathbf{x}) = \mathbf{k}_t(\mathbf{x})(\mathbf{K}_t + \mathbf{I}\sigma^2)^{-1}\mathbf{y}_t, \tag{60}$$

$$k_t(\mathbf{x}, \mathbf{x}') = k(\mathbf{x}, \mathbf{x}') - \mathbf{k}_t(\mathbf{x})(\mathbf{K}_t + \mathbf{I}\sigma^2)^{-1}\mathbf{k}_t^{\mathrm{T}}(\mathbf{x}'), \tag{61}$$

$$\sigma_t^2(\mathbf{x}) = k_t(\mathbf{x}, \mathbf{x}). \tag{62}$$

The covariance matrix $\mathbf{K}_t \in \mathbb{R}^{|\mathcal{X}_t| \times |\mathcal{X}_t|}$ has entries $[\mathbf{K}_t]_{(i,j)} = k(\mathbf{x}_i, \mathbf{x}_j)$ with $\mathbf{x}_i, \mathbf{x}_j \in \mathcal{X}_t$ and the vector $\mathbf{k}_t(\mathbf{x}) = [k(\mathbf{x}, \mathbf{x}_1), \dots, k(\mathbf{x}, \mathbf{x}_{|\mathcal{X}_t|})]$ contains the covariances between the input $\mathbf{x}$ and the observed data points in $\mathcal{X}_t$. The identity matrix is denoted by $\mathbf{I}$.

Given the Gaussian process assumptions, we obtain point-wise confidence estimates from the marginal Normal distribution specified by $\mu_t$ and $\sigma_t$. For finite sets, the Gaussian process belief induces a *joint* normal distribution over function values that is correlated through (61). We can use this to fulfill Assumption 2 for continuous sets by using a union bound and exploiting that samples from a Gaussian process are Lipschitz continuous with high probability (Srinivas et al., 2012, Theorem 2).

## F.1 Information Capacity

One important property of normal distributions is that the confidence intervals contract after we observe measurement data. How much data we require for this to happen generally depends on the variance of the observation noise, $\sigma^2$, and the size of the function class; i.e., the assumptions that we encode through the kernel. In the following, we use results by Srinivas et al. (2012) and use the mutual information to construct such a capacity measure.

Formally, the mutual information between the Gaussian process prior on $f'$ at locations $\overline{\mathcal{X}}$ and the corresponding noisy observations $\mathbf{y}_{\overline{\mathcal{X}}}$ is given by

$$\mathrm{I}(\mathbf{y}_{\overline{\mathcal{X}}}; f') = 0.5 \log |\mathbf{I} + \sigma^{-2}\mathbf{K}_{\overline{\mathcal{X}}}|, \tag{63}$$

where $\mathbf{K}_{\overline{\mathcal{X}}}$ is the kernel matrix $[k(\mathbf{x}, \mathbf{x}')]_{\mathbf{x}, \mathbf{x}' \in \overline{\mathcal{X}}}$ and $|\cdot|$ is the determinant. Intriguingly, for Gaussian process models this quantity only depends on the inputs in $\overline{\mathcal{X}}$ and not the corresponding measurements $\mathbf{y}_{\overline{\mathcal{X}}}$. Intuitively, the mutual information measures how informative the collected samples $\mathbf{y}_{\mathcal{X}}$ are about the function $f$. If the function values are independent of each other under the Gaussian process prior, they provide large amounts of new information. However, if measurements are taken close to each other as measured by the kernel, they are correlated under the Gaussian process prior and provide less information.

The mutual information in (63) depends on the locations $\mathcal{X}_t$ at which we obtain measurements. While it can be computed in closed-form, it can also be bounded by the largest mutual information that any algorithm could obtain from $t$ noisy observations,

$$\gamma_t = \max_{\mathcal{X} \subset D, |\mathcal{X}| \leq t} \mathrm{I}(\mathbf{y}_{\mathcal{X}}; f'). \tag{64}$$

We refer to $\gamma_t$ as the *information capacity*, since it can be interpreted as a measure of complexity of the function class associated with a Gaussian process prior. It was shown by Srinivas et al. (2012) that $\gamma_t$ has a sublinear dependence on $t$ for many commonly used kernels such as the Gaussian kernel. This sublinear dependence is generally exploited by exploration algorithms in order to show convergence.

## F.2 Functions in a Reproducing Kernel Hilbert Space

Instead of the Bayesian Gaussian process framework, we can also consider frequentist confidence intervals. Unlike the Bayesian framework, which inherently models a belief over a random function, frequentists assume that there is an *a priori* fixed underlying function $f'$ of which we observe noisy measurements.

The natural frequentist counterpart to Gaussian processes are functions inside the Reproducing Kernel Hilbert Space (RKHS) spanned by the same kernel $k(\mathbf{x}, \mathbf{x}')$ as used by the Gaussian process in Appendix F. An RKHS $\mathcal{H}_k$ contains well-behaved functions of the form $f(\mathbf{x}) = \sum_{i \geq 0} \alpha_i k(\mathbf{x}, \mathbf{x}_i)$, for given representer points $\mathbf{x}_i \in \mathbb{R}^d$ and weights $\alpha_i \in \mathbb{R}$ that decay sufficiently quickly. For example, the Gaussian process mean function (60) lies in this RKHS. The kernel function $k(\cdot, \cdot)$ determines

the roughness and size of the function space and the induced RKHS norm $\|f'\|_k^2 = \langle f', f'\rangle_k = \sum_{i,j \geq 0} \alpha_i \alpha_j k(\mathbf{x}_i, \mathbf{x}_j)$ measures the complexity of a function $f' \in \mathcal{H}_k$ with respect to the kernel. In particular, the function $f'$ is Lipschitz continuous with respect to the kernel metric

$$d(\mathbf{x}, \mathbf{x}') = \sqrt{k(\mathbf{x}, \mathbf{x}) + k(\mathbf{x}', \mathbf{x}') - 2k(\mathbf{x}, \mathbf{x}')}, \tag{65}$$

so that $|f'(\mathbf{x}) - f'(\mathbf{x}')| \leq \|f'\|_k d(\mathbf{x}, \mathbf{x}')$, see the proof of Proposition 4.30 by Christmann and Steinwart (2008).

### F.2.1 Confidence Intervals

We can construct an estimate together with reliable confidence intervals if the measurements are corrupted by $\sigma$-sub-Gaussian noise. This is a class of noise where the tail probability decays exponentially fast, such as in Gaussian random variables or any distribution with bounded support. Specifically, we have the following definition.

**Definition 1** (Vershynin (2010)). A random variable $X$ is $\sigma$-sub-Gaussian if $\mathbb{P}\{|X| > s\} \leq \exp(1 - s^2/\sigma^2)$ for all $s > 0$.

While the Gaussian process framework makes different assumptions about the function and the noise, Gaussian processes and RKHS functions are closely related (Kanagawa et al., 2018) and it is possible to use the Gaussian process posterior marginal distributions to infer reliable confidence intervals on $f'$.

**Lemma 10** (Abbasi-Yadkori (2012); Chowdhury and Gopalan (2017)). *Assume that $f$ has bounded RKHS norm $\|f'\|_k \leq B$ and that measurements are corrupted by $\sigma$-sub-Gaussian noise. If $\beta_t^{1/2} = B + 4\sigma\sqrt{\mathrm{I}(\mathbf{y}_t; f) + 1 + \ln(1/\delta)}$, then for all $\mathbf{x} \in \mathcal{X}$ and $t \geq 0$ it holds jointly with probability at least $1 - \delta$ that $|f'(\mathbf{x}) - \mu_t(\mathbf{x})| \leq \beta_t^{1/2}\sigma_t(\mathbf{x})$.*

Lemma 10 implies that, with high probability, the true function $f'$ is contained in the confidence intervals induced by the posterior Gaussian process distribution that uses the kernel $k$ from Lemma 10 as a covariance function, scaled by an appropriate factor $\beta_t$. In contrast to Appendix F, Lemma 10 does not make probabilistic assumptions on $f'$. In fact, $f'$ could be chosen adversarially, as long as it has bounded norm in the RKHS.

Since the frequentist confidence intervals depend on the mutual information and the marginal confidence intervals of the Gaussian process model, they inherit the same contraction properties up to the factor $\beta_t$. However, note that the confidence intervals in Lemma 10 hold jointly through the continuous domain $\mathcal{X}$. This is not generally possible for Gaussian process models without employing additional continuity arguments, since Gaussian process distributions are by definitions only defined via a multivariate Normal distribution over *finite* sets. This stems from the difference between a Bayesian belief and the frequentist perspective, where the function is unknown but fixed *a priori*.

### F.3 Extension to multiple dimensions

It is straight forward to extend these models to functions with vector-values outputs by extending the input domain by an extra input argument that indexes the output dimension. While this requires special kernels, they have been analyzed by Krause and Ong (2011).

**Lemma 11** (based on Chowdhury and Gopalan (2017)). *Assume that $f'(\boldsymbol{\theta}, i) = [f'(\boldsymbol{\theta})]_i$ has RKHS norm bounded by $B$ and that measurements are corrupted by $\sigma$-sub-Gaussian noise. Let $\mathcal{X}_t = \mathcal{D}_t \times \mathcal{I}$ denote the measurements obtained up to iteration $t$. If $\beta_t = B + 4\sigma\sqrt{\mathrm{I}(\mathbf{y}_{\mathcal{X}_t}; f') + 1 + \ln(1/\delta)}$, then the following holds for all parameters $\boldsymbol{\theta} \in \mathcal{D}$, function indices $i \in \mathcal{I}$, and iterations $n \geq 0$ jointly with probability at least $1 - \delta$:*

$$\left| f'(\boldsymbol{\theta}, i) - \mu_n(\boldsymbol{\theta}, i) \right| \leq \beta_n \sigma_n(\boldsymbol{\theta}, i) \tag{66}$$

## G   Lipschitz Continuity of Gaussian Process Predictions

Since the mean function is a linear combination of kernels evaluations (features), it is easy to show that it is Lipschitz continuous if the kernel function is Lipschitz continuous (Lederer et al., 2019).

However, existing bounds for the Lipschitz constant for the posterior standard deviation $\sigma_t(\cdot)$ depend on the number of data points. Since our regret bounds depend on $L_\sigma^N$, this would render our regret bound superlinear and thus meaningless.

In the following, we show that the GP standard deviation is Lipschitz-continuous with respect to the kernel metric.

**Definition 2** (Kernel metric). $d_k(\mathbf{x}, \mathbf{x}') = \sqrt{k(\mathbf{x}, \mathbf{x}) + k(\mathbf{x}', \mathbf{x}') - 2k(\mathbf{x}, \mathbf{x}')}$.

We start with the standard deviation.

**Lemma 12.** *For all* $\mathbf{x}$ *and* $\mathbf{x}'$ *in* $\mathcal{X}$ *and all* $t \geq 0$*, we have*

$$|\sigma_t(\mathbf{x}) - \sigma_t(\mathbf{x}')| \leq d_k(\mathbf{x}, \mathbf{x}') \tag{67}$$

*Proof.* From Mercer's theorem we know that each kernel can be equivalently written in terms of an infinite-dimensional inner product, so that $k(\mathbf{x}, \mathbf{x}') = \langle k(\mathbf{x}, \cdot), k(\mathbf{x}', \cdot) \rangle_k$, where $< \cdot, \cdot >_k$ is the inner product in the Reproducing Kernel Hilbert Space corresponding to the kernel $k$. We can think of Gaussian process regression as linear regression based on these infinite-dimensional feature vectors. In particular, it follows from (Kirschner and Krause, 2018, Appendix D) that we can write the Gaussian process posterior standard deviation $\sigma_t(\mathbf{x})$ as the weighted norm of the infinite-dimensional feature vectors $k(\mathbf{x}, \cdot)$,

$$\sigma_t(\mathbf{x}) = \|k(\mathbf{x}, \cdot)\|_{\mathbf{V}_t^{-1}}, \tag{68}$$

where $\mathbf{V}_t = \sigma^2 \mathbf{M}^* \mathbf{M} + \mathbf{I}$ and $\mathbf{M}$ is a linear operator that corresponds to the infinite-dimensional feature vectors $k(\mathbf{x}_i, \cdot)$ of the data points $\mathbf{x}_i$ in $\mathcal{X}_t$ so that $[\mathbf{M}\mathbf{M}^*]_{(i,j)} = k(\mathbf{x}_i, \mathbf{x}_j)$, where $\mathbf{x}_i$ and $\mathbf{x}_j$ are the $i$th and $j$th data point in $\mathcal{X}_t$. Now we have that the minimum eigenvalue of $\mathbf{V}_t$ is larger or equal than one, which implies that the maximum eigenvalue of $\mathbf{V}_t^{-1}$ is less or equal than one. Thus,

$$|\sigma_t(\mathbf{x}) - \sigma_t(\mathbf{x}')| = \left| \|k(\mathbf{x}, \cdot)\|_{\mathbf{V}_t^{-1}} - \|k(\mathbf{x}', \cdot)\|_{\mathbf{V}_t^{-1}} \right| \tag{69}$$

$$\leq \|k(\mathbf{x}, \cdot) - k(\mathbf{x}', \cdot)\|_{\mathbf{V}_t^{-1}}, \tag{70}$$

$$\leq \|k(\mathbf{x}, \cdot) - k(\mathbf{x}', \cdot)\|_k, \tag{71}$$

$$= \sqrt{\langle k(\mathbf{x}, \cdot) - k(\mathbf{x}', \cdot), k(\mathbf{x}, \cdot) - k(\mathbf{x}', \cdot) \rangle_k}, \tag{72}$$

$$= \sqrt{k(\mathbf{x}, \mathbf{x}) - k(\mathbf{x}, \mathbf{x}') - k(\mathbf{x}', \mathbf{x}) + k(\mathbf{x}', \mathbf{x}')}, \tag{73}$$

$$= \sqrt{k(\mathbf{x}, \mathbf{x}) + k(\mathbf{x}', \mathbf{x}') - 2k(\mathbf{x}, \mathbf{x}')}, \tag{74}$$

$$= d_k(\mathbf{x}, \mathbf{x}'), \tag{75}$$

where (69) $\rightarrow$ (70) follows from the reverse triangle inequality. $\square$

To show that Lemma 12 implies Lipschitz continuity of the variance, the key observation is that standard deviation $\sigma_n i(\mathbf{x})$ is bounded for all $t \geq 0$. In particular,

$$\sigma_t(\mathbf{x}) \leq \sigma_0(\mathbf{x}) = \sqrt{k(\mathbf{x}, \mathbf{x})} \leq \max_{\mathbf{x}, \mathbf{x}' \in \mathbb{R}^d} \sqrt{k(\mathbf{x}, \mathbf{x}')} := \sqrt{|k|_\infty} \tag{76}$$

Based on this, we have the following result.

**Lemma 13.** *For all* $\mathbf{x}$ *and* $\mathbf{x}'$ *in* $\mathcal{X}$ *and all* $t \geq 0$*, we have*

$$|\sigma_t(\mathbf{x}) - \sigma_t(\mathbf{x}')| \leq 2\sqrt{|k|_\infty}\, d_k(\mathbf{x}, \mathbf{x}') \tag{77}$$

*Proof.* For any compact domain $\mathcal{D}$ the function $f(x) = x^2$ is Lipschitz continuous for $\mathbf{s} \in \mathcal{D}$ with Lipschitz constant $|df/dx|_\infty = \max_{x \in \mathcal{D}} 2|x|$. Since $0 \leq \sigma_t(\mathbf{x}) \leq \sqrt{|k|_\infty}$, we have

$$|\sigma_t^2(\mathbf{x}) - \sigma_t^2(\mathbf{x}')| \leq 2\sqrt{|k|_\infty}\, |\sigma_t(\mathbf{x}) - \sigma_t(\mathbf{x}')| \tag{78}$$

$$\leq 2\sqrt{|k|_\infty}\, d_k(\mathbf{x}, \mathbf{x}') \tag{79}$$

$\square$

# H    Regret Bound for Gaussian Process model

## H.1    Assumptions about the model

**Assumption 4.** The both the kernel and the kernel metric (65) are Lipschitz continuous.

Note that the kernel metric is not trivially Lipschitz if the kernel is Lipschitz, since the square root function has unbounded derivatives at zero. However, for many commonly used kernels, e.g., the linear and squared exponential kernels, the kernel metric is in fact Lipschitz continuous.

As a direct consequence of Assumption 4 together with Appendix G we know that $\boldsymbol{\sigma}(\cdot)$ is $L_\sigma$-Lipschitz continuous.

**Assumption 5.** The model $f$ has RKHS norm bounded by $B_f$ with respect to a kernel that fulfilles Assumption 4 and $k((\mathbf{s}, \pi(\mathbf{s}), (\mathbf{s}, \pi(\mathbf{s})) \leq 1$ for all $\pi \in \Pi$ and $\mathbf{s} \in \mathcal{S}$.

This assumption allows us to learn a calibrated model of the function $g$. Note that the assumption of a bounded kernel over a compact domain $\mathcal{S}$ is mild, since any scaling can be absorbed into the constant $B_f$. We weaken this assumption in Appendix I, where we bound the domain $\mathcal{S}$ rather than assuming compactness.

Since RKHS functions are linear combinations of the kernel function evaluated at representer points, the continuity assumptions on the kernel directly transfer to continuity assumptions on the function $f$, so that we get the following result.

**Corollary 3.** *Under Assumption 5, the dynamics function $f$ is $L_f$-Lipschitz continuous with respect to the 2-norm.*

*Proof.* For scalar functions, this is a direct consequence of Assumption 5 and (Christmann and Steinwart, 2008, Cor. 4.36). This directly generalizes to the vector case. $\qquad\square$

Since the state $\mathbf{s}$ is observed directly, the Assumption 5 allows us to learn a reliable statistical model of $f$ that conforms with the requirement of a well-calibrated model in Assumption 2. In particular, for each transition from $(\mathbf{s}_n, \mathbf{a}_n)$ to $\mathbf{s}_{n+1}$, we add $p$ observations, one for each output dimension, to $\mathcal{D}_t$ as in Lemma 11.

**Corollary 4.** *Under Assumptions 1 and 5 with $\beta_t$ as in Lemma 11 and a Gaussian process model trained on observations $\mathbf{x}_{n+1}$ based on an input $\mathbf{a} = (\mathbf{s}_n, \mathbf{a}_n)$, the following holds with probability $1 - \delta$ for all $t \geq 0$, $\mathbf{s} \in \mathbb{R}^p$, and $\mathbf{a} \in \mathbb{R}^q$:*

$$|f(\mathbf{s}, \mathbf{a}, i) - \mu_t(\mathbf{s}, \mathbf{a}, i)| \leq \beta_t \sigma_t(\mathbf{s}, \mathbf{a}, i) \tag{80}$$

In the following, we write

$$\boldsymbol{\mu}_t(\mathbf{s}, \mathbf{a}) = (\mu_{tNp}(\mathbf{s}, \mathbf{a}, 1), \dots, \mathbf{a}_{tNp}(\mathbf{s}, \mathbf{a}, p)), \tag{81}$$

$$\boldsymbol{\sigma}_t(\mathbf{s}, \mathbf{a}) = (\sigma_{tNp}(\mathbf{s}, \mathbf{a}, 1), \dots, \sigma_{tNp}(\mathbf{s}, \mathbf{a}, p)) \tag{82}$$

to represent the individual elements as vectors. Note that $\boldsymbol{\mu}_t$ is conditioned on the $tNp$ individual one-dimensional observations after $t$ episodes. Corollary 4 allows us to build confidence intervals on the model error $g$ based on the scaled Gaussian process posterior variance. A direct consequence of these point-wise error bounds is that we can also bound the norm of the error on the vector-output of $f$.

**Corollary 5.** *Under the assumption of Corollary 4, with probability $1 - \delta$ we have for all $t \geq 0$, $\mathbf{s} \in \mathbb{R}^p$, and $\mathbf{a} \in \mathbb{R}^q$ that*

$$\|f(\mathbf{s}, \mathbf{a}) - h(\mathbf{s}, \mathbf{a}) - \boldsymbol{\mu}_t(\mathbf{s}, \mathbf{a})\|_2 \leq \beta_t \|\sigma_t(\mathbf{s}, \mathbf{a})\|_2 \tag{83}$$

*Proof.*

$$\|f(\mathbf{s}, \mathbf{a}) - \boldsymbol{\mu}_t(\mathbf{s}, \mathbf{a})\|_2 = \left( \sum_{i=1}^{p} |f(\mathbf{s}, \mathbf{a}, i) - \mu_t(\mathbf{s}, \mathbf{a}, i)|^2 \right)^{1/2} \tag{84}$$

$$\leq \left( \sum_{i=1}^{p} |\beta_t \sigma_t(\mathbf{s}, \mathbf{a}, i)|^2 \right)^{1/2} = \beta_t \|\boldsymbol{\sigma}_t(\mathbf{s}, \mathbf{a})\|_2 \tag{85}$$

$\square$

## H.2   Bounding $I_T$ for the GP model

In this section, we bound $I_T$ based on the GP assumptions. This allows us to use them together with Theorem 2 to obtain regret bounds. We start with some preliminary lemmas

**Lemma 14** (Srinivas et al. (2012)). $s^2 \leq \frac{s_{\max}^2}{\log(1 + s_{\max}^2)} \log(1 + s^2)$ *for all* $s \in [0, s_{\max}^2]$

**Lemma 15.** *Let* $|\sigma_t(\cdot)| \leq \sigma_{\max}$ *and* $\sigma > 0$. *Then*

$$\sigma_t^2(\mathbf{x}) \leq \frac{\sigma_{\max}}{\log(1 + \sigma^{-2}\sigma_{\max})} \log(1 + \sigma^{-2}\sigma_t^2(\mathbf{x})) \tag{86}$$

*Proof.*

$$\sigma_t^2(\mathbf{x}) \leq \sigma^2(\sigma^{-2}\sigma_t^2(\mathbf{x})) \tag{87}$$

Now $\sigma^{-2}\sigma_t^2(\mathbf{x}) \leq \sigma^{-2}\sigma_{\max}$ by assumption. Thus, we can use Lemma 14 to obtain

$$\sigma_t^2(\mathbf{x}) \leq \sigma^2 \frac{\sigma^{-2}\sigma_{\max}}{\log(1 + \sigma^{-2}\sigma_{\max})} \log(1 + \sigma^{-2}\sigma_t^2(\mathbf{x})) \tag{88}$$

$$= \frac{\sigma_{\max}}{\log(1 + \sigma^{-2}\sigma_{\max})} \log(1 + \sigma^{-2}\sigma_t^2(\mathbf{x})) \tag{89}$$

$\square$

**Lemma 16.** *Let* $\mathcal{D}_{1:T}$ *denote the* $TN$ *p-dimensional observations collected up to iteration* $t$ *and* $\mathbf{y}_{\mathcal{D}_{1:t}}$ *the corresponding observations of the following states. Then*

$$\frac{1}{2} \sum_{t=1}^{T} \sum_{n=0}^{N-1} \sum_{j=1}^{p} \log(1 + \sigma^{-2}\sigma_{(t-1)Np}^2(\mathbf{x}_{n,t}, j)) \leq Np\, \mathrm{I}(\mathbf{y}_{\mathcal{D}_T}; f_{\mathcal{D}_T}) \tag{90}$$

*Proof.*

$$\frac{1}{2} \sum_{t=1}^{T} \sum_{n=0}^{N-1} \sum_{j=1}^{p} \log(1 + \sigma^{-2}\sigma_{(t-1)Np}^2(\mathbf{x}_{n,t}, j)) \tag{91}$$

$$= \sum_{n=0}^{N-1} \sum_{j=1}^{p} \frac{1}{2} \sum_{t=1}^{T} \log(1 + \sigma^{-2}\sigma_{(t-1)Np}^2(\mathbf{x}_{n,t}, j)) \tag{92}$$

$$\leq Np\, \mathrm{I}(\mathbf{y}_{\mathcal{D}_{1:T}}; f_{\mathcal{D}_{1:T}}) \tag{93}$$

Where the second to last step follows from (Srinivas et al., 2012, Lemma 2) together with $\log(1+x) \geq 0$ for $x \geq 0$ and the properties of the mutual information. In particular, the inner sum conditions on $(t-1)Np$ measurements, but sums only over the one element $(\mathbf{x}_{n,t}, j)$. The mutual information in (Srinivas et al., 2012, Lemma 2) instead sums over every element that we condition on in the next step. By adding the missing non-negative terms together with the fact that the mutual information is independent of the order of the observations we obtain the result. Another way to interpret this bound is that, in the worst case, we could hypothetically visit $N$ times the same state during a trajectory and obtain the corresponding $p$-dimensional observation. This explains the $Np$ factor that multiplies the mutual information. $\square$

We can use these two lemmas to obtain:

**Lemma 17.** *For a GP model let $|\sigma_t(\cdot)| \leq \sigma_{\max}$ and $\sigma > 0$. Then*

$$I_T(\mathcal{S}, \mathcal{A}) \leq \frac{\sigma_{\max} N p}{\log(1 + \sigma^{-2}\sigma_{\max})} \gamma_{TNp}(\mathcal{S} \times \mathcal{A} \times \mathcal{I}_p) \tag{94}$$

*Proof.*

$$I_T(\mathcal{S}, \mathcal{A}) = \max_{\mathcal{D}_1,\ldots,\mathcal{D}_T \subset \mathcal{S} \times \mathcal{S} \times \mathcal{A}, |\mathcal{D}_t|=N} \sum_{t=1}^{T} \sum_{\mathbf{s},\mathbf{a} \in \mathcal{D}_t} \|\boldsymbol{\sigma}_{t-1}(\mathbf{s}, \mathbf{a})\|_2^2 \tag{95}$$

$$= \max_{\mathcal{D}_1,\ldots,\mathcal{D}_T \subset \mathcal{S} \times \mathcal{S} \times \mathcal{A}, |\mathcal{D}_t|=N} \sum_{t=1}^{T} \sum_{\mathbf{s},\mathbf{a} \in \mathcal{D}_t} \sum_{j=1}^{p} \sigma^2_{(t-1)Np}(\mathbf{s}, \mathbf{a}, j) \tag{96}$$

$$\leq \frac{\sigma_{\max}}{\log(1 + \sigma^{-2}\sigma_{\max})} \max_{\mathcal{D}_1,\ldots,\mathcal{D}_T \subset \mathcal{S} \times \mathcal{S} \times \mathcal{A}, |\mathcal{D}_t|=N} \sum_{t=1}^{T} \sum_{\mathbf{s},\mathbf{a} \in \mathcal{D}_t} \sum_{j=1}^{p} \log(1 + \sigma^{-2}\sigma^2_{(t-1)Np}(\mathbf{s}, \mathbf{a}, j)) \tag{97}$$

$$\leq \frac{\sigma_{\max}}{\log(1 + \sigma^{-2}\sigma_{\max})} \max_{\mathcal{D}_1,\ldots,\mathcal{D}_T \subset \mathcal{S} \times \mathcal{S} \times \mathcal{A}, |\mathcal{D}_t|=N} Np\, \mathrm{I}(\mathbf{y}_{\mathcal{D}_{1:T}}; f_{\mathcal{D}_{1:T}}) \tag{98}$$

$$\leq \frac{\sigma_{\max} N p}{\log(1 + \sigma^{-2}\sigma_{\max})} \gamma_{TNp}(\mathcal{S} \times \mathcal{A} \times \mathcal{I}_p) \tag{99}$$

$\square$

To obtain an instance-independent bound, we must bound the mutual information by the worst-case mutual information as in (Srinivas et al., 2012).

**Theorem 3.** *Under Assumptions 1–3 let $\mathbf{s}_{n,t} \in \mathcal{X}_t$, $\mathcal{S}_{t-1} \subseteq \mathcal{S}_t$, and $\mathbf{a}_{n,t} \in \mathcal{U}$ for all $n, t > 0$ with compact sets $\mathcal{S}_t$ and $\mathcal{A}$. Let $\|\boldsymbol{\sigma}(\cdot)\|_\infty \leq \sigma_{\max}$. At each iteration, select parameters according to (7). Then the following holds with probability at least $(1 - \delta)$ for all $t \geq 1$*

$$R_T \leq \mathcal{O}\left(\beta_{T-1}^N L_\sigma^N N^2 \sqrt{T p\, \gamma_{pTN}(\mathcal{S}_T \times \mathcal{A} \times \mathcal{I}_p)}\right), \tag{100}$$

*where $\gamma_{pTN}(\mathcal{S} \times \mathcal{A} \times \mathcal{I}_p)$ is the information capacity after $(ptN)$ observations within the extended domain $\mathcal{S} \times \mathcal{A} \times \mathcal{I}_p$.*

*Proof.* From Theorem 2 we have $R_T^2 \leq TL_J^2 N^3 I_T(\mathcal{S}_t, \mathcal{A})$. Together with Lemma 17 we obtain

$$R_T \leq L_J \sqrt{N^3 I_T(\mathcal{S}_t, \mathcal{A})} \tag{101}$$

$$\leq L_J \left(\frac{\sigma_{\max} N^4 p}{\log(1 + \sigma^{-2}\sigma_{\max})} \gamma_{TNp}(\mathcal{S} \times \mathcal{A} \times \mathcal{I}_p)\right)^{1/2} \tag{102}$$

where $L_J = 2L_r(1 + L_\pi)\beta_{t-1}\bar{L}_f^{N-1}$ from Lemma 5 and $\bar{L}_f = 1 + L_f + 2\beta_{t-1}L_\sigma\sqrt{1 + L_\pi}$ from Lemma 4. Plugging in we get $L_J = 2L_r(1 + L_\pi)\beta_{t-1}(1 + L_f + 2\beta_{t-1}L_\sigma\sqrt{1 + L_\pi})^{N-1} = \mathcal{O}(\beta_{t-1}^N L_\sigma^N)$ so that

$$R_T \leq \mathcal{O}\left(L_\sigma^N \beta_{T-1}^N N^2 \sqrt{Tp\gamma_{pTN}(\mathcal{S}_t \times \mathcal{A} \times \mathcal{I}_p)}\right) \tag{103}$$

$\square$

Notably, unlike in Theorem 1 we can actually bound the information capacity $\gamma$ in Theorem 3. For a GP model that uses a squared exponential kernel with independent outputs, we have $\gamma_{pTN} \leq \mathcal{O}(p(p+q)\log(pTN))$ by (Srinivas et al., 2012; Krause and Ong, 2011), which renders the overall regret bound sublinear. Note that for the Matern kernel the best known bound on $\gamma_{pTN}$ is $\mathcal{O}(p(pTN)^c \log(pTN))$ with $0 < c < 1$. This means the regret bound is not sublinear for long trajectories due to the $\beta_t^N$ term in the regret bound. However, the bound is expected to be loose (Scarlett et al., 2017). Tighter bounds can be computed numerically, see (Srinivas et al., 2012, Fig. 3).

Note that the requirement $\|\boldsymbol{\sigma}(\cdot)\|_\infty$ if fulfilled according to

**Lemma 18.** *Under Assumption 5 we have $\boldsymbol{\sigma}(\mathbf{s}) \leq 1$ for all $\mathbf{s} \in \mathcal{S}$.*

*Proof.* This is a direct consequence of (62).  □

### H.3   Comparison to Chowdhury and Gopalan (2019)

In this section, we compare our bound to the one by Chowdhury and Gopalan (2019). This is a difficult endeavour, because they make fundamentally different assumptions. In particular, they assume that the value function $v(x)$ is $L_M$-Lipschitz continuous, which hides all the complexity of thinking about different trajectories, as deviations between the two trajectories can be bounded after one step by $L_M \|\mathbf{s}_1 - \tilde{\mathbf{s}}_1\|$. In contrast, we do not make this high-level assumption and specifically reason about the entire trajectories based on system properties. Note, that the constant $L_M$ is at least $\Omega(N)$ without additional assumptions about the system and generally will depend on the statistical model (GP).

Secondly, they restrict the optimization over dynamics that are Lipschitz continuous, which means their algorithm depends on system properties that are difficult to estimate in general. However, this assumption avoids the dependency $\beta^N$ in our regret bound, since it limits optimization to trajectories that are at most as smooth as the dynamics of the true system. The cost of this is that their algorithm is not tractable to implement or compute.

For completeness, in the following we modify our proof to use their assumption and show a regret bound that is comparable to the one by Chowdhury and Gopalan (2019).

#### H.3.1   Our bound under the assumptions of (Chowdhury and Gopalan, 2019)

Now, we show that if we assume that the optimistic dynamics are Lipschitz, which together with a Lipschitz-continuous policy implies the Lipschitz continuity of the value function that is assumed by Chowdhury and Gopalan (2019), we obtain the same regret bounds.

Let

$$\widetilde{\mathcal{M}}_t = \big\{ f' \,|\, |\boldsymbol{\mu}(\mathbf{s},\mathbf{a}) - f'(\mathbf{s},\mathbf{a})| \leq \beta\boldsymbol{\sigma}(\mathbf{s},\mathbf{a}) \,\forall \mathbf{s}, \mathbf{a} \in \mathbb{R}^p \times \mathbb{R}^q,$$
$$\|f'(\mathbf{s},\mathbf{a}) - f'(\mathbf{s}',\mathbf{a}')\| \leq L_f \|(\mathbf{s},\mathbf{a}) - (\mathbf{s}',\mathbf{a}')\| \,\forall (\mathbf{s},\mathbf{a}), (\mathbf{s}',\mathbf{a}') \in \mathbb{R}^p \times \mathbb{R}^q, \big\}$$

be the set of all Lipschitz continuous dynamics that are compatible with the uncertainty representation in Assumption 2. We now consider a variant of (7) that optimizes over dynamics in this set,

$$\pi_t = \operatorname*{argmax}_{\pi \in \Pi, \, \tilde{f}_t \in \widetilde{\mathcal{M}}_t} J(\tilde{f}_t, \pi) \tag{104}$$

and we implicitly define $\tilde{\mathbf{s}}$ and $\tilde{\mathbf{a}}$ based on $\tilde{f}_t$ in (104) for the remainder of this section, instead of the global definition from (17). Note that this optimization is not tractable in the noisy case.

For the exploration scheme in (104) we have the following results that lead to improved regret bounds that match those in (Chowdhury and Gopalan, 2019) up to constant factors.

**Lemma 19.** *Under the assumptions of Corollary 4, let $\bar{L}_f = L_f$. Then, for any sequence of $\boldsymbol{\eta}_n \in [-1,1]^p$, any sequence of $\boldsymbol{\omega}_n$ with $\tilde{\boldsymbol{\omega}}_n = \boldsymbol{\omega}_n$, $\boldsymbol{\theta} \in \mathcal{D}$, and $n \geq 1$ we have that*

$$\|\mathbf{s}_{n,t} - \tilde{\mathbf{s}}_{n,t}\| \leq 2\beta_{t-1}\bar{L}_f^{N-1} \sum_{i=0}^{n-1} \|\boldsymbol{\sigma}_{t-1}(\mathbf{s}_{i,t})\| \tag{105}$$

*Proof.* Let

$$\tilde{f}(\tilde{\mathbf{s}}_{n,t}) = \boldsymbol{\mu}_{t-1}(\tilde{\mathbf{s}}_n) + \beta_{t-1}\boldsymbol{\Sigma}_{t-1}(\tilde{\mathbf{s}}_n)\boldsymbol{\eta}_n. \tag{106}$$

Then by design we have $\|\tilde{f}(\mathbf{s}) - \tilde{f}(\mathbf{s}')\| \leq L_f \|\mathbf{s} - \mathbf{s}'\|$.

We start by showing that, for any $n \geq 1$, we have

$$\|\mathbf{s}_{n,t} - \tilde{\mathbf{s}}_{n,t}\| \leq 2\beta_{t-1} \sum_{i=0}^{n-1} L_f^{n-1-i} \|\boldsymbol{\sigma}_{t-1}(\mathbf{s}_{i,t})\| \tag{107}$$

by induction.

For the base case we have $\tilde{\mathbf{s}}_0 = \mathbf{s}_0$. Consequently, at $t$ we have

$$\|\mathbf{s}_{1,t} - \tilde{\mathbf{s}}_{1,t}\| = \|f(\mathbf{s}_0) + \boldsymbol{\omega}_0 - \tilde{f}(\mathbf{s}_0) - \tilde{\boldsymbol{\omega}}_0\| \tag{108}$$

$$= \|f(\mathbf{s}_0) - \tilde{f}(\mathbf{s}_0)\| \tag{109}$$

$$= \|f(\mathbf{s}_0) - \boldsymbol{\mu}_{t-1}(\mathbf{s}_0) - \beta_{t-1}\boldsymbol{\Sigma}_{t-1}(\mathbf{s}_0)\boldsymbol{\eta}_0\| \tag{110}$$

$$\leq \|f(\mathbf{s}_0) - \boldsymbol{\mu}_{t-1}(\mathbf{s}_0)\| + \beta_{t-1}\|\boldsymbol{\sigma}_{t-1}(\mathbf{s}_0)\boldsymbol{\eta}_0\| \tag{111}$$

$$\leq \beta_{t-1}\|\boldsymbol{\sigma}_{t-1}(\mathbf{s}_0)\| + \beta_{t-1}\|\boldsymbol{\sigma}_{t-1}(\mathbf{s}_0)\| \tag{112}$$

$$= 2\beta_{t-1}\|\boldsymbol{\sigma}_{t-1}(\mathbf{s}_0)\| \tag{113}$$

For the induction step assume that (107) holds at time step $n$. Subsequently we have at iteration $t$ that

$$\|\mathbf{s}_{n+1,t} - \tilde{\mathbf{s}}_{n+1,t}\| = \|f(\mathbf{s}_n) - \tilde{f}(\tilde{\mathbf{s}}_n)\|$$

$$= \|f(\mathbf{s}_n) - \tilde{f}(\mathbf{s}_n) + \tilde{f}(\mathbf{s}_n) - \tilde{f}(\tilde{\mathbf{s}}_n)\|$$

$$= \|f(\mathbf{s}_n) - \tilde{f}(\mathbf{s}_n)\| + \|\tilde{f}(\mathbf{s}_n) - \tilde{f}(\tilde{\mathbf{s}}_n)\|$$

$$\leq 2\beta_{t-1}\|\boldsymbol{\sigma}_{t-1}(\mathbf{s}_n)\| + L_f\|\mathbf{s}_n - \tilde{\mathbf{s}}_n\|$$

$$\leq 2\beta_{t-1}\|\boldsymbol{\sigma}_{t-1}(\mathbf{s}_n)\| + L_f 2\beta_{t-1}\sum_{i=0}^{n-1} L_f^{n-1-i}\|\boldsymbol{\sigma}_{t-1}(\mathbf{s}_{i,t})\|$$

$$= 2\beta_{t-1}\|\boldsymbol{\sigma}_{t-1}(\mathbf{s}_n)\| + 2\beta_{t-1}\sum_{i=0}^{n-1} L_f^{n-1-i+1}\|\boldsymbol{\sigma}_{t-1}(\mathbf{s}_{i,t})\|$$

$$= 2\beta_{t-1}\sum_{i=0}^{(n+1)-1} L_f^{(n+1)-1-i+1}\|\boldsymbol{\sigma}_{t-1}(\mathbf{s}_{i,t})\|$$

$$= 2\beta_{t-1}\sum_{i=0}^{(n+1)-1} L_f^{(n+1)-i}\|\boldsymbol{\sigma}_{t-1}(\mathbf{s}_{i,t})\|$$

Thus (107) holds. Now since $n \leq N$ we have

$$\|\mathbf{s}_{n+1,t} - \tilde{\mathbf{s}}_{n+1,t}\| \leq 2\beta_{t-1}\sum_{i=0}^{n-1} L_f^{n-1-i}\|\boldsymbol{\sigma}_{t-1}(\mathbf{s}_{i,t})\| \quad \leq 2\beta_{t-1}L_f^{N-1}\sum_{i=0}^{n-1}\|\boldsymbol{\sigma}_{t-1}(\mathbf{s}_{i,t})\| \tag{114}$$

$\square$

**Theorem 4.** *Under Assumptions 1–3 let $\mathbf{s}_{n,t} \in \mathcal{X}_t$, $\mathcal{S}_{t-1} \subseteq \mathcal{S}_t$, and $\mathbf{a}_{n,t} \in \mathcal{U}$ for all $n, t > 0$ with compact sets $\mathcal{S}_t$ and $\mathcal{A}$. Let $\|\boldsymbol{\sigma}(\cdot)\|_\infty \leq \sigma_{\max}$. At each iteration, select parameters according to (104). Then the following holds with probability at least $(1-\delta)$ for all $t \geq 1$*

$$R_T \leq \mathcal{O}\left(L_f^N N^2 \sqrt{T\, p\, \gamma_{pTN}(\mathcal{S}_T \times \mathcal{A})}\right), \tag{115}$$

*where $\gamma_{pTN}(\mathcal{S} \times \mathcal{A})$ is the information capacity after $(ptN)$ observations within the domain $\mathcal{S} \times \mathcal{A}$.*

Thus, our proof strategy also avoids the scaling $\beta^N$ when we assume that optimizing over dynamics in $\mathcal{M}$ is tractable. Thus, the factor $\beta^N$ is the cost that we pay for not being able to do so.

# I   Extension to Unbounded Domains

So far, we have assumed a compact domain $\mathcal{S}$. This is incompatible with the dynamic system in (1), since sub-Gaussian noise includes noise distributions with unbounded support. In this section, we show that we can bound the domain with high probability and that we can use continuity arguments to extend our previous theorem to this more general settings. This also avoids the implicit assumption that the dynamics function is bounded, which is not even true for linear systems.

## I.1  Bound on Aleatoric Uncertainty (Noise Bound)

We start by bounding the norm of the noise vector $\boldsymbol{\omega}_n$ over all time steps $n$.

We know that the $\boldsymbol{\omega}_n$ are i.i.d. sub-Gaussian vectors. We exploit the basic properties of sub-Gaussian random variables and refer to Eldar and Kutyniok (2012, Chapter 5) for a concise review.

**Lemma 20.** *Vershynin (2010, Corollary 5.17) Let $X_1, \ldots, X_p$ be independent centered sub-exponential random variables, and let $2\sigma = \max_i \|X_i\|_{\phi_1}$ be the largest, sub-exponential norm. Then, for every $\epsilon \geq 0$, we have*

$$\mathbb{P}\left\{ \left| \sum_{i=1}^{T} X_i \right| \geq \epsilon p \right\} \leq 2\exp\left[ \frac{-\mathrm{e}T}{2} \min\left( \frac{\epsilon^2}{4\sigma^2}, \frac{\epsilon}{2\sigma} \right) \right] \tag{116}$$

This allows us to bound the 2-norm of the noise vectors in (1).

**Lemma 21.** *Let $\boldsymbol{\omega} = (\omega_1, \ldots, \omega_p)$ be a vector with i.i.d. elements $[\boldsymbol{\omega}]_i = \omega_i$ that are $\sigma$-sub-Gaussian. Then, with probability at least $1 - \delta$, we have that*

$$\|\boldsymbol{\omega}\|_2^2 \leq 2\sigma p + \frac{4\sigma}{\mathrm{e}} \log \frac{2}{\delta} \tag{117}$$

*Proof.* Since the $\omega_i$ are $\sigma$-sub-Gaussian, we have the $\omega_i^2$ are $2\sigma$-sub-exponential (Vershynin, 2010, Lemma 5.14). Thus we have

$$\|\boldsymbol{\omega}\|_2^2 = \sum_{i=1}^{p} \omega_i^2,$$

where the $\omega_i^2$ are i.i.d. $2\sigma$-sub-exponential. Following Lemma 20, we have

$$\mathbb{P}\left\{ \|\boldsymbol{\omega}\|_2^2 \geq \epsilon p \right\} \leq 2\exp\left[ \frac{-\mathrm{e}p}{2} \min\left( \frac{\epsilon^2}{4\sigma^2}, \frac{\epsilon}{2\sigma} \right) \right] \tag{118}$$

Now for $\epsilon \geq 2\sigma$ we have $\epsilon^2/(4\sigma^2) \geq \epsilon/(2\sigma)$. Thus

$$\mathbb{P}\left\{ \|\boldsymbol{\omega}\|_2^2 \geq (2\sigma + \epsilon)p \right\} \leq 2\exp\left[ \frac{-\mathrm{e}p}{2} \frac{(2\sigma + \epsilon)}{2\sigma} \right] \leq 2\exp\left[ \frac{-\mathrm{e}p}{2} \frac{\epsilon}{2\sigma} \right] \tag{119}$$

We want to upper bound the right hand side by $\delta$. so

$$2\exp\left[ \frac{-\mathrm{e}p\epsilon}{4\sigma} \right] \leq \delta, \tag{120}$$

$$\frac{-\mathrm{e}p\epsilon}{4\sigma} \leq \log(\delta/2), \tag{121}$$

$$\frac{\mathrm{e}p\epsilon}{4\sigma} \geq \log(2/\delta), \tag{122}$$

$$\epsilon \geq \frac{4\sigma}{\mathrm{e}p} \log(2/\delta). \tag{123}$$

the result follows by plugging the bound for $\epsilon$ into (119),

$$(2\sigma + \epsilon)p = (2\sigma + \frac{4\sigma}{\mathrm{e}p} \log(2/\delta))p \tag{124}$$

$$= 2\sigma p + \frac{4\sigma}{\mathrm{e}} \log \frac{2}{\delta} \tag{125}$$

$\square$

As the last step, we apply the union bound to obtain confidence intervals over multiple steps.

**Lemma 22.** *Let $\boldsymbol{\omega}_0, \boldsymbol{\omega}_1, \ldots$ be i.i.d. random vectors with $\boldsymbol{\omega}_n \in \mathbb{R}^p$ such that each entry of the vector is i.i.d. $\sigma$-sub-Gaussian. Then, with probability at least $(1 - \delta)$,*

$$\|\boldsymbol{\omega}_n\|_2^2 \leq 2\sigma p + \frac{4\sigma}{\mathrm{e}} \log \frac{(n+1)^2 \pi^2}{3\delta} \tag{126}$$

*holds jointly for all $n \geq 0$.*

*Proof.* At each time step $n$, we apply a probability budget of $\delta/\pi_n$ to the bound in Lemma 21, where $\pi_n \geq 0$ and $\sum_{n \geq 0} \pi_n^{-1} = 1$. In particular, we use $\pi_n = \frac{(n+1)^2 \pi^2}{6}$ as in (Srinivas et al., 2012, Lemma 5.1), so that we apply monotonically decreasing probability thresholds as $n$ increases. We obtain the result by applying a union bound over $n$, since $\sum_{n \geq 0} \delta/\pi_n = \delta$. $\qquad\square$

This means that, for all time steps $n$, the noise is bounded within the hyper-sphere defined through (126) with high probability. In particular, the joint confidence intervals only come at the cost of a $\mathcal{O}(\log n^2)$ increase in the confidence intervals over time.

## I.2 Bounding the Domain Under Aleatoric Uncertainty

We exploit the $\sigma$-sub-Gaussian property of the transition noise and build on Lemmas 20 and 21 to obtain a bound over the domain. We start by applying a union bound on Lemma 21 over the time horizon $N$.

**Lemma 23.** *Let $\boldsymbol{\omega}_0, \ldots, \boldsymbol{\omega}_{N-1}$ be vectors with $\boldsymbol{\omega}_i \in \mathbb{R}^p$ such that each entry of the vector is i.i.d. $\sigma$-sub-Gaussian. Then, with probability at least $(1 - \delta)$,*

$$\sum_{n=0}^{N-1} \|\boldsymbol{\omega}_i\|_2 \leq N \sqrt{2\sigma p + \frac{4\sigma}{\mathrm{e}} \log \frac{2T}{\delta}} \tag{127}$$

*Proof.* Now using Lemma 21 with probability threshold $\delta/T$ and applying the union bound we, get that $\|\boldsymbol{\omega}_i\|_2^2 \leq 2\sigma p + \frac{4\sigma}{\mathrm{e}} \log \frac{2T}{\delta}$ holds for all $0 \leq i \leq N-1$ with probability at least $1 - \delta$.

Now, first using Jensen's inequality and then plugging in the bound for $\|\boldsymbol{\omega}_i\|_2^2$, we obtain

$$\sum_{n=1}^{N} \|\boldsymbol{\omega}_n\|_2 = \sum_{i=0}^{N-1} \sqrt{\|\boldsymbol{\omega}_n\|_2^2} \tag{128}$$

$$\leq \sqrt{T} \sqrt{\sum_{n=0}^{N-1} \|\boldsymbol{\omega}_n\|_2^2} \tag{129}$$

$$\leq \sqrt{T} \sqrt{\sum_{n=0}^{N-1} \left( 2\sigma p + \frac{4\sigma}{\mathrm{e}} \log \frac{2T}{\delta} \right)} \tag{130}$$

$$= N \sqrt{2\sigma p + \frac{4\sigma}{\mathrm{e}} \log \frac{2T}{\delta}} \tag{131}$$

$$\square$$

Lastly, we use a union bound over all iterations similar to (Srinivas et al., 2012, Lemma 5.1).

**Lemma 24.** *Let $\boldsymbol{\omega}_{t,n}$ be the random vectors as in Lemma 23 at iteration $n$. Then, with probability $(1 - \delta)$ we have for all $n \geq 1$ that*

$$\sum_{t=1}^{N} \|\boldsymbol{\omega}_{n,t}\|_2 \leq N \sqrt{2\sigma p + \frac{4\sigma}{\mathrm{e}} \log \frac{N \pi^2 t^2}{3\delta}} \tag{132}$$

*Proof.* At each iteration $n$, we apply a probability budget of $\delta/\rho_t$ to the bound in Lemma 23, where $\rho_t \geq 0$ and $\sum_{t \geq 1} \rho_t^{-1} = 1$. In particular, we use $\rho_t = \frac{t^2 \pi^2}{6}$ as in (Srinivas et al., 2012, Lemma 5.1), so that we apply monotonically decreasing probability thresholds as $t$ increases. We obtain the result by applying a union bound over $t$, since $\sum_{t \geq 1} \delta/\rho_t = \delta$. $\qquad\square$

Now that we can bound the noise over all iterations, we can bound the domain over which the system acts with a compact set.

**Lemma 25.** *Let $f$ be $L_f$-Lipschitz continuous with respect to the norm $\|\cdot\|$. Then we have for all $n \geq 1$ that*

$$\|\mathbf{s}_n - \mathbf{s}_0\| \leq \sum_{i=0}^{n-1} L_{\text{fc}}^i \|f(\mathbf{s}_0) - \mathbf{s}_0\| + \sum_{i=0}^{n-1} L_{\text{fc}}^{n-1-i} \|\boldsymbol{\omega}_i\| \tag{133}$$

$$\leq (1 + L_{\text{fc}})^{n-1} \left( n\|f(\mathbf{s}_0) - \mathbf{s}_0\| + \sum_{i=0}^{n-1} \|\boldsymbol{\omega}_i\| \right) \tag{134}$$

*Proof.* We first proof (133) by induction. For the base case we have

$$\|\mathbf{s}_1 - \mathbf{s}_0\| = \|f(\mathbf{s}_0) + \boldsymbol{\omega}_0 - \mathbf{s}_0\| \tag{135}$$

$$\leq \|f(\mathbf{s}_0) - \mathbf{s}_0\| + \|\boldsymbol{\omega}_0\|, \tag{136}$$

$$= L_{\text{fc}}^0 \|f(\mathbf{s}_0) - \mathbf{s}_0\| + L_{\text{fc}}^0 \|\boldsymbol{\omega}_0\|. \tag{137}$$

For the induction step, assume that the assumption holds for some $n$. Then,

$$\|\mathbf{s}_{t+1} - \mathbf{s}_0\| = \|f(\mathbf{s}_n) + \boldsymbol{\omega}_n - \mathbf{s}_0\| \tag{138}$$

$$= \|f(\mathbf{s}_n) - f(\mathbf{s}_0) + f(\mathbf{s}_0) - \mathbf{s}_0 + \boldsymbol{\omega}_n\| \tag{139}$$

$$\leq \|f(\mathbf{s}_n) - f(\mathbf{s}_0)\| + \|f(\mathbf{s}_0) - \mathbf{s}_0\| + \|\boldsymbol{\omega}_n\| \tag{140}$$

$$\leq L_{\text{fc}} \|\mathbf{s}_n - \mathbf{s}_0\| + \|f(\mathbf{s}_0) - \mathbf{s}_0\| + \|\boldsymbol{\omega}_n\| \tag{141}$$

$$\leq L_{\text{fc}} \left( \sum_{i=0}^{n-1} L_{\text{fc}}^i \|f(\mathbf{s}_0) - \mathbf{s}_0\| + \sum_{i=0}^{n-1} L_{\text{fc}}^{n-1-i} \|\boldsymbol{\omega}_i\| \right) \tag{142}$$

$$+ \|f(\mathbf{s}_0) - \mathbf{s}_0\| + \|\boldsymbol{\omega}_n\| \tag{143}$$

$$= \sum_{i=1}^{(t-1)+1} L_{\text{fc}}^i \|f(\mathbf{s}_0) - \mathbf{s}_0\| + \|f(\mathbf{s}_0) - \mathbf{s}_0\|$$

$$+ \sum_{i=0}^{n-1} L_{\text{fc}}^{(t+1)-1-i} \|\boldsymbol{\omega}_i\| + \|\boldsymbol{\omega}_n\| \tag{144}$$

$$= \sum_{i=0}^{(t-1)+1} L_{\text{fc}}^i \|f(\mathbf{s}_0) - \mathbf{s}_0\| + \sum_{i=0}^{(t+1)-1} L_{\text{fc}}^{(t+1)-1-i} \|\boldsymbol{\omega}_i\| \tag{145}$$

Which concludes the proof. For (134), note that $L_{\text{fc}}^i \leq (1 + L_{\text{fc}})^t$ for all $i \leq t$. Thus we have

$$\sum_{i=0}^{n-1} L_{\text{fc}}^i \|f(\mathbf{s}_0) - \mathbf{s}_0\| + \sum_{i=0}^{n-1} L_{\text{fc}}^{n-1-i} \|\boldsymbol{\omega}_i\| \tag{146}$$

$$\leq L_{\text{fc}}^{n-1} \sum_{i=0}^{n-1} \left( \|f(\mathbf{s}_0) - \mathbf{s}_0\| + \|\boldsymbol{\omega}_i\| \right) \tag{147}$$

$$= L_{\text{fc}}^{n-1} \left( n\|f(\mathbf{s}_0) - \mathbf{s}_0\| + \sum_{i=0}^{n-1} \|\boldsymbol{\omega}_i\| \right) \tag{148}$$

$$\square$$

**Lemma 26.** *Let $b_t = L_{\text{fc}}^{T-1} N \left( B_0 + \sqrt{2\sigma p + \frac{4\sigma}{e} \log \frac{N\pi^2 n^2}{3\delta}} \right)$ and $\|f(\mathbf{s}_0) - \mathbf{s}_0\|_2 \leq B_0$. Then, with probability at least $(1 - \delta)$, we have for all iterations $n \geq 1$ and corresponding time steps $0 \leq n \leq N$ that*

$$\mathbf{s}_{n,t} \in \mathbb{B}(\mathbf{s}_0, b_t), \tag{149}$$

*where $\mathbb{B}(\mathbf{s}_0, b_t) = \{\mathbf{s} \in \mathbb{R}^p \mid \|\mathbf{s} - \mathbf{s}_0\|_2 \leq b_t\}$ is a norm-ball centered around $\mathbf{s}_0$ with radius $b_t$.*

*Proof.* From Lemma 25, we have for all $n \geq 1$, $0 \leq n \leq N$ that

$$\|\mathbf{s}_{t,n} - \mathbf{s}_0\|_2 \leq (1 + L_{\text{fc}})^{n-1} \left( n\|f(\mathbf{s}_0) - \mathbf{s}_0\|_2 + \sum_{i=0}^{n-1} \|\boldsymbol{\omega}_i\|_2 \right) \tag{150}$$

Now by Assumptions 1 and 3 and Combined with Lemma 24, we obtain

$$\|\mathbf{s}_{t,n} - \mathbf{s}_0\|_2 \leq (1 + L_{\text{fc}})^{n-1} \left( n\|f(\mathbf{s}_0) - \mathbf{s}_0\|_2 + n\sqrt{2\sigma p + \frac{4\sigma}{e} \log \frac{t\pi^2 n^2}{3\delta}} \right) \tag{151}$$

$$\leq (1 + L_{\text{fc}})^{T-1} N \left( \|f(\mathbf{s}_0) - \mathbf{s}_0\|_2 + \sqrt{2\sigma p + \frac{4\sigma}{e} \log \frac{N\pi^2 n^2}{3\delta}} \right) \tag{152}$$

$$:= b_t \tag{153}$$

Lastly, we have $\|f(\mathbf{s}_0) - \mathbf{s}_0\|_2 \leq B_0$ by assumption, which concludes the proof. $\qquad\square$

### I.3   Regret bounds over Unbounded Domains

The probability for the noise bound is generally different from the one used for the well-calibrated model. We can derive a joint bound using a simple union bound.

**Lemma 27.** *Under Assumptions 1–3, let* $\|f(\mathbf{s}_0) - \mathbf{s}_0\|_2 \leq B_0$ *and define* $b_t = L_{\text{fc}}^{T-1} N \left( B_0 + \sqrt{2\sigma p + \frac{4\sigma}{e} \log \frac{N\pi^2 n^2}{3\delta}} \right)$. *Then the following hold jointly with probability at least* $(1 - 2\delta)$ *for all* $t \geq 1$ *and* $0 \leq n < N$

    *i)* $|f(\mathbf{s}, \mathbf{a}) - \boldsymbol{\mu}_t(\mathbf{s}, \mathbf{a})| \leq \beta_t \boldsymbol{\sigma}_t(\mathbf{s}, \mathbf{a})$ *elementwise for all* $\mathbf{s} \in \mathbb{R}^p$ *and* $\mathbf{a} \in \mathbb{R}^q$

    *ii)* $\mathbf{s}_{n,t} \in \mathbb{B}(\mathbf{s}_0, b_t)$

*Proof.* This follows directly from applying a union bound over Lemma 26 and Corollary 5 with a probability budget of $\delta/2$ for each. $\qquad\square$

Note that the probability dropped from individual confidences of $1 - \delta$ in Assumption 2 and Lemma 26 to a joint confidence of $1 - 2\delta$.

Thus, we can used Lemma 27 together with Corollary 6 to fulfill both the compact set and the boundedness requirements. The last assumption we need is boundedness of the predictions. For this, we introduce an additional weak assumptions

**Assumption 6** (Boundedness). The system dynamics at the first step are bounded, $\|f(\mathbf{s}_0) - \mathbf{s}_0\|_2 \leq B_0$. Similarly we have $\boldsymbol{\Sigma}(\mathbf{s}_0)$ and, if used, $k(\mathbf{s}_0, \mathbf{s}_0)$ bounded.

These assumptions are not restrictive, since any dynamical system that explodes to infinity after one step is generally not real-world relevant or controllable. Similarly, we cannot expect to do learning if our model's confidence intervals allow infinite predictions.

**Corollary 6.** *Under Assumptions 3 and 6, if the states live in a compact set* $\mathcal{S}_t$, *then* $\boldsymbol{\sigma}(\mathbf{s})$ *is bounded.*

*Proof.* This follows trivially from Assumption 3, since $\mathbf{s}_0 \in \mathcal{S}$ and $\boldsymbol{\sigma}(\mathbf{s}_0)$ is bounded. Thus, by continuity, it must be bounded over a compact set. $\qquad\square$

**Theorem 5.** *Under Assumptions 1–3 let the noise distribution be* $\sigma$-*subGaussian as in Assumption 1 and* $\pi_\theta(\mathbf{s}) \in \mathcal{A}$ *for all* $\pi \in \Pi$ *with* $\mathcal{A}$ *compact. At each iteration, select parameters according to* (7). *Then the following holds with probability at least* $(1 - 2\delta)$ *for all* $T \geq 1$

$$R_T \leq \mathcal{O}\left( \beta_{T-1}^N L_\sigma^N N^2 \sqrt{T\, p\, \gamma_{pTN}(\mathbb{B}(\mathbf{s}_0, b_t) \times \mathcal{A} \times \mathcal{I}_p)} \right), \tag{154}$$

*where* $b_t = L_{\text{fc}}^{T-1} N \left( B_0 + \sqrt{2\sigma p + \frac{4\sigma}{e} \log \frac{N\pi^2 n^2}{3\delta}} \right)$.

*Proof.* By Assumption 1 we know from Lemma 27 that with probability at least $(1 - 2\delta)$ the model is well-calibrated and $\mathbf{s} \in \mathcal{S}_t = \mathbb{B}(\mathbf{s}_0, b_t)$. Boundedness of predictions follows from Corollary 6, so that all requirements of Theorem 1 are satisfied and the result follows. $\qquad\square$

### I.4 Bounding the Maximum Information Capacity for Gaussian Processes

In Theorem 5 the information capacity is a function of the domain size. Given the previous proofs, the radius of the domain increases at a logarithmic rate $b_t \in \mathcal{O}(\log t^2)$, which also increases the information capacity. In the following two lemmas, we show how this affects the information capacity of the Gaussian process model.

**Lemma 28** (Srinivas et al. (2012)). *For the linear kernel $k(\mathbf{s}, \mathbf{s}') = \mathbf{s}^{\mathrm{T}} \mathbf{s}'$ with $\mathbf{s} \in \mathbb{R}^p$ we have*

$$\gamma_t(\mathbb{B}(\mathbf{s}_0, b_t)) = \mathcal{O}(p \log(t)) \tag{155}$$

**Lemma 29.** *For the squared exponential kernel we have*

$$\gamma_t(\mathbb{B}(\mathbf{s}_0, b_t)) = \mathcal{O}\big(b_t^p (\log(t))^{p+1}\big) \tag{156}$$

*Proof.* The proof is the same as in (Srinivas et al., 2012). In their notation, we have $n_T = \mathcal{O}\big(b_t^d \log(b_t^d)\big)$ while analyzing the terms in the eigenvalue bound leads to $B_k(T^*) \sim b_t^d$. The remainder of the proof follows through as in the original paper, which leads to the result. $\qquad \square$

Thus, the information capacity grows proportionally to the volume of the domain. Since $b_t$ in Theorem 5 is $\mathcal{O}(\log t^2)$ this means that this costs us only an additional logarithmic factor in the regret relative to a fixed domain $\mathcal{S}$.

Note that we are using a composite kernel to model the different output dimensions. Thus these bounds need to be combined with the methodology from Krause and Ong (2011) in order to obtain bounds for the composite kernels. However, this does not affect the result.