[Reviews · NeurIPS 2020]

Review 1

Summary and Contributions: Paper is concerned with model-based RL and exploration in continuous-state MDPs. Presents a way to fold epistemic uncertainty over the model into an optimal-action-selection formulation, with both theoretical regret bounds and empirical results on a range of standard control problems.

Strengths: The focus on MBRL is important, as model-free methods seem to run out of steam by requiring far too many trials. The analytical results and experiments seem solid and convincing. The paper is very well written with a judicious mix of rigor and insight.

Weaknesses: Could perhaps be considered a little bit incremental, as a combination of existing ideas (Jaksch et al for UCRL, Moldovan for hallucination, various authors for proofs).

Correctness: I have not checked all the proofs, but the analysis seems to be careful and demonstrates an impressive command of the theoretical techniques developed by various authors working in this area.

Clarity: Mostly very clear and accurate. One key area for improvement is in the introduction (l.36). The core explanation in the introduction is "enlarging the control space of the agent with hallucinated control inputs besides the real control inputs. These hallucinated inputs are bounded by the agent’s epistemic uncertainty about the transition dynamics". I'm afraid this does not help the reader at all. What are these actions that are not real? What does it mean to bound an action by uncertainty? Once I have read section 3.1 it becomes a bit clearer, but it's essential to be able to explain things in order! l.299 greedy exploitation struggles, -> greedy exploitation struggles: l.303 Thompson Sampling -> Thompson sampling l.304 higher dimensional -> higher-dimensional l.314 "the action has a non-negative signal when it is close to zero" - rewrite, I'm not sure what is being said here. l.332 medical decisions making etc. -> medical decision making, etc. Quite a few errors in the references, mainly capitalization of book titles, journals, publishers: eg "cambridge university press".

Relation to Prior Work: Generally excellent and precise discussion of related work. Might be worth noting connections to early papers on Bayesian RL (Dearden et al.)

Reproducibility: Yes

Additional Feedback:


Review 2

Summary and Contributions: Conventionally when rollout-based MBRL algorithms apply an optimistic exploration strategy like UCB, aleatoric and epistemic uncertainty are often conflated into a single pointwise measure of uncertainty at each state in the rollout sequence. This submission proposes a novel augmented policy class that explicitly interacts with the model’s epistemic uncertainty to hypothesize the best possible outcome for any particular action sequence. In addition to proof-of-concept experiments on easy Mujoco control tasks, the authors provide regret bounds for their exploration strategy applied to purely rollout-based MBRL methods, including a sublinear regret bound for GP dynamics models. My greatest concern with this submission lies with the reproducibility of the results. There is no mention of code, and simple, crucial implementation details are missing. In my judgement it would be very difficult to reproduce this submission based solely on the paper in its current form. The theoretical analysis is interesting, but (as is usually the case in MBRL) there are a couple gaps between the theory and praxis of the submission. The empirical evaluation is commendably thorough in some aspects and unfortunately lacking in others. For these reasons I am assigning this paper a borderline accept score. --- UPDATE 08-17-2020 --- Thanks for the clarifications in the response. I feel that you have addressed my main concerns, and will update my score accordingly. It is worth noting that randomly sampled initial conditions are _not_ equivalent to a stochastic environment. As you are doubtless aware, in the standard episodic MDP formulation there is a clear distinction between the initial state distribution and the transition dynamics. If the environment dynamics are not truly Markovian then the dependence on initial conditions could be modeled as aleatoric noise. I don't know if that is what you were getting at in your response. Perhaps a brief discussion on the sources of randomness you consider could help clarify things for me on this point. Given the reviewing format I have to take you at your word that you will provide more practical implementation details in your camera-ready, but based on the attention to detail in the current manuscript I feel comfortable doing so.

Strengths: The exploration strategy is well-motivated. To my knowledge this submission is one of the first practical optimistic exploration strategies for MBRL that does not conflate aleatoric and epistemic uncertainty. The authors should be commended for contributing to the effort to understand the theoretical properties of MBRL. The relevance and potential impact of the idea is significant, since the most popular exploration strategies in continous control are little better than random search (e.g. SAC).

Weaknesses: Theoretical Grounding: - Since the regret bound relies on bounding the state reconstruction error at each point in a finite-length sequence, it seems clear to me that the bound only applies to MBRL algorithms that rely exclusively on full-episode rollouts for planning (e.g. PILCO). This criticism in particular seems to apply to the practical algorithm used in the submission, since it relies on a parametric value function approximator to make the planning horizon more tractable. - In practice we don't know \beta _a priori_. How should \beta be chosen? Empirical Evaluation: - The method is not evaluated in a stochastic environment. If you have set out to solve the issue of conflation of epistemic and aleatoric uncertainty, you should evaluate your method in a stochastic environment, not deterministic Mujoco environments. As it is the experiments give the impression of a bait-and-switch. - No comparison to competing methods only an ablation of the proposed exploration strategy with greedy improvement and approximate Thompson Sampling. - No ablations of β (presumably a crucial design choice). - No demonstration that their dynamics models as implemented satisfy their calibration assumption. Significance/Impact: - Information on the specific implementation details is fairly sparse (e.g. what learning rate and batch size did you use for your dynamics models? Did you reuse a previous implementation of MPO?). Reproducing the authors' findings would likely prove very difficult. MBRL is notorious in the machine learning field for reproducibility issues. If you (the authors) had to reimplement your method from scratch tomorrow, what details would you need?

Correctness: Empirical Methodology: - It is impossible to assess the sample-efficiency and general characteristics of the proposed method, since the main experimental result (Figure 3) only reports the final performance. What is the shape of the reward curve? Do your agents converge to good policies?

Clarity: The paper is well-written and easy to understand. Needs more detail about a practical implementation. There are also some points of redundancy between the main text and the appendix (e.g. Section 3.1 and Appendix C have significant overlap).

Relation to Prior Work: The authors adequately position the submission in relation to the literature, demonstrating a thorough knowledge of prior work. The submission could be better placed in context if the experimental results included comparisons to previous results (e.g. SAC, PETS, POPLIN, POLO).

Reproducibility: No

Additional Feedback:


Review 3

Summary and Contributions: This paper studies optimistic exploration in model-based RL with probabilistic dynamics models. Standard MBRL approaches in this field average out over all (epistemic) uncertainty when they plan over the learned model. The authors argue that we should rather be optimistic with respect to this uncertainty. It is infeasible to solve for the optimal path while propagating all uncertainty over timesteps. Instead, they propose a variant in which we optimize within the pointwise confidence interval at every timestep. They do so by introducing a set of auxiliary action variables, which capture the 1-step confidence interval at every timestep. They then apply standard RL techniques to this augmented action space. The authors provide both theoretical and empirical evaluation of their approach. ----- Update after authors rebuttal ------: Thanks for your feedback, it did give me better insight and improved my rating of the paper. Some suggestions that still stand in my opinion: - I had read L190-201 as a summary of the possible options to solve for \pi and \eta. I now see you mention in L213-214 that you actually use MPO. Maybe you could explicitly make your choice a separate paragraph, now it sort of hides inside the Dyna-MPC section, and I could not find it back. - The ability to deal with stochastic environment is quite crucial for your probabilitistic method. Random starts (as you mention in your rebuttal) are not a form of stochastic transition dynamics, and cannot test robustness against this property. In any case, I would more clearly discuss this topic, and indicate that one environment does have additive Gaussian noise. - Regarding the action penalties, I understand that you use the same set-up as PETS, but it would have been preferable if you left the magnitude of the action penalties unchanged. Since performance so crucially relied on the size of the action penalty, it now seems as if your method really needs the correct action penalty size to work. - I did not understand your comment on the generalization in \pi and \eta space. So you do learn a neural network that directly predicts in \eta space? My main concern was that learning of \eta will be unstable, especially when \Sigma changes fast. When you mention "restarting between episodes", do you mean that you for example keep \pi, but reset \eta between episodes? (that would loose much information) Or that you reset the \Sigma estimate in the new episode? (that makes sense, but a quite change in \Sigma will make your \eta estimates of?) Maybe I miss something here, but I think this could be explained more clearly.

Strengths: - The paper is well-written and deals with a problem that has received much previous attention in RL literature (provably efficient exploration). - The algorithm they eventually evaluate is essentially model predictive control where we also optimize at every step for the 1SD confidence interval of the epistemic uncertainty. This idea is interesting, and according to the authors has not been tried before.

Weaknesses: – My main worry is whether this approach scales to larger problems, especially when you would learn a parametrized policy network. We would then have to generalize in \eta space. A second problem occurs here, since the underlying \Sigma estimates may change fast, which introduces extra instability. Maybe you only intend to optimize over the \eta variables in a model predictive control setting, where you locally optimize them. But then you should not mention in Sec 3.1 that we can also apply policy search methods, like TRPO and PPO, which use global policy network approximations. In any case, this requires further discussion. – You mention the possible presence of stochasticity (aleatoric uncertainty) when you introduce the problem, but then ignore it later on. What happens to your method if the next state distribution is multi-modal? The uncertainty around the mean will be off in case of multimodality, where the mean may not even be a feasible next state at all. I think the paper would profit if you comment on this. – L209-218: I don’t think “Dyna-MPC” is new. What you describe sounds like Monte Carlo planning in a model, where you bootstrap with a learned value function at a certain depth. The only resemblance with Dyna is that you plan in a learned model, but this already happens in you other approaches as well (e.g., L202-208). So the term ‘Dyna’ is misleading. The real difference with standard MPC is that you now bootstrap, but bootstrapping happens all over RL literature, and is not a particular property of Dyna. – L231: “Discontinuous reward function are generally very difficult to optimize”. I disagree. At least it sounds as if these problems are not solvable, while a large part of RL literature deals with discontinuous reward functions. You for example mention Atari games in your own related work section, which have discontinuous reward functions. I would rather say you choose to focus on continuous MDPs with smooth reward functions. – I would prefer some extra visualization of results in the paper, like learning curves. I later on found out you did extensively include these in the appendix. I would more clearly mention in the paper that we can find more detailed visualization in the appendix. – There is hardly any discussion of your results. The paper would benefit from some remarks about the limitations of your algorithms, and possible future work. Minor: – Figure 1: I would change the legend to also have ordering red-green-blue. Also, I printed the paper without color, and then could not discriminate the different bars. The ordering of your legend therefore made me extra confused. – You never define the abbreviation UCRL.

Correctness: - I am not familiar with the theoretical derivations in this subfield. As far as I can judge, the theoretical derivations seem sound. - The empirical results are clear, although not very extensive. They mostly focus on the setting where there are action penalties, which is relevant setting, but only one of their environments (Half-Cheetah) does not have these penalties.

Clarity: - Yes the paper is clearly written. - I would have preferred to see some learning curves in the main paper (they are now in the appendix). The provide more information than the current plots.

Relation to Prior Work: - Related work is well-covered, and I like the systematic discussion of exploration approaches in Sec. 2.2. - I find it hard to fully assess the size of the contribution of this paper. The idea to be optimistic with respect to the epistemic uncertainty was already presented by Chowdhury and Gopalan (2019). The main extension is to make this optimization tractable through a point-wise optimization at every step. Begin optimistic with respect to uncertainty has a long history in RL itself, but less in the context of epistemic uncertainty. As such, it seems a relevant idea, although not groundbreaking.

Reproducibility: Yes

Additional Feedback:


Review 4

Summary and Contributions: This paper introduces a variant of the upper confidence RL algorithm for efficient model based RL. The method introduces a hallucinated control input for the dynamics model which are supposed to be bounded by the agent’s epistemic uncertainty about the transition dynamics (due to lack of data). This reduces an intractable UCRL objective to greedy exploitation over a system with hallucinated inputs. The paper has theoretical analysis showing regret bounds for the algorithm and experiments on continuous control benchmark Mujoco tasks showing comparable or superior performance compared to baselines.

Strengths: The method appears to be novel and connection to previous work is well made. Applying UCRL in a tractable way to model based RL seems significant and useful. The theoretical result appears to be sound and significant but I admit I do not have sufficient background to analyze it fully.

Weaknesses: Section 2.2 overview of exploration methods could use more explanation of the pros and cons of each method and why the authors decided to go with UCRL and what benefits it has over the other methods. The method only shows superior performance on the benchmark mujoco tasks when the action penalty is high (except for Half-Cheetah). It feels like the authors had to tweak the action penalty to show a performance difference. More explanation of why this could be the case would be useful. Otherwise, the significance of the experimental results is perhaps questionable. Another option might be to choose other sparse reward tasks where a performance benefit would be clear without having to tweak parts of the reward (including action penalty) or task. It is not very clear to me why the hallucinated inputs allow you to reduce the UCRL problem to greedy exploitation. Line 183 “can exert much control as the current epistemic uncertainty of the model affords” could use a more intuitive explanation of what this means, how this is implemented, and its implications. The significance of Theorem 1 could be made more clear. How does this theorem relate to practice, how important are the other terms in the bound and what are the assumptions on those terms? Is it possible to empirically verify the bound on a toy domain or constructed MDP?

Correctness: Claims and methods are overall correct.

Clarity: The paper is hard to follow at times especially without sufficient background and relevant reading.

Relation to Prior Work: Relation to prior work is overall thorough and differences are clearly made. This paper is recent but it would be good to discuss relation to [1]. [1] Pacchiano, Aldo & Ball, Philip & Parker-Holder, Jack & Choromanski, Krzysztof & Roberts, Stephen. (2020). On Optimism in Model-Based Reinforcement Learning.

Reproducibility: Yes

Additional Feedback:

[Author Response · NeurIPS 2020]

We thank all reviewers for their valuable feedback and comments. Please find our responses below.

**Reviewer 1** - Explanation in the introduction: we strive for clarity and we appreciate this comment. We will
elaborate on the algorithm description accordingly.

- Relationship to Bayesian RL. While we discussed Bayesian RL in the context of posterior sampling, we will elaborate
on the relation with Dearden et al. (1998) in the revised version. We thank the reviewer for pointing this out.

**Reviewer 2** - Reproducibility: We will release the code upon acceptance of the paper and include the experimental
details in the revised version.

- Full-rollouts: The analysis is not limited to planning on full rollouts. Instead, any algorithm that solves the HUCRL
objective (7) retains the theoretical guarantees. This can be done in many ways as discussed in Appendix C.

- Selection of $\beta$: As stated in Lemmas 10 and 11, $\beta$ has to grow sub-logarithmically with time in order for the confidence
intervals to be correct. The theoretical value used for the bounds is rather conservative however. In practice, we simply
set $\beta = 1$ (without tuning), as it is often done in bandit algorithms. We will clarify this in the experimental setup.

- Comparison to competing methods: Our competing methods are Thompson Sampling and Greedy. SAC/PPO/POPLIN
can be used to solve the greedy planning problem by simulating transitions with the learned model. For example, Greedy
with NN ensembles is the PETS-DS algorithm (which we compare against). POLO and POPLIN are approximate
planning algorithms given a model and, as such, could be used to optimize the HUCRL objective (7).

- Calibration Assumption: although this is crucial for the theory, in practice we verified that the resulting confidence
intervals contained the true predictions (not necessarily calibrated) but did not employ any re-calibration procedure.

- Tasks: We evaluate on the same tasks as in Chua et al. (2018), which have randomly sampled initial conditions. Also,
the sparse inverted pendulum (which is not a Mujoco task) has transitions with additive Gaussian noise.

**Reviewer 3** - Scaling: in lines 190-201 we clarify that we parameterize $\pi$ and $\eta$ and optimize them jointly using a
policy search algorithm. To address instabilities, we could restart the optimization algorithm for $\pi$ and $\eta$ as there is no
need for such policies to be close to each other between episodes. We did not encounter such instabilities in practice.

- Difference to Gopalan & Chowdhury (2019): as per your review "The main extension is to make this optimization
tractable", we find this a big contribution (the previous method is not implementable/practical). Furthermore, we discuss
the differences in Appendix H.3, and in lines 73-78.

- Stochasticity: In the HUCRL algorithm (7), $J$ is the expectation w.r.t. the stochastic uncertainty, we will clarify this.

- Multi-modality: When the next-state distribution is multi-modal the choice of the model likelihood is crucial. We can
still be optimistic w.r.t. the epistemic uncertainty in the model and use a multi-modal stochastic noise model.

- "Dyna" name: We use Monte Carlo sampling with bootstrapping in a receding-horizon fashion to solve objective (7),
which is common as you mention. The difference is that we learn $\pi_\theta$ using samples from the model (Dyna) and then
guide the Monte Carlo sampling for MPC with such a policy, hence Dyna-MPC.

- Discontinuous-reward functions: We meant that discontinuous rewards are known to be hard to optimize (even in the
bandit setting it is NP-hard). We will clarify that this does not mean that discontinuous rewards are impossible to solve.
Note that Atari has discrete states and that rewards are linear functions of the tabular encoding of the state. Furthermore,
in most games there is local reward information, whereas "hard" exploration games were recently solved using optimism.

- Experiments: We will use the extra space of the revised version to include the learning curves of environments.

- Action-penalties: All environments have action penalties. There is a typo in the cheetah reward (l. 681), it is $r =$
$\min(v, 10) - \rho \sum_i a_i^2$. This is common in continuous control as it shapes the intended behaviour to consume less energy.

- Colors: Thanks for catching this. We will augment the colors with line-styles and markers.

**Reviewer 4** - Choice of action penalties/environments: We used the same environments as in PETS by Chua et al.
(2018), which used action penalties. Higher action penalties are commonly used to reduce the energy consumption
while achieving the same goal. Simultaneously, higher action penalties favour zero-action against random actions. Thus
they also penalize exploration when the greedy exploration objective is used.

- Reduction to Greedy: when considering the joint inputs (a, $\eta$), (7) can be solved with standard planning algorithms.

- Significance of Thm 1: i) Having sublinear regret implies that as $T \to \infty$ the average incurred cost converges to the
optimal cost. ii) In Fig. 1, optimism finds a good policy whereas TS and Greedy do not. iii) Assumptions: After each
assumption we clarify whether it is met in practice.

- Reference: This was published after our submission. We will include a reference to it as contemporary work.

[Meta-Review · NeurIPS 2020]

summary: The paper presents a practical algorithm for model-based reinforcement learning that addresses the exploration problem with an optimistic approach. In particular they convert epistemic uncertainty into “hallucinated controls” that are optimized, thereby leading to optimistic behavior. pros: - contribution: tractable algorithm for optimistic RL for continuous state and action spaces - relevant theoretical guarantees (sublinear regret) - important topic - well situated in literature cons: - experimental evaluation not entirely convincing - some reviewers found it potentially somewhat incremental meta review: All reviewers agree that this is a good, potentially impactful paper.